# DAComp: Benchmarking Data Agents across the Full Data Intelligence Lifecycle

**Fangyu Lei**[*,1,2,3] **Jinxiang Meng**[*,1,2] **Yiming Huang**[5] **Junjie Zhao**[3] **Yitong Zhang**[6]
**Jianwen Luo**[1,2] **Xin Zou**[3] **Ruiyi Yang**[3] **Wenbo Shi**[3] **Yan Gao**[3] **Shizhu He**[1,2] **Zuo Wang**[3]
**Qian Liu**[4] **Yang Wang**[3] **Ke Wang**[3,†] **Jun Zhao**[1,2] **Kang Liu**[1,2,†]

[1]The Key Laboratory of Cognition and Decision Intelligence for Complex Systems,
  Institute of Automation, Chinese Academy of Sciences, Beijing, China
[2]School of Artificial Intelligence, University of Chinese Academy of Sciences
[3]ByteDance Seed   [4]TikTok   [5]UC San Diego   [6]NUS

## Abstract

Real-world enterprise data intelligence workflows encompass data engineering that turns raw sources into analytical-ready tables and data analysis that convert those tables into decision-oriented insights. We introduce DAComp, a benchmark of 210 tasks that mirrors these complex workflows. Data engineering (DE) tasks require repository-level engineering on industrial schemas, including designing and building multi-stage SQL pipelines from scratch and evolving existing systems under evolving requirements. Data analysis (DA) tasks pose open-ended business problems that demand strategic planning, exploratory analysis through iterative coding, interpretation of intermediate results, and the synthesis of actionable recommendations. Engineering tasks are scored through execution-based, multi-metric evaluation. Open-ended tasks are assessed by a reliable, experimentally validated LLM-judge, which is guided by hierarchical, meticulously crafted rubrics. Our experiments reveal that even state-of-the-art agents falter on DAComp. Performance on DE tasks is particularly low, with success rates under 20%, exposing a critical bottleneck in holistic pipeline orchestration, not merely code generation. Scores on DA tasks also average below 40%, highlighting profound deficiencies in open-ended reasoning and demonstrating that engineering and analysis are distinct capabilities. By clearly diagnosing these limitations, DAComp provides a rigorous and realistic testbed to drive the development of truly capable autonomous data agents for enterprise settings. Our data and code are available at `da-comp.github.io`.

## 1 Introduction

Data intelligence, the process of transforming raw and fragmented data into actionable insights, has become a cornerstone of modern enterprises. The remarkable reasoning and code generation capabilities of Large Language Models (LLMs) (OpenAI, 2025; Anthropic, 2025; Gemini, 2025) have opened new avenues for automating data intelligence tasks. LLM-based agents have demonstrated considerable promise across a wide range of applications, including text-to-SQL (Yu et al., 2018; Li et al., 2024b; Lei et al., 2024), software engineering (Jimenez et al., 2023; Chan et al., 2024), and general computer control (Zhou et al., 2024; Xie et al., 2024; Wei et al., 2025). However, the advancement of these agents into enterprise data intelligence remains constrained by the absence of benchmarks that faithfully reflect real-world complexity.

This gap between existing benchmarks and real enterprise practice calls for a benchmark that evaluates agents along two distinct axes: *Hard* (engineering realism) and *Soft* (analytical openness). The *Hard* axis reflects the capacity for systematic large-scale code implementation, similar to the responsibilities of data engineers. For example, this means not only generating a single SQL query but also orchestrating and evolving complex data workflows under changing requirements. The

---

[*]Equal contribution.
[†]Corresponding authors.

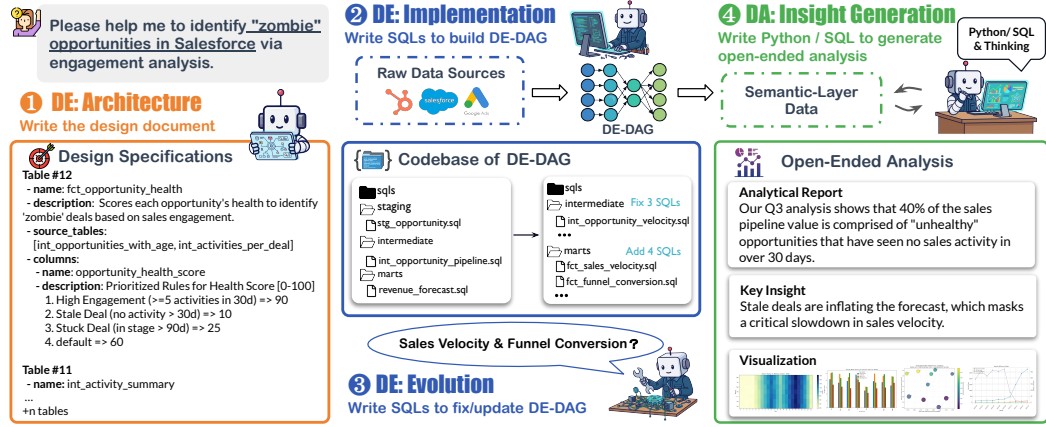

Figure 1: DAComp aims to evaluate LLMs on full-lifecycle data intelligence workflows, encompassing repository-level data engineering (*DE*) and open-ended data analysis (*DA*).

*Soft* axis reflects the capacity for strategic reasoning, aligning more closely with the role of data analysts. For example, this involves facing an open-ended business question, planning multi-step analytical workflows, synthesizing insights across analytical results, generating visualizations and crafting decision-oriented reports. Most benchmarks fail to capture these two key dimensions. They reduce complex engineering to isolated code snippet generation, missing the *Hard* axis, and reduce open-ended analysis to deterministic answers, missing the *Soft* axis.

To fill this gap, we present **DAComp**, benchmarking agents on full lifecycle data intelligence tasks, as illustrated in Fig. 1. **DAComp-DE** is the first to introduce repository-level data engineering tasks where agents must orchestrate multi-layered data workflows by generating a DAG on complex enterprise schemas. It includes three distinct task types: (1) **DE-Arch**itecture tasks focus on the high-level planning of detailed engineering specifications. (2) **DE-Impl**ementation tasks require agents to build multi-stage data pipelines from scratch; (3) **DE-Evolution** tasks challenge them to modify existing systems in response to new requirements; and Both DE-Impl and DE-Evol tasks are demanding, often requiring large-scale code changes that involve over $4,000$ lines of code across more than 30 files, mirroring real-world engineering workloads. **DAComp-DA** is the first to pioneer real-world, open-ended data analysis. In these scenarios, agents are presented with complex questions over downstream analytical data. Unlike prior work with deterministic answers (Jing et al., 2024; Lei et al., 2024), the tasks resemble real analyst settings: agents must write SQL/Python to aggregate, compute, and analyze intermediate results in order to generate insights, reports and visualizations, thereby emphasizing both the rigor of analytical precision and the practical utility for human decision-making. To facilitate broad applicability, we also release **DAComp-zh**, a high-quality Chinese adaptation of the benchmark, along with baseline results.

The evaluation methods of such complex tasks are non-trivial. For deterministic DE-Impl and DE-Evol tasks, we adopt an execution-based method to systematically evaluate the repo-level code generation performance. The open-ended DA and DE-Arch tasks are assessed by an LLM judge (Li et al., 2024a), whose evaluation is guided by our novel rubric framework. Instead of relying on a single answer key, this framework explicitly defines and assesses multiple valid solution paths for each open-ended problem, enabling a robust, multifaceted assessment that rewards diverse analytical strategies. The reliability of this LLM judge has been confirmed through rigorous validation experiments, which show strong agreement with human experts.

Our experiments on DAComp underscore a significant challenge for current models: even state-of-the-art agents falter when confronted with its enterprise-level complexity. In DE tasks, agent capabilities are pushed to their limits, with average scores below 40% and strict success rates under 10%, revealing a critical gap in real repository-level engineering capabilities. In the same vein, agents also exhibit poor performance on open-ended problems requiring autonomous planning. Performance on DA tasks plummets to below 50% for most models, with only a few proprietary systems demonstrating more robust analytical skills. Ultimately, progress in data agents demands a shift from mere code accuracy to the nuanced capabilities—planning, open-ended reasoning, and systematic synthe-

sis—required to deliver insights that are both analytically rigorous and strategically actionable. By providing this rigorous, realistic testbed, DAComp aims to shift the focus of data agent development from isolated skills to the integrated, full-lifecycle capabilities required in the real-world scenarios.

## 2 BENCHMARK CONSTRUCTION

### 2.1 TASK DEFINITION

To bridge this gap, we design tasks that evaluate data agents on real-world challenges. Specifically, we assess their ability to act as data engineers performing *repository-level data engineering* and as data analysts navigating *open-ended data analysis*, as depicted in Fig. 1.

**DAComp-DE.** An agent $\pi^{de}$ is tasked with handling the full DE lifecycle including architecture, implementation, and evolution. Formally, the process is modeled as $(\mathcal{S}, \mathcal{C}_\star) = \pi^{de}(\mathcal{Q}_{de}, \mathcal{C}_0, \mathcal{B})$, where $\mathcal{Q}_{de}$ is the initial high-level requirement, $\mathcal{S}$ denotes the engineering specification (e.g., a Data Contract), $\mathcal{B}$ is the database and $\mathcal{C}_\star$ is the final DE repository. This unified capability is evaluated across three task types: *(1) DE-Arch*: Given a high-level requirement $\mathcal{Q}_{de}$ and an initial repository $\mathcal{C}_0$, this task evaluates the agent's ability to produce the engineering specification $\mathcal{S}$. *(2) DE-Impl*: Given a detailed specification $\mathcal{S}$ and an empty repository ($\mathcal{C}_0 = \varnothing$), this task evaluates the agent's ability to implement the DE repository $\mathcal{C}_\star$ from scratch. *(3) DE-Evol*: Given an existing repository $\mathcal{C}_0$ and a new specification $\mathcal{S}$, this task evaluates the agent's ability to update the repository into $\mathcal{C}_\star$.

**DAComp-DA.** Given an analysis-ready data $\mathcal{D}$ (semantic layer) and an open-ended question $\mathcal{Q}_{da}$, an agent with policy $\pi^{da}$ produces analysis artifacts $\mathcal{O} = \pi^{da}(\mathcal{Q}_{da}, \mathcal{D})$ (e.g., analytical reports, key insights and actionable recommendations). This task is inherently open-ended, as a single question may be approached through multiple valid analytical paths, without a fixed standard answer.

### 2.2 EVALUATION METRICS

**LLM-judge with hierarchical rubrics and GSB scoring.** The LLM judge evaluates outputs $\mathcal{O}$ along six dimensions: *Completeness*, *Accuracy*, *Insightfulness*, *Readability*, *Analytical depth* and *Visualization* (see App. A.3.1). The hierarchical rubric assesses the first three, while the Good–Same–Bad (GSB) score (Zheng et al., 2023) covers the latter three. *Visualization* specifically assesses the agent's ability to translate numerical results into intuitive chart. As shown in Fig. 2, the rubric ($\mathcal{R}$) decomposes a question $\mathcal{Q}$ into requirements and sub-reqirements. Each subrequirement admits multiple valid solution paths, each path carrying its own rubric items (colored leaf nodes). Human experts enumerate these paths and merge equivalent solutions in a single path. For scoring, the LLM judge selects the best-matching path for each sub-requirement, applies only that path's items, then

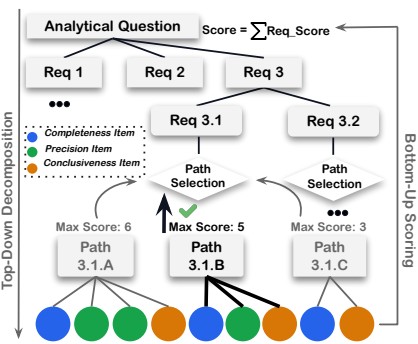

Figure 2: Details of hierarchical rubrics.

aggregates scores bottom-up. This design accommodates diverse correct approaches without penalizing method choice. We show a detailed rubric example for the penetration and profitability analysis in Tab. 11, with a discussion of the path enumeration scheme provided in App.G.1. The rubric score is a normalized, weighted sum of satisfied items: $\text{Score}_{rubric}(\mathcal{O}, \mathcal{R}) = \frac{\sum_{k=1}^N s_k}{\sum_{k=1}^N w_k}$, $s_k = \Lambda(c_k, \mathcal{O}) \in [0, w_k]$. For the Good-Same-Bad (GSB), the LLM judge only compares the final analytic results against five pre-provided baseline reports, guided by the dedicated rubrics for these axes, yielding the score: $\text{Score}_{gsb}(\mathcal{O}, \mathcal{O}_{base}) = \frac{\max(0, |G| - |B|)}{|G| + |S| + |B|}$. The final score for a DA task is a weighted combination of these two components: $\text{Score}_{da} = \alpha \cdot \text{Score}_{rubric} + (1 - \alpha) \cdot \text{Score}_{gsb}$. The open-ended DE-Arch tasks are assessed similarly, though they employ a standard, non-hierarchical rubric and do not incorporate the GSB component. Further details are provided in App. A.

**Execution-based evaluation for deterministic tasks.** DE-Impl and DE-Evol tasks are evaluated with three execution-based metrics of increasing strictness: (1) the partial credit *Component Score*

*(CS)*, $\mathrm{CS}_{\text{DE-Impl/Evol}} = \sum_j w_j s_j$, which evaluates each node *in isolation* (using gold-standard upstream inputs) to measure total component-level SQL generation; (2) the *Cascading Failure Score (CFS)*, which evaluates nodes *sequentially along the DAG* and nullifies a node's score if any upstream dependency is incorrect, thus measuring end-to-end data integrity; and (3) the strict *Success Rate (SR)*, $\mathrm{SR}_{\text{DE-Impl/Evol}} = \mathbb{I}[\forall j : s_j = 1]$, which requires every single component to be perfect. This suite of metrics is crucial for diagnosing the primary bottleneck: the gap between an agent's component-level generation and its ability to perform holistic pipeline orchestration. Further details are provided in App. A.1.

### 2.3 ANNOTATION PIPELINE

DAComp is constructed by 8 experts through a rigorous pipeline to ensure realism, quality, and consistency. Further details and examples are provided in App. E.

**1) Data collection.** The benchmark is grounded in permissively licensed assets (e.g., Apache-2.0, MIT). For the DE task, we collect 73 enterprise-scale SaaS schemas with data transformation projects, averaging 400 columns each, and populate them with large-scale, relationally consistent synthetic data (see App. E). For the DA task, we curate 100 complex databases from the Web and supplement them with analytical modeling layers derived from DE-transformed data.

**2) Task design.** At this stage, we generate the DAComp questions. For DA , annotators first draft 8 open-ended analytical questions per analysis-ready table. Five annotators then vote based on realism and difficulty, and the top 2 are retained. For DE-Evol , practicing data engineers author new business requirements aligned with enterprise scenarios and professional standards. For DE-Impl , we reverse engineer selected SaaS transformation projects into a single `data_contract.yaml`, capturing the full DAG and semantics. For DE-Arch , starting from the analytics layer of DE-Impl and DE-Evol examples, DA annotators propose 5 candidate business requirements per project, from which a data engineer selects 1 feasible yet challenging requirement.

**3) Evaluation construction.** We design evaluation protocols for each task. For *DA*, annotators build hierarchical rubrics as described in §2.2, with at least 3 annotators annotate each question, followed by alignment discussion to resolve discrepancies. For the *GSB* protocol, experienced data analysts author shared scoring criteria, and baseline reports are created by combining outputs from multiple LLMs. A critical aspect of this rubric design is the enumeration of valid solution *Paths*, a process governed by three key principles: (i) ensuring Paths represent distinct, methodologically-sound strategies, not incremental steps; (ii) validating deterministic outputs against programmatically calculated and verifiable anchor values; and (iii) utilizing methodology-based soft constraints to fairly evaluate valid but unenumerated solution paths. (see examples in App.C.4, discussion in App.G.1). To ensure the comprehensiveness of our rubric, we perform a validation step: we sample outputs from five diverse LLMs and confirm that our enumerated `paths` can account for all observed solution strategies, which minimizes the risk of false negatives by ensuring that valid but unanticipated solutions are not unfairly penalized. For DE-Impl and DE-Evol , solutions are deterministic: we implement execution scripts to automatically validate outputs against gold repositories, assigning partial credit at the node/layer level to capture step-wise correctness.

### 2.4 DATASET STATISTICS

We present a statistical analysis of DAComp, highlighting its main features in comparison with prior datasets in Tab. 1, and providing more detailed characteristics in Tab. 2.

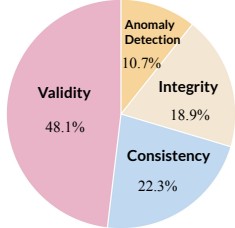

Figure 3: Data cleaning tasks of DE-Impl staging layer.

**DAComp-DE quantifies enterprise-scale engineering complexity.** The statistics for DAComp-DE underscore its large scale and complexity—defined by its repo-level paradigm, schemas averaging 412 columns, and solutions requiring over 2,000 lines of code—setting it apart from prior data agent benchmarks. Unlike benchmarks that focus on generating isolated scripts, DAComp introduces tasks on industrial schemas with an average of 32 tables and 412 columns. The engineering effort required is substantial. Implementation tasks involve building entire pipelines from scratch, averaging 4,612 lines of code across 43 distinct files. Evolution tasks simulate realistic maintenance with edits averaging 1,718 LOC across

Table 1: Comparison of DAComp with other agent benchmarks, highlighting key differences in task scope, task paradigm, and evalution method. DAComp-zh shares the identical task set.

| Benchmark | Field | # Tasks | Repo-Level | # Cols/ Schema | Code Scale (LOC) | Primary Output | Open-ended | Evaluation Method |
|---|---|---|---|---|---|---|---|---|
| *Agentic Benchmarks* | | | | | | | | |
| SWE-Bench (Jimenez et al., 2023) | Software Engineering | 2,294 | ✓ | N/A | 32.8 | Code Patch | ✗ | Execution-based |
| WebArena (Zhou et al., 2024) | Web Navigation | 812 | ✓ | N/A | N/A | Actions | ✗ | Execution-based |
| OSWorld (Xie et al., 2024) | Computer Control | 369 | ✓ | N/A | N/A | Actions | ✗ | Execution-based |
| BrowserComp (Wei et al., 2025) | Deep Research | 2,000 | ✓ | N/A | N/A | Answer | ✓ | Objective |
| *Data Agent Benchmarks* | | | | | | | | |
| DS-1000 (Lai et al., 2023) | Data Science | 1,000 | ✗ | N/A | 3.6 | 1 Script | ✗ | Execution-based |
| BIRD (Li et al., 2024b) | Text-to-SQL | 12,751 | ✗ | 54 | 23.5 | 1 SQL | ✗ | Execution-based |
| Spider 2.0 (Lei et al., 2024) | Text-to-SQL | 632 | ✗ | 320 | 104.6 | 1 SQL | ✗ | Execution-based |
| BIRD-CRITIC (Li et al., 2025) | SQL Debugging | 1,100 | ✗ | 54 | 50~70 | 1 SQL | ✗ | Execution-based |
| DA-Code (Huang et al., 2024) | Data Science | 500 | ✗ | 50 ~ 100 | 85 | 1 Script | ✗ | Objective |
| DSBench (Jing et al., 2024) | Data Science | 540 | ✗ | 27 | 10~20 | N Scripts | ✗ | Objective |
| KramaBench (Lai et al., 2025) | Data Science Pipelines | 104 | ✗ | 13 | 50~100 | N Scripts | ✓ | LLM-judge |
| BLADE (Gu et al., 2024) | Data Analysis | 259 | ✗ | 10 ~ 12 | 70~80 | Report | ✓ | LLM-judge |
| DABStep (Egg et al., 2025) | Data Analysis | 450 | ✗ | 10 ~ 12 | 100 | Answer | ✗ | Objective |
| **DAComp** | **Data Engineering & Data Analysis** | 210 | ✓ | 382 | ∼ 2,000 | **Doc + Report N SQL/Script** | **Both** | **Execution-based & LLM-judge(rubrics)** |

Table 2: Key statistics for DAComp. All metrics are per-example averages, except #Total tasks.

| Metric | Value | Metric | Value |
|---|---|---|---|
| **Overall (DE-Arch/DE-Impl/DE-Evol/DA)** | | **DAComp-DE** | |
| #Total tasks | 30 / 30 / 50 / 100 | DE-Impl raw data (#Tab. / #Col.) | 23.3 / 381.6 |
| #Question Tokens | 166 / 30,883 / 6,508 / 90 | #LOC code scale (Impl / Evol) | 2,296 / 949.6 |
| **DAComp-DA** | | #Change files (Impl / Evol) | 37.0 / 11.7 |
| Columns / Tables | 84.7 / 3.9 | #Change columns (Impl / Evol) | 1,239 / 530.9 |
| LOC | 433 | #DE-Arch rubric | 18.5 |
| Rubrics (Reqs / Sub-reqs / Paths / Items) | 3.1 / 5.7 / 12.7 / 22.4 | DE-Impl layer (#Staging / #Core / #Mart) | 16.0 / 11.8 / 8.8 |
| Completeness / Accuracy / Insightfulness | 14% / 66% / 20% | DE-Evol table change types (#create / #edit) | 3.76 / 7.90 |

13 files, agents need to manage data transformation across a multi-layered data model (**staging**, **core**, and **mart**). The staging layer involves **data cleaning** operations, a central topic in data governance, which we categorize into four types: validity constraints, consistency constraints, integrity & uniqueness, and anomaly detection (as shown in Fig. 3). Intermediate and marts layers typically focus on complex business logic, entity integrations, and metric aggregations.

**DAComp-DA measures analytical depth and methodological diversity.** The design of DAComp-DA moves beyond simple question-answering to assess deep analytical reasoning. Uniquely, DAComp evaluates both deterministic engineering and open-ended analysis, a distinction from prior benchmarks that typically focus on only one paradigm. Its open-ended nature is quantified by our hierarchical rubrics, which decompose each of the 100 DA tasks into an average of 3.1 requirements and 5.7 sub-requirements, accommodating roughly 13 valid solution paths. This methodological diversity is evaluated with a multi-faceted rubric where scoring items are weighted toward Accuracy (66%) but also reward Completeness (14%) and Insightfulness (20%). While the analytical schemas are more focused than in DE tasks (averaging 4 tables and 85 columns), the required reasoning is still complex, reflected in an average solution length of 347 lines of code—significantly longer than typical text-to-SQL or single-script data science tasks. Crucially, DAComp-DA places a strong emphasis on **open-ended data visualization**, requiring agents to autonomously select and generate charts that effectively communicate their findings.

## 3 EXPERIMENTS

### 3.1 EXPERIMENTAL SETUP

We evaluate state-of-the-art LLMs, including open-source models like Qwen3 (Yang et al., 2025), DeepSeek-V3.1 (Liu et al., 2024), and Kimi-K2 (Team et al., 2025), as well as proprietary ones such as the Gemini (Team et al., 2023), and GPT (OpenAI, 2023) families. We utilize the widely adopted OpenHands (CodeAct-Agent) framework (Wang et al., 2024) for both DE and DA tasks. Additionally, we developed a custom baseline named **DA-Agent** for DAComp-DA, which operates via Bash and file system interactions and is capable of executing Python and SQL. The performance

Table 3: **DAComp-DE** Baseline Performance. All models are evaluated using the **DE-Agent** framework (details in App. B.2) across both **Implementation** (CFS, Max-CFS@8, CS, Max-CS@8) and **Evolution** (SR@8, CFS, Max-CFS@8); see App. A.1 for metric definitions. The final column reports the aggregated **DE Score**.

| Method | Architecture | Implementation | | | | Evolution | | | DE Score |
|---|---|---|---|---|---|---|---|---|---|
| | | CFS | Max-CFS@8 | CS | Max-CS@8 | CFS | Max-CFS@8 | SR@8 | |
| GPT-5 | 63.93(±2.33) | 30.79 | 39.87 | 61.98 | 68.77 | 38.75 | 47.23 | 20.00 | 43.45 |
| Gemini-2.5-Pro | 51.96(±1.78) | 27.66 | 36.88 | 55.32 | 65.32 | 23.97 | 38.92 | 8.00 | 32.88 |
| Qwen3-Coder | 51.43(±3.14) | 23.64 | 32.86 | 54.21 | 63.78 | 27.12 | 39.77 | 12.00 | 32.80 |
| DeepSeek-V3.1 | 52.66(±2.88) | 22.33 | 30.73 | 50.04 | 60.46 | 24.11 | 35.01 | 10.00 | 31.41 |
| o3 | 48.32(±2.13) | 15.07 | 22.32 | 35.55 | 47.81 | 24.42 | 32.07 | 6.00 | 28.39 |
| Qwen3-235B-A22B | 50.73(±2.05) | 2.43 | 5.77 | 20.15 | 31.03 | 12.43 | 21.89 | 2.00 | 20.15 |
| Qwen3-8B | 45.12(±2.06) | 1.31 | 2.34 | 15.33 | 21.23 | 15.89 | 19.12 | 2.00 | 19.89 |

Table 4: **DAComp-DE-zh(Chinese)** Baseline Performance.

| Method | Architecture | Implementation | | | | Evolution | | | DE Score |
|---|---|---|---|---|---|---|---|---|---|
| | | CFS | Max-CFS@8 | CS | Max-CS@8 | CFS | Max-CFS@8 | SR@8 | |
| GPT-5 | 63.60(±2.14) | 30.49 | 39.24 | 61.85 | 68.43 | 37.88 | 46.91 | 20.00 | 42.88 |
| Gemini-2.5-Pro | 51.90(±3.43) | 26.98 | 36.73 | 55.18 | 65.07 | 24.28 | 38.27 | 8.00 | 32.55 |
| Qwen3-Coder | 51.11(±3.35) | 23.23 | 32.97 | 54.59 | 63.69 | 26.59 | 39.37 | 12.00 | 32.36 |
| DeepSeek-V3.1 | 53.08(±2.54) | 22.62 | 30.84 | 50.22 | 60.34 | 24.69 | 35.17 | 8.00 | 31.87 |
| o3 | 48.02(±1.79) | 15.00 | 22.15 | 35.10 | 47.45 | 24.23 | 32.59 | 6.00 | 28.20 |
| Qwen3-235B-A22B | 50.61(±2.50) | 2.31 | 5.83 | 20.03 | 31.27 | 13.01 | 21.27 | 0.00 | 20.35 |
| Qwen3-8B | 46.22(±1.90) | 1.21 | 2.16 | 15.78 | 21.59 | 15.19 | 19.35 | 0.00 | 19.84 |

Table 5: Detailed performance breakdown on the **DAComp-DA** benchmark.

| Method | Completeness | Accuracy | Insightfulness | Readability | Analytical Depth | Visualization | DA Score |
|---|---|---|---|---|---|---|---|
| *OpenHands Baseline* | | | | | | | |
| GPT-5 | 60.98 | 40.3 | 49.39 | 35.51 | 69.8 | 21.4 | 46.99 |
| Gemini-2.5-Pro | 45.02 | 30.22 | 40.71 | 48.2 | 31.0 | 15.0 | 33.38 |
| o3 | 40.13 | 25.5 | 20.45 | 26.22 | 27.11 | 6.8 | 26.57 |
| DeepSeek-V3.1 | 49.88 | 33.25 | 41.66 | 36.0 | 33.2 | 11.0 | 33.87 |
| Qwen3-Coder | 33.42 | 21.21 | 25.06 | 20.0 | 13.73 | 4.8 | 24.28 |
| Qwen3-235B-A22B | 30.7 | 12.23 | 22.11 | 3.6 | 1.8 | 0.8 | 12.43 |
| *DA-Agent Baseline* | | | | | | | |
| GPT-5 | 64.23(±2.37) | 43.81(±3.43) | 56.89(±6.48) | 43.59(±6.08) | 76.80(±4.91) | 27.44(±4.44) | 50.84(±3.12) |
| Kimi-K2 | 52.31(±1.13) | 33.56(±2.09) | 46.82(±2.48) | 62.20(±3.01) | 63.75(±2.84) | 14.40(±2.33) | 41.89(±1.78) |
| Gemini-2.5-Pro | 45.43(±1.34) | 30.30(±0.27) | 41.45(±0.71) | 51.60(±2.73) | 35.75(±2.35) | 13.40(±2.94) | 34.70(±1.39) |
| DeepSeek-V3.1 | 48.74(±2.09) | 32.97(±1.40) | 42.43(±1.89) | 37.25(±2.21) | 35.00(±1.57) | 11.45(±1.31) | 34.33(±0.45) |
| o3 | 40.73(±0.63) | 29.54(±2.93) | 23.95(±3.86) | 25.24(±2.51) | 23.81(±3.37) | 7.32(±1.27) | 28.20(±1.37) |
| Qwen3-Coder | 35.12(±2.21) | 20.05(±2.35) | 25.53(±1.83) | 19.37(±1.44) | 13.42(±2.38) | 5.15(±0.85) | 25.13(±0.82) |
| Doubao-Seed-1.6 | 37.45(±1.95) | 18.45(±2.55) | 27.51(±2.00) | 13.25(±2.48) | 9.01(±1.25) | 6.80(±1.96) | 20.74(±0.82) |
| Qwen3-235B-A22B | 29.37(±1.09) | 13.11(±1.33) | 21.50(±1.81) | 3.64(±0.33) | 1.56(±0.81) | 1.87(±0.78) | 13.25(±0.65) |
| Qwen3-8B | 9.89(±2.46) | 4.12(±0.32) | 5.05(±1.70) | 0.13(±0.15) | 0.00(±0.00) | 0.15(±0.19) | 4.47(±0.63) |

of each agent is measured using the metrics detailed in §2.2. We also report two aggregate scores: the DE Score, which is the mean score across all DE tasks (using CFS for Implementation/Evolution), and the Overall Score, representing the mean across the entire benchmark. For the DA score, we use $\alpha = 0.6$ to aggregate the rubric and GSB scores, with Gemini-2.5-Flash serving as the LLM judge. Further details on the experimental setup and additional results are provided in App. B.

## 3.2 MAIN RESULTS

**DE results.** As shown in Tab. 4, GPT-5 establishes a definitive lead, consistently achieving the highest aggregated DE Scores across different orchestration frameworks. Notably, specialized open-source models like Qwen3-Coder and DeepSeek-V3.1 demonstrate exceptional efficacy, effectively rivaling general-purpose proprietary models such as Gemini-2.5-Pro. However, the absolute performance metrics reveal a sobering reality regarding the complexity of repository-level engineering: even the state-of-the-art GPT-5 achieves a modest DE Score of approximately 42.88% and a strict Success Rate of merely 20.00%. This profound performance ceiling underscores that while framework optimizations can stabilize interaction, DAComp-DE poses a rigorous challenge that current LLMs—regardless of their scale or specialization—have yet to master, highlighting a critical gap between isolated code generation and holistic system orchestration.

Table 6: Detailed performance breakdown on the **DAComp-DA-zh (Chinese)** benchmark.

| Method | Completeness | Accuracy | Insightfulness | Readability | Analytical Depth | Visualization | DA Score |
|---|---|---|---|---|---|---|---|
| ***OpenHands Baseline*** | | | | | | | |
| GPT-5 | 70.56 | 47.08 | 57.19 | 19.6 | 46.4 | 22.0 | 43.69 |
| Gemini-2.5-Pro | 55.51 | 29.9 | 47.17 | 38.8 | 18.8 | 10.2 | 31.22 |
| o3 | 49.79 | 30.73 | 40.74 | 17.55 | 10.61 | 8.2 | 27.87 |
| DeepSeek-V3.1 | 54.5 | 32.93 | 42.56 | 8.2 | 5.0 | 3.6 | 24.16 |
| Qwen3-Coder | 43.14 | 20.38 | 25.69 | 2.47 | 1.1 | 2.04 | 21.84 |
| Qwen3-235B-A22B | 29.44 | 14.27 | 17.35 | 1.22 | 0.0 | 0.98 | 11.5 |
| ***DA-Agent Baseline*** | | | | | | | |
| GPT-5 | 72.69(±1.41) | 46.96(±1.94) | 61.56(±2.51) | 39.35(±2.19) | 66.40(±3.43) | 25.40(±1.87) | 49.49(±1.04) |
| Gemini-2.5-Pro | 54.63(±2.53) | 33.33(±1.58) | 48.56(±0.50) | 49.95(±3.84) | 26.20(±2.47) | 9.00(±3.52) | 33.75(±1.67) |
| Kimi-K2 | 57.08(±0.55) | 33.54(±2.99) | 47.64(±1.32) | 34.52(±2.35) | 20.28(±3.07) | 3.86(±2.14) | 31.22(±0.75) |
| o3 | 51.10(±1.75) | 30.68(±2.97) | 34.92(±1.29) | 20.00(±0.57) | 12.54(±2.54) | 6.35(±1.22) | 28.70(±1.15) |
| DeepSeek-V3.1 | 55.15(±2.49) | 34.01(±2.36) | 44.62(±2.89) | 7.15(±1.98) | 4.65(±2.00) | 6.30(±2.42) | 27.75(±2.04) |
| Qwen3-Coder | 43.35(±1.76) | 22.75(±3.15) | 30.83(±2.38) | 4.07(±0.98) | 1.55(±1.02) | 1.75(±0.50) | 22.64(±1.19) |
| Doubao-Seed-1.6 | 45.92(±2.07) | 18.73(±2.05) | 33.23(±1.06) | 3.23(±1.12) | 0.75(±0.68) | 1.55(±0.66) | 17.83(±1.33) |
| Qwen3-235B-A22B | 31.64(±2.71) | 13.48(±0.19) | 22.27(±1.22) | 0.87(±0.64) | 0.13(±0.12) | 0.33(±0.42) | 12.74(±0.33) |
| Qwen3-8B | 14.55(±1.04) | 6.30(±2.18) | 6.08(±2.15) | 0.00(±0.00) | 0.00(±0.00) | 0.00(±0.00) | 6.33(±1.25) |

**DA results.** The results in Tab. 6 reveal a significant capability gap in open-ended analysis, with the top overall score solely reaching 56.14%. A dimension-wise analysis uncovers three critical insights. First, *Analytical Depth* and *Insightfulness* serve as the primary differentiators between tiers. While GPT-5 dominates by maintaining high scores across all dimensions, reasoning-focused models like o3 exhibit a distinct "calculator behavior": despite achieving competitive *Accuracy* (40.99) and *Completeness* (60.73), o3 suffers severely in *Readability* (24.63) and *Depth* (13.37), indicating an ability to compute correct numbers but a failure to synthesize them into human-readable insights. Second, the gap between DeepSeek-V3.1 (39.16%) and the code-specialized Qwen3-Coder (28.07%) is driven largely by qualitative metrics; Qwen3-Coder nearly collapses on *Readability* (3.15) and *Visualization* (1.93), suggesting that open-ended analysis requires holistic reasoning beyond mere SQL generation. Finally, the task complexity establishes a strict capacity threshold, where smaller models like Qwen3-8B fail to generate coherent analytical artifacts.

## 3.3 Performance Analysis of Repository-level Data Engineering

**Holistic orchestration is the core bottleneck in data engineering.** Across DE tasks, models plan well but struggle to execute end-to-end. Evolution scores are relatively high (e.g., GPT-5: $37 \sim 38\%$), yet strict SR for Evolution are much lower (typically $< 20\%$). The drop from component-level correctness (CS) to cascading failure scores (CFS) is pronounced for strong models, revealing a pipeline-level orchestration bottleneck beyond single-file correctness; for example, GPT-5 (DAComp-DE-Agent) in Implementation falls from CS 61.85 to CFS 30.49, and in Evolution from CFS 37.88 to SR 20.00. By contrast, weaker open-source models (e.g., Qwen3-8B) exhibit very low CS (Implementation 1.21), indicating deficits already at the component level; orchestration then compounds failure but is not the sole cause. The uniformly low CFS across models confirms that coordinating dependencies in a live repository—rather than generating isolated correct code—is the dominant challenge in DAComp-DE.

**Medium-scale code edits are the most difficult to perform.** To gain a more granular understanding, we delve into a node-level analysis, studying the scores for individual SQL file modifications (Fig. 4). We classify these modifications into two types—**editing** an existing file or **creating** a new file, and group them by the required number of lines. For **create** tasks, models like GPT-5 have a clear "sweet spot" on medium-scale creations ($20 - 150$ lines), while all models struggle with very large files ($> 150$ lines). In In contrast, **edit** tasks exhibit a non-linear difficulty trend. Contrary to intuition,

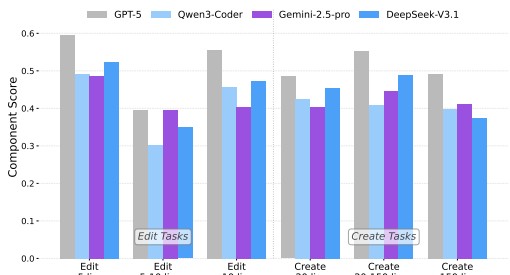

Figure 4: Component-level performance analysis.

medium-scale edits prove to be the most challenging. This is because minor edits are often triv-

ial, while very large edits frequently involve repetitive, boilerplate transformations with clear logic. In contrast, medium-scale edits tend to contain the most complex and nuanced changes to business logic, aggregations, and calculations, thus posing the greatest reasoning challenge.

**Analytical complexity and failure rates escalate in higher pipeline layers.** Fig. 5 reveals that the difficulty of data engineering tasks escalates significantly as agents move from the initial data ingestion layer to the more complex analytical layers. The *staging* layer, focused on basic cleaning, consistently has the fewest local errors and the highest task survival rate. The challenge intensifies dramatically in the *intermediate (core)* layer. This is where the most complex business logic and entity integration occurs, and as

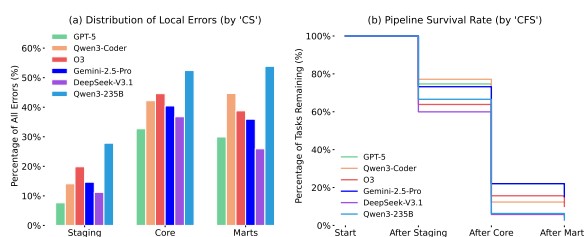

Figure 5: Error distribution (left), pipeline survival rate (right).

Panel (a) shows, it is where the largest share of local errors originates. The severe impact of this difficulty is evident in Panel (b), which shows the sharpest drop in pipeline survival occurring after this stage. Finally, the *marts* layer remains highly challenging. Failures in this final stage are often a direct consequence of inheriting upstream errors from the *core* layer, with fewer than 20% of the initial tasks surviving to completion. Together, these results demonstrate a clear hierarchy of difficulty, with the analytical complexity of the *core* and *marts* layers posing a substantially greater challenge than the initial *staging* layer.

**Top-performing agents exhibit stable and task-aligned interaction patterns.** Fig. **??** shows the distribution of interaction turns in DE tasks. High-performing models such as GPT-5 maintain moderate turn counts with compact variance across both Implementation and Evolution settings, reflecting efficient yet sufficiently thorough reasoning. In contrast, weaker models like Qwen3 either generate excessively long and volatile traces in Implementation or display unusually short traces in Evolution, where premature termination often corresponds to incorrect or incomplete outputs. These patterns indicate that stable and centered turn distributions are more characteristic of effective agents than simply minimizing the number of turns.

## 3.4 ANALYSIS OF OPEN-ENDED DATA ANALYSIS TASKS

**Performance across analytical objectives.** To investigate how performance correlates with the nature of the analytical task, we manually classify each DA task into five categories based on its primary objective: Descriptive, Diagnostic, Strategic, Pattern Recognition, and Profiling (see App. C.4 for definitions). As shown in Fig. 6, this classification reveals a distinct performance hierarchy. Agents excel at concrete Descriptive tasks (*what happened?*), but their scores drop sharply on more abstract Diagnostic (*why did it happen?*) and Strategic (*what should we do?*) tasks. This confirms that these more complex objectives are not only more challenging but also serve as better differentiators of advanced model capabilities.

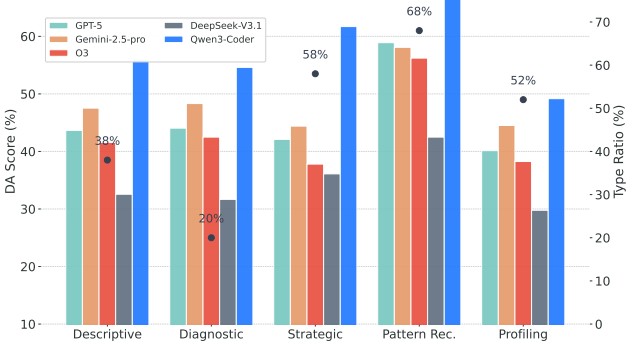

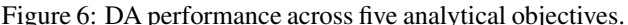

Figure 6: DA performance across five analytical objectives.

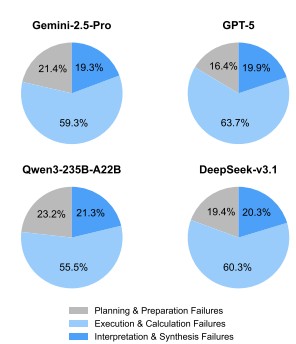

Figure 7: DA error distribution.

**Error analysis.** As shown in Fig. 7, we classify DA failures into three stages: Planning, Execution, and Interpretation. The quantitative breakdown reveals a consistent hierarchy of difficulty across all models: **Execution & Calculation Failures** dominate the error distribution, averaging **59.05%** of all failures. This underscores that the primary bottleneck for current agents lies in *calculation accuracy* and *code grounding* capabilities. However, the challenges are not solely technical; Planning (**20.65%**) and Interpretation (**20.30%**) remain significant sources of error. Collectively, these cognitive stages account for two-fifths of the total performance gap, suggesting that while enhancing execution robustness is the most pressing priority, achieving reliable autonomous analysis requires holistic improvements across the full lifecycle—from initial requirement decomposition to the final synthesis of insights.

## 3.5 VALIDATION OF LLM-JUDGE METHOD

To rigorously validate the reliability of our evaluation framework, we conduct extensive analyses across four dimensions: human-model alignment, cross-judge consistency, stochastic stability, hyperparameter robustness with 50 examples.

**Human-model alignment.** To validate our LLMs-as-Judge method, we conduct a large-scale agreement study on a dataset of 300 model responses generated by 8 distinct LLMs. These responses were manually annotated by expert humans against over 7,000 specific rubric items and GSB documents. We establish a reliable ground truth by measuring inter-rater agreement, which yielded high consistency scores (e.g., Rubric case ICC=0.925, Item $\kappa_w$=0.906), confirming the robustness of our human baseline (Tab. 7). With this human baseline, we benchmark several candidate

Table 7: Inter-rater and human–model agreement. (Details in App.B.4)

| Model / Metric | Rubric ($N$=300, 7k items) | | | GSB Item ($N$=600 pairs) | | |
|---|---|---|---|---|---|---|
| | Item ($\kappa_w$) | Case (ICC(A,1)) | Model ($\tau_b$) | Read. ($\kappa_w$) | Prof. ($\kappa_w$)) | Vis. ($\kappa_w$) |
| Human Inter | 0.906 | 0.925 | 1.000 | 0.601 | 0.751 | 0.753 |
| o4-mini | 0.827 | 0.881 | 1.000 | 0.609 | 0.758 | 0.742 |
| Gemini-2.5-Flash | 0.834 | 0.890 | 1.000 | 0.604 | 0.759 | 0.735 |
| GPT-4.1 | 0.797 | 0.848 | 1.000 | 0.596 | 0.786 | 0.748 |
| Gemini-2.5-Pro | 0.808 | 0.878 | 1.000 | 0.602 | 0.765 | 0.751 |
| Kimi-K2-Thinking | 0.808 | 0.872 | 1.000 | 0.575 | 0.732 | – |
| DeepSeek-V3.1 | 0.782 | 0.870 | 1.000 | 0.588 | 0.725 | – |
| Qwen3(-VL)-235B | 0.737 | 0.758 | 1.000 | 0.531 | 0.713 | 0.682 |
| Qwen3(-VL)-30B | 0.680 | 0.775 | 1.000 | 0.507 | 0.691 | 0.656 |

judges (e.g., Gemini 2.5 Flash, o4-mini, GPT-4.1) at three primary levels of agreement: (i) *case-level agreement*, which measures how consistently the judge scores a single task compared to human experts; (ii) *model-level agreement*, which validates whether the judge's final ranking of all models matches the human-derived leaderboard; and (iii) *item-level agreement*, which evaluates the consistency of atomic judgments between the model and human experts at the granularity of individual rubric items or GSB document pairs. As shown in Tab. 7, **Gemini-2.5-Flash** demonstrates exceptional alignment, achieving the highest Rubric Item $\kappa_w$ (0.834) and Case ICC (0.890) among all models, effectively matching human-level *reliability*. While GSB Readability scores show expected variance due to subjectivity ($\kappa_w \approx 0.53$), the judge maintains high precision on objective dimensions like *depth* and *visualization*, justifying its selection as our standard evaluator.

Table 8: Ranking stability across judges. High correlations ($\tau_b$) confirm leaderboard robustness against family bias.

| Agent Model | Primary | Alternative Judges | | | |
|---|---|---|---|---|---|
| | Flash | Pro | GPT-4.1 | Qwen-235B | Qwen-30B |
| GPT-5 | 56.14 | 59.52 | 63.37 | 71.57 | 53.72 |
| o3 | 36.08 | 40.08 | 44.25 | 50.76 | 31.63 |
| Gemini-2.5-Pro | 39.46 | 45.69 | 50.98 | 55.48 | 35.70 |
| DeepSeek-V3.1 | 39.16 | 44.68 | 50.61 | 54.58 | 41.44 |
| Qwen3-Coder | 28.07 | 32.12 | 36.14 | 43.79 | 25.86 |
| Qwen3-235B | 18.84 | 20.85 | 21.77 | 23.81 | 18.30 |
| Kimi-K2 | 36.94 | 43.77 | 47.83 | 53.55 | 32.93 |
| **Rank Corr. ($\tau_b$)** | — | **1.00** | **1.00** | **1.00** | **0.90** |

Table 9: Ranking stability across weighting hyperparameters ($\alpha$). Results show perfect invariance ($\tau_b = 1.00$).

| Agent Model | Primary | Alternative $\alpha$ | | |
|---|---|---|---|---|
| | $\alpha = 0.6$ | $\alpha = 0.5$ | $\alpha = 0.8$ | $\alpha = 0.9$ |
| GPT-5 | 56.79 | 52.14 | 58.30 | 60.49 |
| o3 | 36.33 | 30.45 | 39.89 | 43.86 |
| Gemini-2.5-Pro | 39.36 | 34.36 | 42.05 | 44.83 |
| DeepSeek-V3.1 | 33.82 | 26.86 | 38.33 | 43.54 |
| Qwen3-235B | 18.84 | 14.39 | 21.69 | 24.98 |
| **Rank Corr. ($\tau_b$)** | — | **1.00** | **1.00** | **1.00** |

**Cross-judge consistency.** To rigorously mitigate concerns regarding family-specific bias (e.g., self-preference) and verify leaderboard reproducibility, we conducted a ranking stability analysis using a diverse set of proprietary and open-source judges. As presented in Tab. 8, the relative rankings of agents exhibit exceptional consistency, achieving perfect correlation ($\tau_b = 1.00$) across the majority of evaluators. Crucially, evaluating the Gemini agent with non-Gemini judges (e.g., GPT-4.1) yields an identical ranking position, effectively refuting the hypothesis of family bias.

Consequently, given that the choice of judge model does not statistically alter the leaderboard, we standardize on **Gemini-2.5-Flash** for its superior balance of stability and cost-efficiency.

**Hyperparameter robustness.** The final DA score is a weighted aggregation: $Score_{da} = \alpha \cdot Score_{rubric} + (1 - \alpha) \cdot Score_{gsb}$. While DAComp's granular dimensional design allows developers to adjust $\alpha$ according to their specific preference for accuracy versus presentation, we standardize on $\alpha = 0.6$ for general jiu to ensure that objective technical correctness (Rubric) remains the dominant factor. To verify the validity of this choice, we conduct a sensitivity analysis across configurations ($\alpha \in \{0.5, 0.8, 0.9\}$). As detailed in Tab. 9, the relative rankings remain invariant ($\tau_b = 1.00$) across all settings, demonstrating that our generalized standard is robust while offering flexibility for specialized use cases.

**Stochastic stability.** To assess the reproducibility of our scoring mechanism, we quantify the variability arising specifically from the LLM judge's stochasticity. We performed 8 independent grading runs on a fixed set of identical agent responses. As shown in Tab. 10, the standard deviations of the final scores are consistently negligible ($< 0.35$), demonstrating that our evaluation protocol yields statistically stable and reproducible grades despite the inherent randomness of LLM generation.

Table 10: Variability of scores across 8 independent grading runs on fixed outputs (mean ± std).

| Model | DE-Arch | DA |
|---|---|---|
| GPT-5 | 61.3 ± 0.18 | 56.1 ± 0.16 |
| DeepSeek-V3.1 | 53.2 ± 0.25 | 39.1 ± 0.22 |
| Gemini 2.5 Pro | 51.0 ± 0.21 | 39.4 ± 0.22 |
| O3 | 54.8 ± 0.19 | 36.1 ± 0.20 |
| Qwen3-235B | 50.4 ± 0.31 | 18.8 ± 0.29 |

## 4 RELATED WORK

**Agentic benchmarks.** As LLM-based agents mature, benchmarks span tool use (Yao et al., 2024), software engineering (Jimenez et al., 2023; Zan et al., 2025), mobile interaction (Rawles et al., 2024), web navigation (Deng et al., 2023; Zhou et al., 2024), computer use (Xie et al., 2024), scientific discovery (Chen et al., 2024), and deep research (Phan et al., 2025; Wei et al., 2025), collectively advancing the field. In parallel, evaluation has moved beyond fixed-answer grading toward open-ended assessment (Li et al., 2024a; Wu et al., 2025; Du et al., 2025; Arora et al., 2025; **?**; **?**; **?**). DAComp is, to our knowledge, the first benchmark to cover the data-intelligence workflow, evaluating end-to-end data agents on both repository-level data engineering and open-ended data analysis, with the aim of advancing autonomous engineering and analytical capability.

**Benchmarks for data agents.** A data agent is an LLM-driven autonomous system that plans and executes end-to-end workflows, acquiring, transforming, and analyzing data via tool use and code execution to achieve user-defined objectives. Early work emphasizes single-shot tasks such as text-to-SQL (Yu et al., 2018; Li et al., 2024b) and code generation (Lai et al., 2023; Yin et al., 2023); more recent efforts push toward realistic SQL generation over real scenarios (Lei et al., 2024; Li et al., 2025; **?**), multi-turn data-science code generation (Hu et al., 2024; Huang et al., 2024; Jing et al., 2024) with iterative execution, and data analysis in business settings (Gu et al., 2024; Egg et al., 2025; Lai et al., 2025). DAComp goes beyond these efforts by introducing the first benchmark spanning enterprise data-intelligence workflows, encompassing repository-level engineering and open-ended analysis, and offering a rigorous testbed for advancing autonomous agents.

## 5 CONCLUSION

In this work, we presented **DAComp**, a comprehensive benchmark designed to evaluate data agents across the full data intelligence lifecycle. DAComp bridges the gap between isolated code generation and real-world enterprise demands by introducing two rigorous testbeds: **DAComp-DE** for repository-level pipeline orchestration and **DAComp-DA** for open-ended analytical reasoning. Our extensive experiments reveal a significant capability gap: even state-of-the-art models falter in holistic system maintenance and strategic insight synthesis, with success rates falling below 20% on engineering tasks. Furthermore, the inclusion of **DAComp-zh** paves the way for assessing agent robustness in multilingual environments, fostering the development of globally adaptable systems. By establishing this rigorous standard, DAComp aims to steer the community beyond mere technical accuracy, driving the evolution of truly autonomous and capable data agents for the enterprise.

## ACKNOWLEDGEMENTS

This work was supported by Beijing Natural Science Foundation (L243006) and the National Natural Science Foundation of China (No.62376270).

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

# A   EVALUATION METHODS DETAILS

## A.1   DACOMP-DE-IMPL/EVOL

The DAComp-DE-Impl/Evol evaluated using three execution-based metrics that progressively increase in strictness: Component Score (CS), Cascading Failure Score (CFS), and Success Rate (SR). Fig. 8 illustrates how these metrics differ in scoring a simple pipeline when an intermediate node fails.

**Component score (CS).**   Let $\mathcal{D}$ be the set of tasks. For task $d \in \mathcal{D}$, let layers be $\mathcal{L}$ (e.g., staging/intermediate/marts), and for each layer $\ell \in \mathcal{L}$ let $\mathcal{T}_{d,\ell}$ be its tables with weights $w_{d,t} \geq 0$. Define a table match indicator $m_{d,t} \in \{0,1\}$ by exact equivalence of *schema+data* between predicted and gold outputs (checked in DuckDB) under *perfect upstream inputs* (progressive/hybrid evaluation). The per-layer score and task-level CS are

$$S_{d,\ell} \;=\; \frac{\sum_{t \in \mathcal{T}_{d,\ell}} w_{d,t}\, m_{d,t}}{\sum_{t \in \mathcal{T}_{d,\ell}} w_{d,t}}, \qquad \mathrm{CS}_d \;=\; 100 \cdot \sum_{\ell \in \mathcal{L}} \alpha_\ell\, S_{d,\ell}, \quad \text{with } \alpha_\ell \geq 0,\ \sum_\ell \alpha_\ell = 1.$$

We report the benchmark CS as $\mathrm{CS} = \frac{1}{|\mathcal{D}|} \sum_{d \in \mathcal{D}} \mathrm{CS}_d$.

**Cascading failure score (CFS).**   For task $d$, let the pipeline DAG be $G_d = (V_d, E_d)$ with node weights $w_{d,j} \geq 0$ and ancestor set $\mathrm{Anc}_d(j)$. Let $m_{d,j} \in \{0,1\}$ be the node-level exact match (schema+data) under *predicted* upstreams. Define the cascading indicator recursively

$$s_{d,j}^{\mathrm{CFS}} \;=\; m_{d,j} \prod_{k \in \mathrm{Anc}_d(j)} s_{d,k}^{\mathrm{CFS}},$$

and the task-level CFS

$$\mathrm{CFS}_d \;=\; 100 \cdot \frac{\sum_{j \in V_d} w_{d,j}\, s_{d,j}^{\mathrm{CFS}}}{\sum_{j \in V_d} w_{d,j}}.$$

We report $\mathrm{CFS} = \frac{1}{|\mathcal{D}|} \sum_{d \in \mathcal{D}} \mathrm{CFS}_d$.

**Success rate (SR).**   A task is successful only if *every* component matches:

$$\mathrm{SR}_d \;=\; \prod_{j \in V_d} m_{d,j} \;\in\; \{0,1\}.$$

The benchmark success rate is the fraction of perfectly solved tasks:

$$\mathrm{SR} \;=\; \frac{1}{|\mathcal{D}|} \sum_{d \in \mathcal{D}} \mathrm{SR}_d.$$

In the evaluation process, we introduce the following tolerance measures to ensure the fairness and flexibility of the evaluation:

**Key Column Evaluation:**

*1)Evaluate only key columns.*   To focus the evaluation on the core components of the task, we evaluate only the key columns in the data (e.g., business-related columns, important computational columns). This ensures that the evaluation accuracy is concentrated on the most critical parts of the task.

*2)Exclude time columns.*   To avoid interference from time columns (e.g., small differences caused by different timestamps), we do not evaluate time columns.

**Tolerance for Numerical Columns:**

**Round to two decimal places.** When evaluating numerical columns, we allow a certain margin of error. Specifically, for numerical columns, all values are rounded to two decimal places to ensure consistency in data precision and avoid the influence of small fluctuations on the evaluation results.

However, for DE-Evol tasks, given the high strictness of the cascading metric, we adopt a threshold-based definition where a task is deemed successful if it maintains sufficient pipeline integrity (specifically, $\mathrm{CFS}_d \geq 80$).

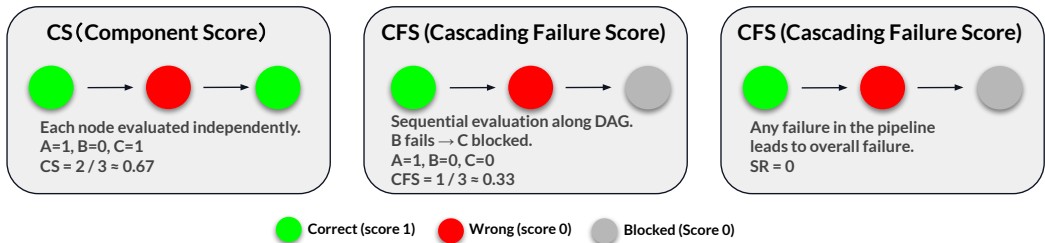

Figure 8: Illustration of how CS, CFS, and SR differ in scoring a simple pipeline when an intermediate node fails.

## A.2 DAComp-DE-Arch

**Three rubric dimensions.** The evaluation of the DE-Arch tasks is conducted across three key dimensions, which are defined as follows:

*1) Business Alignment and Semantic Accuracy:* This dimension assesses how well the solution aligns with business requirements and ensures semantic correctness. It evaluates whether the proposed solution comprehensively addresses the task's objectives while maintaining semantic integrity in the context of the recruitment cost analysis system.

*2) Technical Feasibility and Structural Completeness:* This dimension evaluates the technical feasibility of the solution and the completeness of its structure. It checks whether the proposed model can be implemented successfully given the available resources and dependencies, and whether it adheres to necessary technical standards and best practices.

*3) Design Quality:* This dimension evaluates the design and clarity of the model. It looks at how well the model is structured, the clarity of its naming conventions, and the organization of the components. It also considers the use of modular design principles to ensure that the solution is maintainable and scalable.

**DAComp-de-arch judge prompt.** This prompt standardizes how a model blueprint is evaluated against a given user question and rubric. It defines clear scoring logic (deterministic vs. path-based criteria), enforces an evidence-first policy (no evidence, no points), and constrains the final score to requirement-level sums. A canonical JSON output schema captures per-criterion analysis, evidence, and scores, enabling reproducible, auditable assessments across tasks.

---

**DE-Arch Judge Prompt**

```
## Task Description
You are a professional data architect. Evaluate a model blueprint using the provided user
question and scoring rubric. First, study the rubric, then assess the blueprint strictly
according to the rubric and determine the extent to which it meets the standards.

## Scoring Framework
Total Score is the sum of all requirement scores. Each requirement contains multiple
scoring criteria:
1. Deterministic criteria: can be scored directly without considering different
implementation paths.
2. Non-deterministic criteria: may have multiple implementation paths. Select the best
matching path based on the assistants response and score using the sub-criteria of that
path. If no path clearly matches, use your own expertise to judge whether the response
satisfies the requirement goal. If it does, assign points, but the score for this
requirement cannot exceed the maximum of the defined paths.

## Final Scoring Logic
Final Score = sum of all requirement scores.
Requirement Score = sum of its criteria scores.
Each criterion score is one of: direct score, best matching path score, unmatched path
score, or sum of sub-criteria.

## Evidence Policy
Provide explicit evidence for every scored item. If evidence is missing, assign zero. If
uncertain, do not guess; assign zero.
```

```
<User Question Start>
{user_query}
</User Question End>

<Model Blueprint Start>
{model_blueprint}
</Model Blueprint End>

<Scoring Rubric Start>
{rubric}
</Scoring Rubric End>

You must analyze and score each rubric item one by one.

Response format:
{
  "Requirement1": {
    "Criterion1.1": {
      "Analysis": "Carefully read the content of the model blueprint, determine whether it
meets Criterion1.1, and assign a score",
      "Criterion1.1.x.1": {
        "Analysis": "Carefully read the content of the model blueprint, determine whether
it meets Criterion1.1.x.1, and assign a score",
        "Evidence": [],
        "Score": 0
      },
      "Criterion1.1.x.2": {
        "Analysis": "Carefully read the content of the model blueprint, determine whether
it meets Criterion1.1.x.2, and assign a score",
        "Evidence": [],
        "Score": 0
      },
      "Score": 0
    },
    "Criterion1.2": {
      "Analysis": "Analyze the reason for the best matching path, determine the best
matching path: Path1.2.x",
      "Criterion1.2.x.1": {
        "Analysis": "Carefully read the content of the model blueprint, determine whether
it meets Criterion1.2.x.1, and assign a score",
        "Evidence": [],
        "Score": 0
      },
      "Criterion1.2.x.2": {
        "Analysis": "Carefully read the content of the model blueprint, determine whether
it meets Criterion1.2.x.2, and assign a score",
        "Evidence": [],
        "Score": 0
      },
      "Score": 0
    },
    "Total Score": 0
  },
  "Requirement2": {
    "Criterion2.1": {
      "Analysis": "Analyze the reason for the best matching path, determine that there is
no best matching path. Based on your own knowledge, determine whether it meets Criterion2
.1. Referencing other paths, it should meet Criterion2.1.notfound.1: xxx; Criterion2.1.
notfound.2: xxx",
      "Criterion2.1.x.1": {
        "Analysis": "Carefully read the content of the model blueprint, determine whether
it meets Criterion2.1.x.1, and assign a score",
        "Evidence": [],
        "Score": 0
      },
      "Criterion2.1.x.2": {
        "Analysis": "Carefully read the content of the model blueprint, determine whether
it meets Criterion2.1.x.2, and assign a score",
        "Evidence": [],
        "Score": 0
      },
      "Score": 0
    }
  },
  "Total Score": 0
}
```

## A.3 DACOMP-DA

### A.3.1 HIERARCHICAL RUBRIC

**Six rubric dimensions.** The evaluation of DA tasks is conducted across six key dimensions, which are defined as follows:

*1) Completeness*: This dimension assesses whether the agent's response comprehensively addresses all explicit and implicit requirements of the prompt. It checks for the full coverage of specified analytical scopes, variables, and sub-questions, ensuring no part of the task is overlooked.
*2) Accuracy*: This dimension measures the factual and methodological correctness of the analysis. It includes the validity of the code logic, the correctness of calculations, and the factual precision of all reported figures and statistical results against a verifiable ground truth.
*3) Insightfulness*: This dimension evaluates the agent's ability to move beyond mere data reporting to generate valuable interpretations. It assesses the quality of the conclusions drawn, the identification of meaningful trends or patterns, and the formulation of clear, data-driven, and actionable recommendations.
*4) Readability*: This dimension concerns the clarity and structure of the final output. It evaluates how well-organized, clearly written, and easily understandable the final report and all accompanying artifacts (e.g., code, tables, visualizations) are for a human audience.
*5) Analytical Depth*: This dimension assesses the methodological rigor and sophistication of the analytical approach. It distinguishes between superficial analyses (e.g., simple averages) and more profound approaches that involve appropriate statistical tests, control for variables, and demonstrate a deeper understanding of the underlying data and business context.
*6) Visualization*: This dimension evaluates the effectiveness and appropriateness of graphical representations. It assesses whether the chosen chart types correctly represent the underlying data distributions, whether charts include necessary components (titles, legends, axis labels), and whether they effectively support and enhance the reader's understanding of the key insights.

**Hierarchical rubrics example.** As shown in Tab. 11, we provide a hierarchical scoring rubric that decomposes the task into requirements and sub-standards, with explicit checkpoints and point allocations for consistent evaluation.

**Hierarchical rubric prompt.** Below is the Hierarchical Rubric Prompt.

```
Hierarchical Rubric Prompt

# Task Description
You are a data analysis expert. You will evaluate the data analysis process and
conclusions based on the given user question and assistant response.
Your task is to read a data-analysis-oriented response and a scoring rubric, and then,
according to that rubric, determine how well the assistant's response meets the criterion.

# Assistant Response
<<assistant_response>>

# Scoring Criterion
<<rubric>>

# Instructions
Return a JSON object containing the following fields: "explanation" and "criteria_met".
- "explanation" should be a string explaining why the response does or does not satisfy
the criterion.
- "criteria_met" should be a boolean (true/false) indicating whether the response
satisfies the criterion. If a criterion contains multiple sentences or sub-criteria, all
must be considered. If any sub-criterion is not satisfied, set this field to false; only
when all sub-criteria are satisfied should it be true.

# Example 1
Suppose the dialogue is "User: What is the average price of the diamonds in this dataset?
Assistant: The average price of the diamonds is $5000.", and after calculation, the
correct average is about $3932. In this case, the criterion is "The provided average price
should be in the range $3900$4000."

```json
{
```

Table 11: Hierarchical rubric for the business analysis task defined as follows: *Compare the business performance across the four major regions (Central, East, South, West), analyze the differences in penetration rate and profitability of each region in the three market segments (Consumer, Corporate, Home Office) during 2015, 2016, and 2017, identify the region-market combination with the best performance, and provide recommendations for expansion.*

| Requirement & Standard | | Path | Item (Sub-standard) & Key Description | Points |
|---|---|---|---|---|
| **Req. 1:** Penetration & Profitability Analysis (Max 8 pts) | **Std. 1.1:** Penetration Rate Analysis | **1.1.A** (Sales) | **1.1.A.1 (Completeness):** Define & calculate sales penetration (annual + 3-yr avg). | 1 |
| | | | **1.1.A.2 (Accuracy):** Calculations must match anchors (e.g., West-Consumer avg $\approx 29.72\%$). | 2 |
| | | | **1.1.A.3 (Conclusion):** Derive $\geq 3$ valid conclusions on market position (e.g., East-/West duopoly). | 1 |
| | | **1.2.B** (Risk-Adj. Margin) | **1.2.B.1 (Completeness):** Define & calculate risk-adjusted profit margin (e.g., mean $- 0.5 \times$ std). | 1 |
| | | | **1.2.B.2 (Accuracy):** Calculations must match anchors (e.g., Central-Home Office adj $\approx 16.37$). | 2 |
| | | | **1.2.B.3 (Conclusion):** Derive $\geq 2$ insights on risk/return (e.g., identify stable vs. high-risk yields). | 1 |
| | **Std. 1.2:** Profitability Analysis | **1.2.A** (Basic Margin) | **1.2.A.1 (Completeness):** Define & calculate basic profit margin (annual + 3-yr avg). | 1 |
| | | | **1.2.A.2 (Accuracy):** Calculations must match anchors (e.g., Central-Corporate $\approx$ 20.22%). | 1 |
| | | | **1.2.A.3 (Conclusion):** Derive $\geq 2$ conclusions on profit tiers and strategic priorities. | 1 |
| | | **1.1.B** (Orders) | **1.1.B.1 (Completeness):** Define & calculate order penetration (annual + 3-yr avg). | 1 |
| | | | **1.1.B.2 (Accuracy):** Cross-validate sales vs. order trends; calculations must be correct. | 1 |
| | | | **1.1.B.3 (Conclusion):** Analyze avg. order value to derive insights on customer structure. | 1 |
| **Req. 2:** Regional Perf. Comparison (Max 3 pts) | **Std. 2.1:** Multi-dim. Evaluation | **2.1.A** (Weighted Score) | **2.1.A.1 (Completeness):** Define & compute a weighted composite score from normalized penetration | profit. |
| | | | **2.1.A.2 (Accuracy):** Final rankings are consistent with the chosen weights and normalized values. | 1 |
| | | | **2.1.A.3 (Conclusion):** Derive regional roles (Leaders, Potentials, etc.) based on composite scores. | 1 |
| **Req. 3:** Identify Best Combo (Max 2 pts) | **Std. 3.1:** Optimal ID | **3.1.A** (Composite Rank) | **3.1.A.1 (Accuracy):** Identify TOP3 combinations using a weighted score; must match $\geq 2$ anchors (e.g., East-Home Office pen. $\approx 35.00\%$, margin $\approx 18.06\%$). | 1 |
| | | | **3.1.A.2 (Conclusion):** Analyze TOP3 for strategic value (core vs. growth) and inherent risks. | 1 |
| **Req. 4:** Expansion Strategy (Max 2 pts) | **Std. 4.1:** Strategic Recs. | **4.1.A** (Action Plan) | **4.1.A.1 (Conclusion):** Provide a comprehensive plan covering:
• Business positioning & priorities
• Actionable steps with KPIs
• Strategic rationale & risk control
• Implementation timeline | 2 |

```
  "explanation": "The assistant provided $5000, which is not within the required
$3900$4000 range, so it does not meet the criterion.",
  "criteria_met": false
}
```

# Example 2
Please note that sometimes the rubric may describe an undesirable condition with a
negative score. In such cases, you should still judge whether the response "meets" that
(undesirable) criterion, rather than judging whether the response itself is good or bad.

Case A (High-quality response):
Dialogue:
"User: Compute the energy consumption share of each department. Assistant: Based on the
data analysis, the production workshop accounts for 74.99% of total energy consumption,
the R&D center 12.51%, warehousing and logistics 11.57%, and the office area 0.92%. The
calculation process is complete and data sources are clear."
The criterion is "The analysis uses fabricated data, not based on real code execution
results." Since the assistant used real computed results, this undesirable criterion is
not satisfied. Return:

```json
{
  "explanation": "The assistant's data matches actual computation and clearly states the
process and data sources, with no fabricated data. Therefore, the (undesirable) criterion
is not satisfied.",
  "criteria_met": false
}
```

Case B (Low-quality response):
Dialogue:
"User: Analyze the relationship between the share of industrial water use and economic
development. Assistant: Generally, regions with a higher share of industrial water have
better economic development, with about 60% correlation. I think this trend is reasonable."
The criterion is "The analytical conclusion lacks a specific correlation coefficient
calculation and uses subjective speculation rather than data-driven conclusions." Since
the assistant did not provide a concrete calculation and used subjective judgment, this
undesirable criterion is satisfied. Return:

```json
{
  "explanation": "Statements such as 'about 60% correlation' and 'I think this trend is
reasonable' are subjective, without a specific correlation computation or data support.
Therefore, the (undesirable) criterion is satisfied.",
  "criteria_met": true
}
```

In other words, for negative-scoring criteria: a high-quality response should be judged as
false (because it does not meet the undesirable criterion), and only low-quality responses
that actually meet the undesirable criterion should be judged as true.

# Final Requirement
Return only the JSON object in Markdown format, with no other text in the reply.
```

### A.3.2 GOOD-SAME-BAD JUDGE

**Good-Same-Bad Judge Prompt**

```
You are a data analysis evaluation expert. You need to judge whether the following two
reports are good or bad.
Evaluate them in detail from the following two dimensions:
1. The report is highly readable and easy to understand.
2. The analysis is professional and in-depth.

Give a score for each dimension, with a range of -10 to 10.
Notes:
+ The analysis and scoring are comparative: compare the report to be evaluated with the
baseline report.
+ -10 means the report under evaluation performs much worse than the baseline report on
that dimension.
+ 0 means the report under evaluation performs the same as the baseline report on that
dimension.
```

```
+ 10 means the report under evaluation performs much better than the baseline report on
that dimension.
+ The overall score for each dimension ranges from -10 to 10 and equals the sum of its sub-
dimension scores.

Details:
Readability is specifically reflected in the following sub-dimensions:
- Convey complex information concisely so readers can quickly grasp key points (e.g., use
Markdown to structure the report; use bold/italic to highlight key information). Score
range: -4 to 4.
- Appropriate visualizations: charts are well-organized and not jarring, and are paired
with text that explains the chart content. Score range: -3 to 3.
- Follows a clear writing structure, such as a "general--specific--general" flow, with
clear hierarchy (e.g., use subheadings). Score range: -2 to 2.
- Concise language: avoid verbosity and repeated expressions. Score range: -1 to 1.

Professionalism and depth of analysis are reflected in the following sub-dimensions:
- Analyze from multiple dimensions and perspectives, considering different factors and
scenarios. Score range: -4 to 4.
- Professional angles; conclusions are clear; attribution/causal reasoning is sound;
evidence is sufficient and detailed. Score range: -3 to 3.
- Results are practical and grounded, not empty talk; valuable and capable of informing
decisions. Score range: -2 to 2.
- Estimate the potential impact of recommendations. Score range: -1 to 1.

Output format:
```json
{
    "Readability": {
        "Analysis": "On sub-dimension xxx, the baseline report's strengths/weaknesses are
xxx, and the report under evaluation's strengths/weaknesses are xxx. Contrastive analysis
of the differences; the report under evaluation scores xx on this sub-dimension.",
        "Summary": "Summary of the readability analysis for the report under evaluation",
        "Score": int
    },
    "Analytical Depth": {
        "Analysis": "On sub-dimension xxx, the baseline report's strengths/weaknesses are
xxx, and the report under evaluation's strengths/weaknesses are xxx. Contrastive analysis
of the differences; the report under evaluation scores xx on this sub-dimension.",
        "Summary": "Summary of the professionalism and depth analysis for the report under
evaluation",
        "Score": int
    }
}
```
```

# B EXPERIMENTS SETTING

## B.1 AGENT BASELINE

For our data engineering baseline, we develop an agent framework inspired by the **ReAct** (Yao et al., 2022). This framework enables the agent to perform complex, repository-level tasks through multi-turn interactions within a sandboxed, interactive file system environment.

To facilitate these interactions, we define a concise yet powerful set of four actions, as detailed in Tab. 12. The agent iteratively generates a thought process, selects an action, and observes the outcome from the file system, continuing this loop until the task is complete. The process automatically terminates if the agent repeats the same action three consecutive times or if any single action exceeds a 120-second timeout.

For more complex tasks, such as DE-Impl and DE-Evol, we extend the framework to a multi-agent approach. In this setup, each agent is assigned a specific SQL task, represented by a YAML specification. Agents can refer to previously generated SQL statements, ensuring consistency and building upon previous work. A dependency graph is established based on SQL relationships, with each agent operating in the prescribed order according to this graph. Upon completion of each SQL task, the agent is prompted to validate its output using a testing script, facilitating error correction and refinement. The framework also includes a validation agent, responsible for ensuring that the entire data pipeline runs smoothly. To optimize performance, each agent is constrained to a maximum of 50 steps, while the validation agent is allowed up to 100 steps.

For DE-Evol tasks, we implement a dual-agent approach. One agent is dedicated to validation, ensuring the correctness of modifications, while the other agent handles implementation, including the modification or addition of SQL statements. After each update, the agent executes a testing script to refine the SQL, ensuring alignment with the evolving task requirements. Each agent is limited to a maximum of 100 steps.

Table 12: The core action space for our DE agent baseline. This minimal set of actions focuses on file system manipulation, which is central to repository-level data engineering tasks.

| Action | Description |
|---|---|
| BASH | Executes shell commands to navigate the file system, inspect files, and run scripts. |
| CREATE_FILE | Creates a new file with specified content. |
| EDIT_FILE | Edits or overwrites the content of an existing file. |
| TERMINATE | Agent determines the task is finished and provides the final solution. |

## B.2 OPENHANDS DETAILS

We have integrated OpenHands(Wang et al., 2024) into our DE and DA tasks, utilizing the Codeact agent. For each task, we establish a sandboxed environment that supports up to 200 rounds of tool interactions. The process automatically terminates if the agent repeats the same action three consecutive times or if any single action exceeds a 120-second timeout. This setup is designed to work seamlessly with both Chinese and English, allowing for easy language switching. Three sets of tools are provided, as detailed in Tab. 13.

Table 13: The Core Action Space for OpenHands. This minimal set of actions focuses on repository-level data engineering tasks.

| Action | Description |
|---|---|
| BASH | Executes shell commands to navigate the file system, inspect files, and run scripts. |
| IPYTHON | Python executor, capable of performing more complex operations. |
| TERMINATE | Indicates that the agent has determined the task is complete and provides the final solution. |

## B.3 ADDITIONAL EXPERIMENTAL RESULTS

**Task complexity and scale are key determinants of performance.** The overall complexity of a data engineering task, measured by the number of nodes in the dependency graph or the total lines of code, strongly impacts agent performance, as shown in Fig. 9. For *Implementation* tasks, we observe a general decline in the Component Score as the number of nodes increases, with models like GPT-5 showing a significant performance drop on tasks with more than 50 nodes. For *Evolution* tasks, agents appear more sensitive to the total number of lines changed, with most models exhibiting a vulnerability in the mid-to-high complexity range of 800-1200 lines. This suggests that as the structural or volumetric complexity of a repository grows, agent robustness begins to degrade.

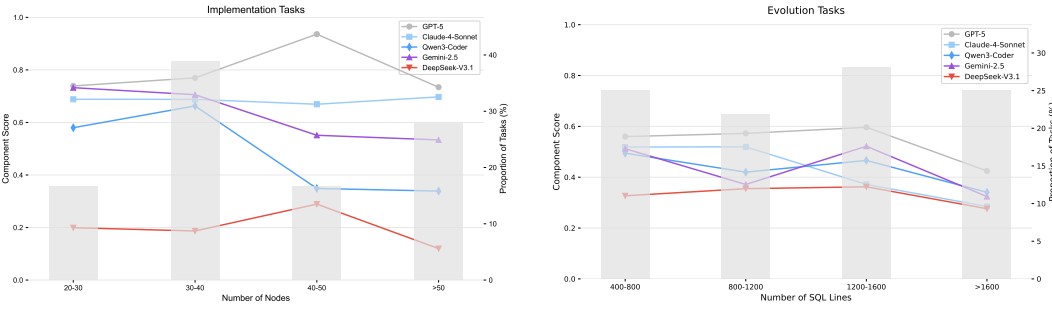

Figure 9: Effect of node count and line count.

Table 14: Rubric validity analysis. **Panel A**: Reliability on open-ended (Tier 3) tasks. **Panel B**: Defense against fluent hallucinations.

| A. Reliability on Open-Ended (Tier 3) Tasks | | |
|---|---|---|
| **Metric** | **Full Dataset** | **Tier-3 Only** |
| Case-Level (ICC) | 89.0% | 85.2% |
| Ranking ($\tau_b$) | 100.0% | 98.5% |
| **B. Adversarial "Honeypot" Test (Score / 10)** | | |
| **Sample Type** | **Readability** | **Accuracy** |
| Fluent-and-Correct | 9.5 | 9.2 |
| Fluent-but-Wrong | 9.4 | **1.1** |

**Rubric validity.** To validate our framework's resilience against novel solutions and "fluent hallucinations," we conducted a three-part analysis. First, regarding **scope**, we found that 12.1% of validation cases required principle-based (Tier 3) scoring, confirming that unenumerated solutions are a non-trivial component of the benchmark. Second, to verify **reliability**, we isolated this Tier-3 subset. As shown in Tab. 14 (Panel A), the judge maintains exceptional alignment with humans (ICC 85.2%), with negligible degradation compared to the full dataset. Finally, to test **defense**, we conducted a "honeypot" adversarial attack using fluent but logically flawed samples. Tab. 14 (Panel B) shows the judge correctly penalized the *Accuracy* of these samples (1.1) despite their high *Readability* (9.4). This confirms that our dimensional separation strategy successfully forces the judge to prioritize methodological substance over superficial fluency.

### B.4 HUMAN-LLM AGREEMENT EXPERIMENTS METRICS

We assess agreement between human annotators and LLM judges at three granularities—item, case, and model—each aligned with the statistical nature of DAComp's scoring signals.

**Item-level (Krippendorff's $\alpha$ / Weighted $\kappa$).** Rubric items are ordinal with heterogeneous weights, while GSB labels are categorical (*Good/Same/Bad*). For rubric items we compute Krippendorff's $\alpha$:

$$\alpha = 1 - \frac{D_o}{D_e},$$

where $D_o$ and $D_e$ denote observed and expected disagreement. For GSB, we use weighted Cohen's $\kappa$:

$$\kappa_w = 1 - \frac{\sum_{i,j} w_{ij} O_{ij}}{\sum_{i,j} w_{ij} E_{ij}},$$

with $O_{ij}$ the observed contingency table, $E_{ij}$ its chance expectation, and $w_{ij}$ quadratic penalties. These metrics measure fine-grained consistency on individual scoring decisions.

**Case-level (ICC(A,1)).** Each DA task yields a **numerical aggregated score** derived from the items:

$$S^{\text{rubric}} = \frac{\sum_{k=1}^{N} s_k}{\sum_{k=1}^{N} w_k}, \qquad S^{\text{gsb}} = \frac{\max(0, |G| - |B|)}{|G| + |S| + |B|}.$$

We quantify task-level agreement using the **two-way single-measure intraclass correlation coefficient for absolute agreement**, denoted as ICC(A,1):

$$\text{ICC}(A, 1) = \frac{MS_R - MS_E}{MS_R + (k-1)MS_E},$$

where $MS_R$ and $MS_E$ are the between-target and residual mean squares, and $k$ is the number of raters. Unlike simple correlation coefficients (e.g., Pearson), which only measure linear association, this ICC formulation strictly captures the absolute alignment of scores between the human and the LLM.

**Model-level (Kendall's $\tau_b$).** To evaluate ranking consistency of full-model performance, we compute Kendall's $\tau_b$ between human and LLM leaderboards:

$$\tau_b = \frac{n_c - n_d}{\sqrt{(n_c + n_d + t_x)(n_c + n_d + t_y)}},$$

where $n_c, n_d$ count concordant and discordant pairs and $t_x, t_y$ correct for ties. Because benchmark outcomes are ordinal and include ties, $\tau_b$ offers a robust measure of ranking reliability.

**Interpretation.** Item-level metrics validate micro-level decision consistency; ICC(A,1) assesses task-level score reliability; and Kendall's $\tau_b$ ensures that the LLM judge preserves global model rankings. Together, these metrics provide a principled and comprehensive validation of the LLM-as-judge framework.

## C EXAMPLES

### C.1 DE-ARCHITECTURE TASK

This task aims to derive a data engineering blueprint for a business question. As an illustration, we present a Salesforce-related question along with its evaluation rubrics.

---

**DE-Architecture: Business Requirement**

```
Can we build a "true performance profile" for each sales representative? I want to
understand not just their sales volume, but more importantly, the quality of the customers
they acquire. Will these customers continue to do business with us? And do details of the
sales process (like the pace of opportunity advancement, customer communication frequency,
etc.) affect the long-term value of the customer?
```

---

**DE-Architecture: Evaluation Rubric**

```
Requirement I: Business Alignment & Semantic Accuracy
- Ensures that the data models correctly reflect the core business logic.
- Customer Metrics:
  * Customer quality, LTV, and repeat business metrics must be correctly attributed.
  * Metrics must fall within logically valid ranges (e.g., 0--100).
- Sales Process Metrics:
  * Sales cycle and communication quality scores must be implemented and populated.
  * Metrics must demonstrate realistic values.

Requirement II: Technical & Structural Integrity
- Validates the technical soundness and completeness of the data tables.
- Model Completeness:
  * The final mart table (...performance_profile) must be fully populated for all valid
profiles.
  * No nulls allowed in key identifier fields.
- Data Consistency:
  * Records for each sales representative must be consistent across all related
intermediate and mart tables.
- Sufficient Volume:
  * The pipeline must produce at least 200 valid profiles to ensure analytical robustness.

Requirement III: Analytical Value & Logic
- Verifies that the final outputs provide meaningful insights and adhere to business
hypotheses.
- Value Profile Classification:
  * The "Tree Planter" classification for high-value reps must be applied to all eligible
candidates.
  * Must identify a sufficient cohort (e.g., >= 150).
- Business Logic Validation:
  * The final model must satisfy key business hypotheses.
  * Example: a positive correlation between customer quality scores and repeat business
rates.
```

## C.2 DE-IMPLEMENTATION TASK

This task evaluates an agent's ability to build an entire data engineering repository from scratch based on a detailed technical specification.

---

**DE-Implementation: DE Design Specifications**

```
staging_layer:
  example: stg_salesforce__account
  purpose: >
    Transform raw Salesforce account records into clean staging tables.
    Apply heavy-duty data cleaning:
    - normalize_email(), format_phone()
    - enforce DECIMAL(15,2) precision on revenue
    - quarantine() invalid records, nullify_field() for soft failures
    Guarantee: no null in account_id, owner_id; business fields standardized.
  ... ...
intermediate_layer:
  example: int_salesforce__account_enhanced
  purpose: >
    Construct enriched account model with business logic.
    Join staging tables with user dimension     add owner + hierarchy info.
    Add derived fields (activity_score, account_health).
    Grain = "1 row per account".
    Note: Designed as reusable building block for multiple marts.
  ... ...
marts_layer:
  example: fct_salesforce__sales_pipeline
  purpose: >
    Deliver pipeline fact table for exec-level analytics & forecasting.
    Row grain = "1 opportunity per reporting_date".
    Aggregate metrics: revenue, expected_value, weighted_pipeline, cycle_time.
    Attach dimensions: region, industry, owner, fiscal_calendar.
    Feeds dashboards, KPIs, and predictive modeling.
```

---

**DE-Implementation: Ground-truth DE project repository**

```
Staging Layer:
    stg_salesforce__account_history.sql, stg_salesforce__account.sql,
    stg_salesforce__contact_history.sql, stg_salesforce__contact.sql,
    stg_salesforce__event.sql, stg_salesforce__lead.sql,
    stg_salesforce__opportunity_history.sql, stg_salesforce__opportunity_line_item.sql,
    stg_salesforce__opportunity.sql, stg_salesforce__order.sql,
    stg_salesforce__product_2.sql, stg_salesforce__task.sql,
    stg_salesforce__user_role.sql, stg_salesforce__user.sql

Intermediate Layer:
    int_salesforce__account_enhanced.sql, int_salesforce__activity_summary.sql,
    int_salesforce__date_spine.sql, int_salesforce__lead_conversion_funnel.sql,
    int_salesforce__opportunity_aggregation_by_owner.sql,
    int_salesforce__opportunity_pipeline.sql, int_salesforce__user_performance.sql

Mart Layer:
    dim_salesforce__user.sql, fct_salesforce__account_engagement.sql,
    fct_salesforce__lead_performance.sql, fct_salesforce__sales_pipeline.sql,
    salesforce__account_daily_history.sql, salesforce__contact_daily_history.sql,
    salesforce__contact_enhanced.sql, salesforce__daily_activity.sql,
    salesforce__manager_performance.sql, salesforce__opportunity_daily_history.sql,
    salesforce__opportunity_enhanced.sql, salesforce__opportunity_line_item_enhanced.sql,
    salesforce__owner_performance.sql, salesforce__revenue_analytics.sql,
    salesforce__sales_snapshot.sql, salesforce__team_performance.sql
```

---

## C.3 DE-EVOLUTION TASK

This task evaluates an agent's ability to plan, surface complete requirements, and produce SQL by adapting an existing SQL repository to a revised business specification—identifying scope and metric changes, updating definitions and dependencies, and delivering a final, fit-for-purpose project that fully aligns with the new requirement.

---

**DE-Evolution: Requirement Specifications**

```
Business Pain Point:
  - Current opportunity management lacks robust cost-effectiveness analysis.
  - Cannot measure acquisition cost, maintenance, and ROI consistently.

Objectives:
  - Multi-dimensional cost allocation (travel, marketing, labor, shared resources).
  - Lifecycle c o s t revenue matching (one-time, subscription, multi-year).
  - Multi-scenario ROI analysis with sensitivity & scenario modeling.

Implementation Highlights:
  - Flexible allocation rules (time weighting, channel path, dynamic labor rates).
  - ROI logic per revenue model (rolling 12M, discounted LTV, IRR).
  - Time-based alignment of costs and revenues.
  - Data quality checks (missing value fill, anomaly detection).
```

---

**DE-Evolution: Ground-truth solution**

```
Modified SQL:
  - int__opportunity_pipeline.sql
  - fct__sales_pipeline.sql
  - revenue_analytics.sql
  - fct__account_engagement.sql

Key Enhancements in fct__sales_pipeline:
  - Added cost allocation fields (acquisition, travel, marketing, labor).
  - Added ROI metrics (roi_percentage, cost_per_dollar_revenue, LTV ratio).
  - Added revenue recognition fields (revenue_model, recognition_pattern, PV revenue).
  - Added cost variance & risk indicators (variance %, anomaly flag, risk level).
  - Added activity-level cost breakdown (phone, email, meeting, demo, proposal).
  - Added efficiency & ranking metrics (cost_efficiency_tier, investment_priority_rank).
```

## C.4 DA TASK

In this section, we show the detailed classification of the task types solved by DAComp-DA in Tab. 15.

# D ERROR ANALYSIS

## D.1 DE-ARCHITECTURE ERROR ANALYSIS

**Error distribution in de-arch tasks.** The error analysis for the DE-Arch tasks reveals several architectural shortcomings across the evaluated models, as shown in Table 16. The models exhibit varying levels of function point omission, dependency errors, missing entity models, naming inconsistencies, and improper model layering. Models such as `Qwen3-8B` and `Qwen3-235B-A22B` demonstrate higher error rates across multiple dimensions, indicating more significant architectural flaws. In contrast, `GPT-5` and `Gemini-2.5-Pro` perform relatively better, with fewer errors in dependency management and model structure, though they still show room for improvement, particularly in entity model completeness and naming consistency.

**DE-Architecture error case.** As shown in Fig. 10, we present a "DE-Arch Error Case" panel: the left side shows a minimal blueprint, while the right side scores 16 checklist items (final score: 5/16), revealing several systemic weaknesses.

## D.2 DE-IMPLEMENTATION ERROR ANALYSIS

**Divergence in schema fidelity.** The analysis of schema-level constraints—specifically Data Type and Missing Column errors—reveals distinct behavioral patterns in Table 17. Data Type errors remain consistently marginal (ranging from 2% to 7%) across all evaluated models, suggesting a universal proficiency in handling fundamental SQL type systems. In stark contrast, Missing Column errors serve as a sharp discriminator of model capability: while SOTA models like GPT-5 achieve

Table 15: Definitions and Examples for the Five DA Task Type Categories.

| Category Name | Definition & Objective | Example Question |
|---|---|---|
| **Descriptive** | Focuses on summarizing historical data to answer "What happened?". Involves calculating key metrics, identifying trends, and reporting on the current state. | *Analyze sales trends in the three categories of office supplies, technology, and furniture from 2015 to 2018, identify the fastest-growing product category for each year, and evaluate performance differences among regional managers based on regional sales data.* |
| **Diagnostic** | Aims to uncover the root causes of a particular outcome, answering "Why did it happen?". Involves drilling down into data, identifying anomalies, and discovering factors that influence a result. | *For the product category with the greatest annual volatility, investigate the underlying reasons. Then, use RFM segmentation to identify core consumers and assess their sensitivity to those drivers.* |
| **Strategic** | Focuses on providing data-driven recommendations for future actions, answering "What should we do?". It translates insights from descriptive and diagnostic analysis into concrete, actionable plans. | *As the sales leader for Coca-Cola, which sales outlet types should I increase or decrease our contracts with? Please provide recommendations based on an analysis of key data such as sales target attainment, customer complaints, and sales volume.* |
| **Pattern Recognition** | Involves exploring data to uncover previously unknown relationships, correlations, or patterns, answering "What are the hidden connections?". It is often open-ended and seeks to generate new hypotheses. | *Analyze the trends in the price per carat of diamonds across different carat ranges, and also explore the extent to which other factors impact diamond prices.* |
| **Profiling** | Aims to group a population (e.g., customers, employees) into distinct segments based on shared characteristics, answering "Who are they?". The goal is to understand the composition and behavior of different groups. | *Based on a comprehensive ranking that considers effective work hours, overall production quantity, and quality, please analyze the characteristics of our top performers and recommend the ideal profile for future hires.* |

Table 16: Detailed analysis for DE-Arch tasks.

| Model | Function Point Omission | Dependency Errors | Missing Entity Models | Naming Inconsistencies | Improper Model Layering |
|---|---|---|---|---|---|
| GPT-5 | 26.51 | 17.14 | 18.91 | 6.41 | 7.21 |
| Gemini-2.5-Pro | 27.22 | 18.33 | 20.64 | 8.53 | 9.16 |
| Qwen3-Coder | 30.56 | 22.26 | 24.33 | 11.19 | 12.14 |
| DeepSeek-V3.1 | 31.43 | 23.18 | 25.25 | 12.52 | 13.00 |
| Qwen3-235B-A22B | 35.38 | 36.59 | 27.81 | 11.42 | 13.82 |
| Qwen3-8B | 44.00 | 35.23 | 36.01 | 13.73 | 14.35 |

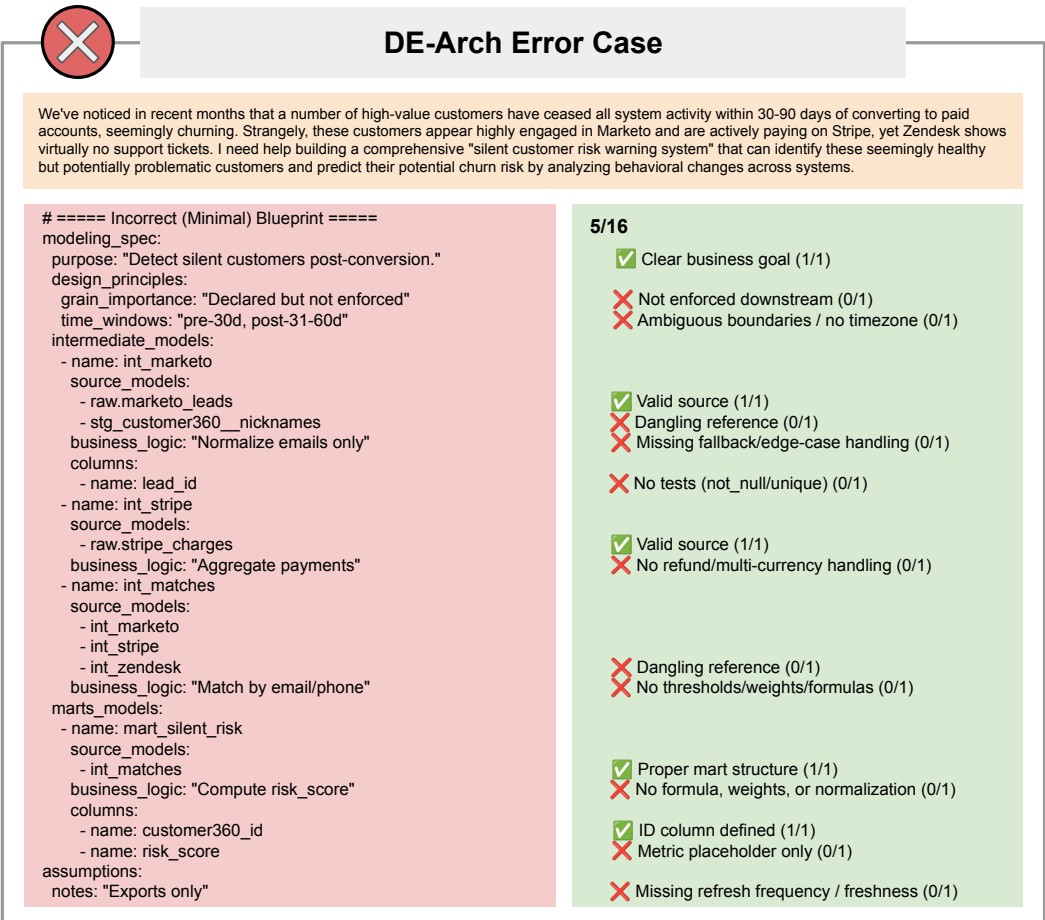

Figure 10: DE-Arch error case. Key issues include: (1) a clear business objective but weak downstream enforcement, ambiguous boundaries, and no timezone convention; (2) dangling references in intermediate models, missing fallback and edge-case handling, and absence of basic tests such as not_null/unique; (3) no treatment for refunds and multi-currency scenarios; (4) missing thresholds, weights, and formulas in aggregation and metric layers, with some fields not provided by sources; (5) placeholder metrics only and no refresh-frequency/freshness policy. Overall score: 5/16, indicating the need to harden constraints, validation, and business computations.

Table 17: Detailed error analysis for various models on DE-Impl and DE-Evol tasks.

| Model | Data Type Errors | Missing Column Errors | SQL Omission | Calculation Logic Errors | Dependency Errors |
|---|---|---|---|---|---|
| *DE-Impl Tasks* | | | | | |
| GPT-5 | 2.22 | 0.29 | 5.18 | 40.65 | 66.01 |
| Gemini-2.5-Pro | 4.25 | 5.74 | 15.17 | 37.58 | 67.31 |
| Qwen3-Coder | 5.54 | 1.38 | 22.88 | 36.91 | 66.13 |
| DeepSeek-V3.1 | 6.88 | 3.00 | 28.58 | 37.45 | 65.68 |
| Qwen3-235B-A22B | 5.74 | 34.73 | 89.79 | 42.06 | 73.94 |
| Qwen3-8B | 4.57 | 16.82 | 95.74 | 34.62 | 70.55 |
| *DE-Evol Tasks* | | | | | |
| GPT-5 | 2.09 | 10.05 | 11.69 | 28.93 | 56.45 |
| Gemini-2.5-Pro | 4.18 | 27.35 | 16.94 | 40.56 | 64.98 |
| Qwen3-Coder | 3.32 | 23.88 | 19.09 | 35.66 | 63.29 |
| DeepSeek-V3.1 | 2.46 | 29.53 | 34.00 | 31.87 | 59.23 |
| Qwen3-235B-A22B | 1.00 | 54.36 | 65.79 | 23.62 | 53.86 |
| Qwen3-8B | 0.85 | 44.06 | 58.17 | 27.88 | 53.44 |

near-perfect coverage (0.29% error rate), base models exhibit significant deficiencies (up to 34.73%). This disparity indicates that while correct type inference is readily accessible, ensuring exhaustive field retention requires a higher order of instruction-following precision found primarily in top-tier models.

**Dominance of dependency errors.** The comprehensive error analysis presented in Table 17 identifies Dependency Errors as the primary bottleneck constraining model performance in DE-Impl tasks. Regardless of model capacity—from SOTA models like GPT-5 to smaller counterparts—dependency error rates consistently exceed 65%. Further decomposition in Table 20 reveals a balanced distribution between "Missing dependencies" and "Extra dependencies." This equilibrium suggests that current LLMs struggle to construct accurate global data lineage graphs, lacking the precise contextual awareness required to manage long-range dependencies effectively within complex data engineering frameworks.

**Complexity sensitivity in sql omission.** Analysis of SQL Omission exhibits significant stratification based on model capability and architectural depth. As illustrated in Table 18, omission rates escalate as the data architecture evolves from the foundational Staging layer to the highly aggregated Marts layer, reflecting the penalty imposed by increasing business logic complexity. While weaker models (e.g., Qwen3-8B) suffer catastrophic failure in the Marts layer with omission rates approaching 100%, advanced models demonstrate superior robustness, maintaining omission rates below 10%, thereby highlighting a distinct capability gap in handling complex, multi-level data transformations.

**Cascading effects in calculation logic.** A granular examination of Calculation Logic Errors uncovers a significant "error cascading effect" within data pipelines. As shown in Table 19, for high-performing models (e.g., GPT-5 and Gemini-2.5-Pro), the predominant source of calculation errors is not faulty reasoning at the current node (Intrinsic Errors), but rather the propagation of inaccuracies from preceding layers (Upstream Errors). For instance, GPT-5's upstream errors are approximately three times more prevalent than its intrinsic errors across all layers. This insight implies that optimizing DE-Impl performance requires shifting focus from merely improving single-node code generation to enhancing the model's capacity for fault tolerance and consistency maintenance across the entire lineage.

**DE-Implementation error case.** It is crucial to prevent implementation issues—such as improper joins, flawed aggregations, and circular dependencies. The cases in Fig. 11 and Fig. 12 serve as representative DE-Impl examples.

## D.3 DE-EVOLUTION ERROR ANALYSIS

**Contextual awareness in dependency management.** The comparative analysis of dependency errors illustrates that DE-Evol and DE-Impl tasks present fundamentally different challenges. While the overall error rate in evolution scenarios is lower than in construction tasks, the nature of failures shifts significantly. As indicated in Table 20, weaker models in DE-Evol exhibit a pronounced bias towards "Missing Dependencies," contrasting with the more balanced error distribution observed in DE-Impl. This suggests that preserving the integrity of an existing pipeline imposes specific

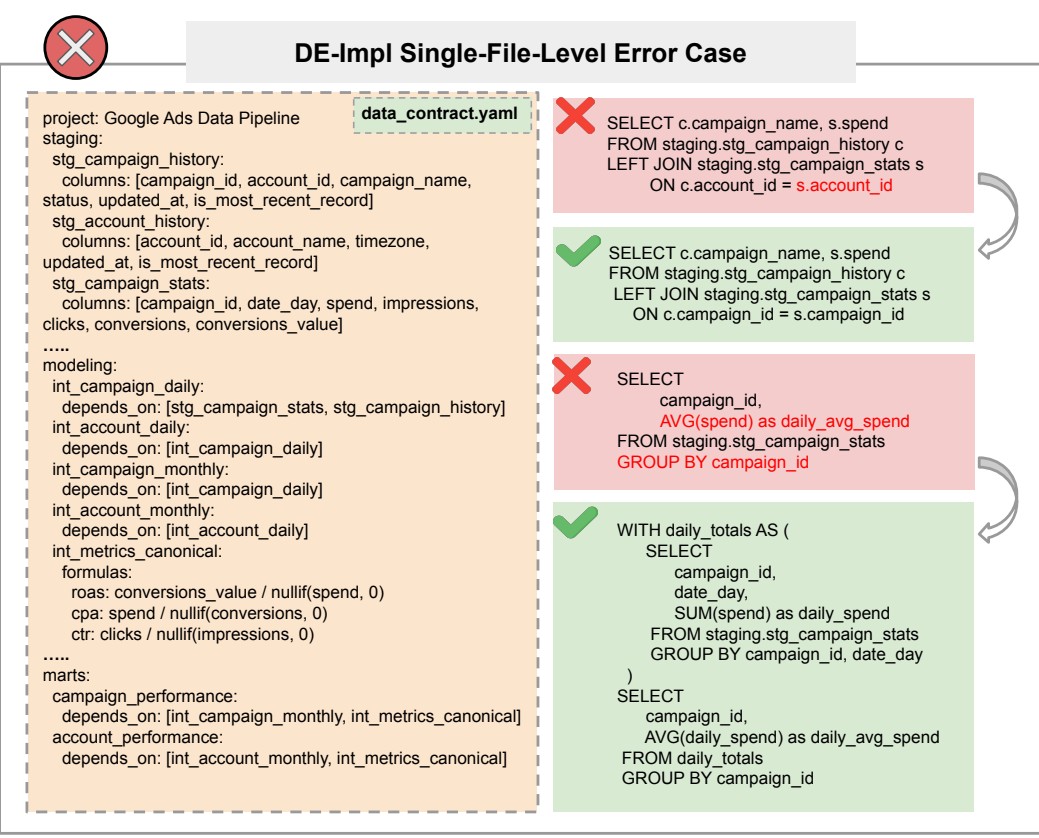

Figure 11: Examples of errors in DE-Impl: the red-crossed cases show mistakes such as joining on mismatched keys (account_id instead of campaign_id) and incorrect aggregation without respecting daily granularity, while the green-checked cases illustrate valid implementations with proper joins and staged aggregation.

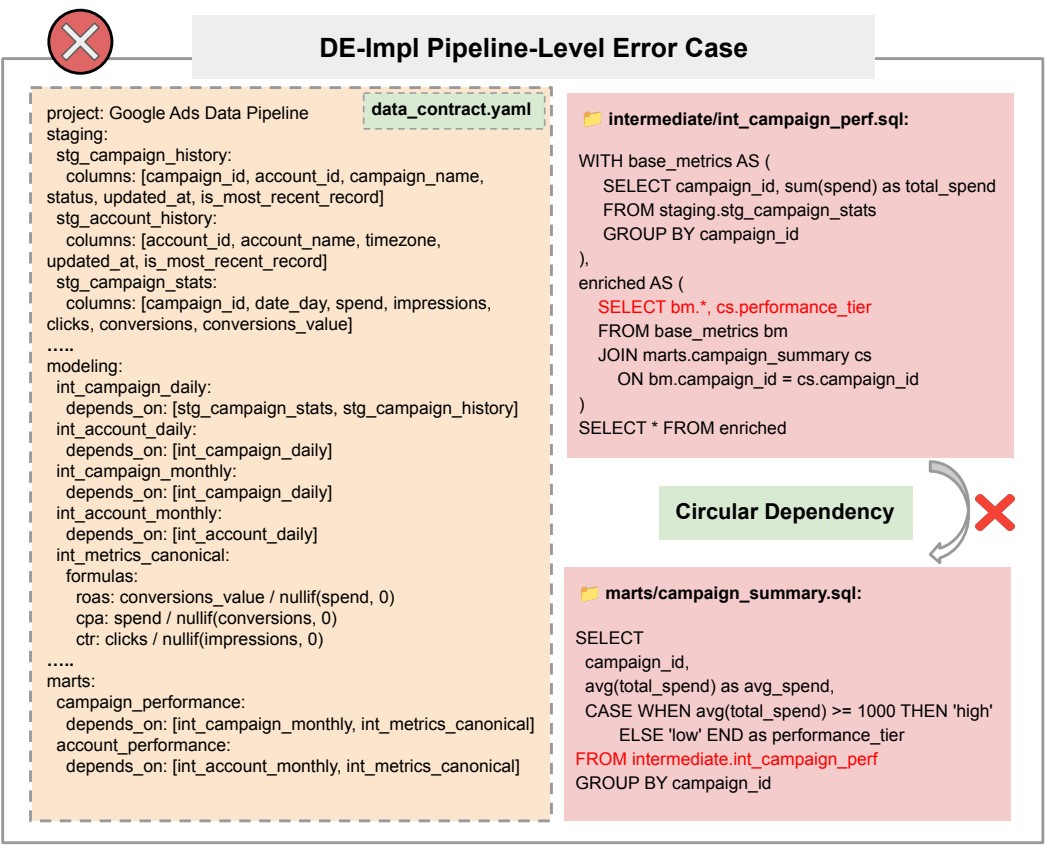

Figure 12: Examples of errors in DE-Impl: a circular dependency in which `int_campaign_perf.sql` depends on `campaign_summary.sql`, creating a loop in the data pipeline.

Table 18: Detailed analysis for SQL Omission.

| Model | Staging | Intermediate | Marts | Total |
|---|---|---|---|---|
| **DE-Impl Tasks** | | | | |
| GPT-5 | 4.14 | 3.66 | 9.37 | 5.18 |
| Gemini-2.5-Pro | 11.22 | 9.58 | 26.67 | 15.17 |
| Qwen3-Coder | 18.68 | 19.00 | 34.56 | 22.88 |
| DeepSeek-V3.1 | 23.91 | 22.73 | 42.23 | 28.58 |
| Qwen3-235B-A22B | 84.26 | 91.74 | 96.81 | 89.79 |
| Qwen3-8B | 92.05 | 97.81 | 99.50 | 95.74 |
| **DE-Evol Tasks** | | | | |
| GPT-5 | – | 8.99 | 15.34 | 11.69 |
| Gemini-2.5-Pro | – | 14.16 | 21.78 | 16.94 |
| Qwen3-Coder | – | 15.08 | 23.47 | 19.09 |
| DeepSeek-V3.1 | – | 30.96 | 39.42 | 34.00 |
| Qwen3-235B-A22B | – | 73.94 | 60.59 | 65.79 |
| Qwen3-8B | – | 66.66 | 48.31 | 58.17 |

Table 19: Detailed performance analysis for Calculation Logic Errors in DE-Impl and DE-Evol tasks.

| Model | Staging | Intermediate | | | Marts | | | All | | |
|---|---|---|---|---|---|---|---|---|---|---|
| | | Upstream Errors | Intrinsic Errors | Total | Upstream Errors | Intrinsic Errors | Total | Upstream Errors | Intrinsic Errors | Total |
| **DE-Impl Tasks** | | | | | | | | | | |
| GPT-5 | 29.85 | 35.78 | 6.95 | 42.73 | 40.47 | 3.95 | 44.42 | 30.41 | 10.05 | 40.46 |
| Gemini-2.5-Pro | 21.05 | 31.68 | 9.97 | 41.65 | 36.41 | 6.37 | 42.78 | 26.62 | 10.96 | 37.58 |
| Qwen3-Coder | 25.11 | 33.23 | 7.71 | 40.94 | 33.60 | 5.63 | 39.23 | 26.03 | 10.88 | 36.91 |
| DeepSeek-V3.1 | 23.94 | 31.93 | 9.93 | 41.86 | 34.65 | 6.21 | 40.86 | 25.77 | 11.68 | 37.45 |
| Qwen3-235B-A22B | 37.08 | 22.78 | 29.16 | 51.94 | 65.91 | 9.09 | 75.00 | 10.82 | 31.24 | 42.06 |
| Qwen3-8B | 42.86 | 42.26 | 36.22 | 78.48 | 74.34 | 10.62 | 84.96 | 5.77 | 28.85 | 34.62 |
| **DE-Evol Tasks** | | | | | | | | | | |
| GPT-5 | – | 10.91 | 18.54 | 29.45 | 20.39 | 12.68 | 33.07 | 14.97 | 13.96 | 28.93 |
| Gemini-2.5-Pro | – | 9.80 | 26.29 | 36.09 | 38.30 | 15.41 | 53.71 | 22.51 | 18.05 | 40.56 |
| Qwen3-Coder | – | 11.64 | 25.57 | 37.21 | 22.53 | 17.61 | 40.14 | 16.09 | 19.57 | 35.66 |
| DeepSeek-V3.1 | – | 9.00 | 20.96 | 29.96 | 20.89 | 17.43 | 38.32 | 14.22 | 17.65 | 31.87 |
| Qwen3-235B-A22B | – | 0.46 | 21.16 | 21.62 | 12.47 | 19.22 | 31.69 | 5.87 | 17.75 | 23.62 |
| Qwen3-8B | – | 0.48 | 25.02 | 25.50 | 10.78 | 25.06 | 35.84 | 5.66 | 22.22 | 27.88 |

Table 20: Detailed analysis for Dependency Errors in DE-Impl and DE-Evol tasks.

| Model | Missing dependencies | Extra dependencies | Missing ∪ Extra |
|---|---|---|---|
| **DE-Impl Tasks** | | | |
| GPT-5 | 45.42 | 52.61 | 66.01 |
| Gemini-2.5-Pro | 51.02 | 50.44 | 67.31 |
| Qwen3-Coder | 47.92 | 50.81 | 66.13 |
| DeepSeek-V3.1 | 48.53 | 49.26 | 65.68 |
| Qwen3-235B-A22B | 61.43 | 55.46 | 73.94 |
| Qwen3-8B | 52.60 | 56.24 | 70.55 |
| **DE-Evol Tasks** | | | |
| GPT-5 | 39.15 | 39.50 | 56.45 |
| Gemini-2.5-Pro | 52.49 | 42.87 | 64.98 |
| Qwen3-Coder | 48.65 | 43.70 | 63.29 |
| DeepSeek-V3.1 | 45.01 | 38.82 | 59.23 |
| Qwen3-235B-A22B | 43.31 | 28.74 | 53.86 |
| Qwen3-8B | 46.92 | 20.88 | 53.44 |

demands on context retention, where limited-capacity models fail to identify the downstream consequences of schema changes.

**Predominance of upstream error propagation in evolution tasks.** The comprehensive error profile presented in Table 17 elucidates a fundamental distinction between the *ab initio* synthesis of Data DAGs in DE-Impl and the structural preservation required in DE-Evol. While the aggregate dependency error magnitude is lower in evolution scenarios (e.g., GPT-5 decreases from 66.01% to 56.45%), the taxonomy of failures exhibits a qualitative shift. As further detailed in Table 20, lower-capacity models in DE-Evol display a marked propensity for "Missing Dependencies," a deviation from the more balanced error profile observed in DE-Impl. This asymmetry suggests that preserving the integrity of an existing pipeline imposes a distinct cognitive burden related to context retention, where models struggle to fully trace the downstream ramifications of schema modifications.

**Architecture-level attrition in file identification.** Table 18 demonstrates a notable performance inversion regarding scope identification. Unlike DE-Impl where the scope is constructive, DE-Evol

demands discriminative identification of files for modification. Surprisingly, for SOTA models like GPT-5, the rate of SQL Omission is higher in DE-Evol (11.69%) compared to DE-Impl (5.18%). This data suggests that the discriminative task of identifying specific files for modification within a large codebase presents a greater challenge than the constructive task of pipeline generation. While base models struggle with the sheer complexity of DE-Impl, advanced models are constrained by the precision required for impact analysis in DE-Evol, indicating that identifying the modification scope remains a distinct bottleneck.

**DE-Evolution error case.** To illustrate how evolution errors can propagate across layers and distort downstream business metrics, we present a pipeline-level DE-Evol example in Fig. 13.

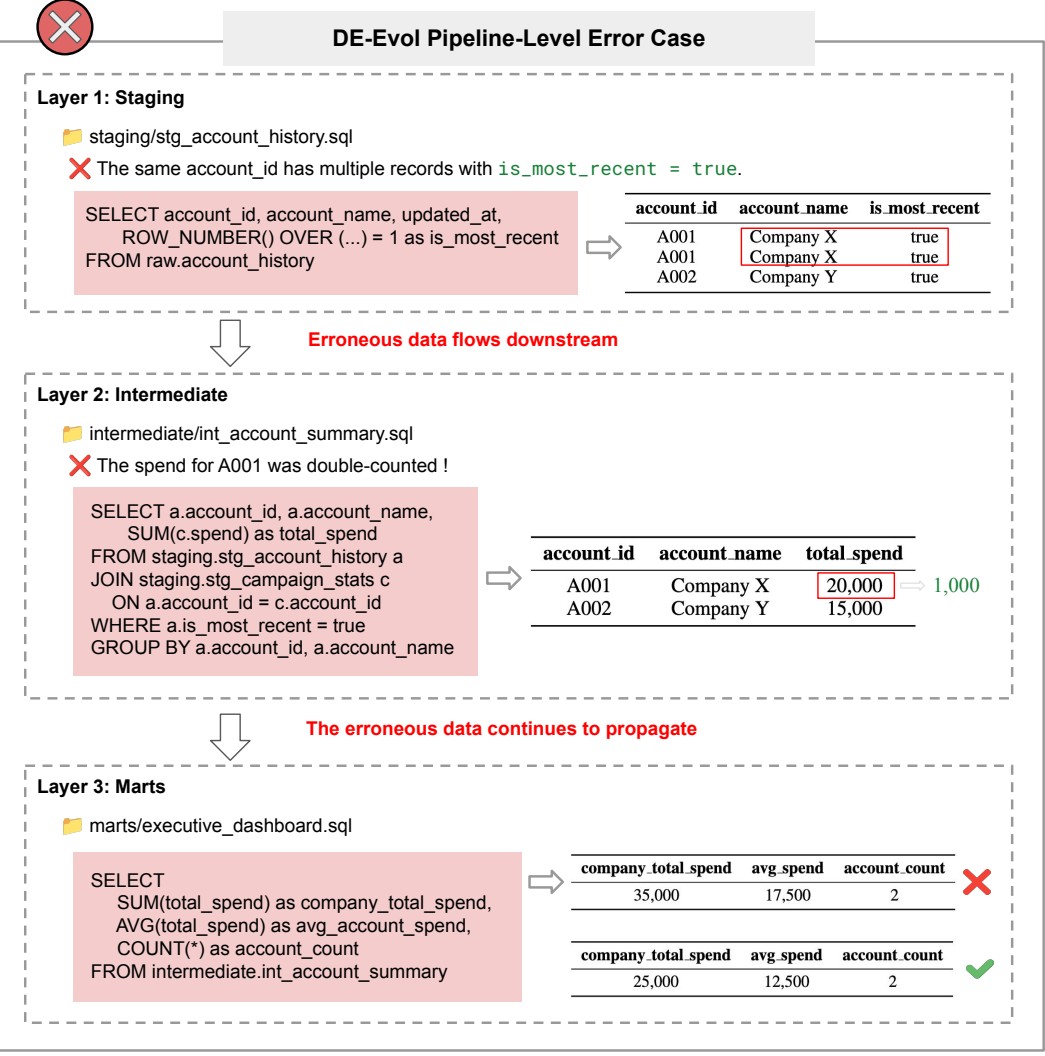

Figure 13: DE-Evol pipeline-level error case. Layer 1 (Staging) contains duplicate "current" rows where the same `account_id` appears multiple times with `is_most_recent = true`. Layer 2 (Intermediate) joins to campaign stats while filtering on `is_most_recent = true`, causing A001's spend to be double-counted (`total_spend` becomes 20,000 instead of 10,000). Layer 3 (Marts) aggregates the erroneous intermediate table, inflating `company_total_spend` and `avg_account_spend` (35,000 and 17,500) compared with the correct values (25,000 and 12,500). The figure highlights how a seemingly small staging inconsistency can cascade into materially incorrect executive metrics.

## D.4 DA CASE

To illustrate typical failure modes in DA tasks, Tab. 21, Tab. 22, and Tab. 23 present focused case studies of Planning, Execution, and Interpretation errors, respectively. First, a scoping lapse omitted required unstructured data, yielding a biased sample and invalidating all downstream analysis. Second, despite a sound plan, a key metric was computed with an incorrect formula (simple average instead of weighted average), producing misleading channel insights. Third, even with flawless calculations, the agent failed to synthesize findings into a context-aware conclusion and omitted mandatory limitations and a safety disclaimer. Together, these cases demonstrate that reliable DA outputs require aligned rigor across *planning, implementation, and interpretation*, with checks that prevent any single stage from compromising the whole.

Table 21: Focused case study of a critical Planning Error. This table analyzes the agent's plan against a pivotal standard (Data Scoping), highlighting omitted steps (in red) that led to a fundamentally flawed analysis.

| Required Planning Step (from Rubric) | Agent's Plan vs. Actual Action | Outcome |
|---|---|---|
| **Case Study: Standard 1.1 — Data Understanding & Scoping** | | |
| **Step 1.1.A.1:** Filter using the structured `Education Requirement` column. | The agent correctly planned and executed this step. | ✓ **PASS** |
| **Step 1.1.A.2:** Additionally extract candidates from the `Job Description` column. | **CRITICAL PLANNING FAILURE:** This step was entirely omitted from the agent's plan; it never considered searching this column. | ✗ **FAIL** |
| **Step 1.1.A.3:** Further apply complex filtering rules to the `Job Description` column. | **CRITICAL PLANNING FAILURE:** This more advanced step was likewise completely absent from the agent's plan. | ✗ **FAIL** |

**Consequence of the Flawed Plan:** By omitting two required data sources in the planning stage, the agent analyzed an incomplete and biased sample (9,073 records instead of the correct 11,838), thereby invalidating all subsequent analysis. This is a textbook Planning Error: once the initial strategy is faulty, execution quality cannot rescue the outcome.
**Final score for this standard: 1 / 4**.

## E ANNOTATION DETAILS

### E.1 DATA COLLECTION

**Data synthesis for de tables.** Our DE tables originate from 73 enterprise-grade SaaS domains and their companion data-transformation projects, providing production-style schemas and realistic dependencies. Starting from a minimal business contract (target grain, primary/foreign keys, required metrics), we expand to end-to-end datasets and scale them while preserving business semantics and referential integrity. To keep the data mock both controllable and realistic, we highlight only the key steps: 1) *Schema fidelity*: retain PK/FK, uniqueness, not-null, and domain constraints; 2) *Distributions & dependencies*: fit marginal distributions and model conditional links (e.g., `country⇒currency/timezone`); 3) *Temporal coherence*: inject seasonality, trend, and holiday effects while maintaining fact–dimension integrity; 4) *Noise & edge cases*: introduce controlled missingness/outliers/type coercions and design stressors that expose pipeline fragility (e.g., duplicate "current" rows, currency conflicts, timezone mismatches). The synthesis pipeline is implemented in Python (pandas, numpy, faker) with custom generators to scale volume while honoring inter-column dependencies and business invariants.

### E.2 CONSTRUCTION DETAILS OF DACOMP-DE

This subsection presents our experience constructing the DAComp-DE corpus. We outline an end-to-end process across three tracks—Architecture, Implementation, and Evolution—spanning the

Table 22: Focused case study of a critical *Execution Error*. This table examines the agent's implementation for Standard 2.1 (Channel Performance Metrics), illustrating how an otherwise sound plan partially failed due to an improper formula for a key metric.

| Calculation Required by the Rubric | Agent Implementation vs. Correct Method | Outcome |
|---|---|---|
| **Case: Standard 2.1 — Channel Performance Metrics** | | |
| **Sub-standard 2.1.A.1:** Compute `Sales Volume` by channel. | The agent correctly used 'GROUP BY' with 'SUM(sales_volume)'. | ✓ **PASS** |
| **Sub-standard 2.1.A.2:** Compute `Total Revenue` by channel. | The agent correctly used 'GROUP BY' with 'SUM(total_revenue)'. | ✓ **PASS** |
| **Sub-standard 2.1.A.3/4:** Compute `Average Unit Price` by channel. | **CRITICAL EXECUTION ERROR:** The agent treated unit price as a simple average rather than a revenue-weighted average. As a result, the reported average prices were incorrect and led to misleading conclusions about channel profitability. | ✗ **FAIL** |

**Impact of the Execution Deviation:** Although the agent's overall plan for channel analysis was sound, using the wrong formula for a single critical metric (Average Unit Price) produced misleading conclusions about channel profitability, directly undermining any price-based strategic recommendation. This constitutes a canonical *Execution Error*.
**Final score for this standard: 5 / 6**.

baseline derived from *73 enterprise-grade SaaS domains and their data-transformation projects* to pure-SQL normalization and validation, high-level requirement setting for blueprinting, contract-driven realization into working SQL, and change-oriented migration under realistic constraints. The summary reflects decisions and best practices agreed upon by domain experts to ensure rigor, reproducibility, and evaluability.

### E.2.1 CONSTRUCTION DETAILS OF DAComp-DE-ARCHITECTURE

**Baseline curation and normalization.** We first select open-source `dbt` projects that are license-compliant and empirically verified to be error-free, and normalize them into pure-SQL repositories by expanding materializations and macros while freezing model dependencies. Senior data engineers conduct a systematic audit of join semantics, analytical grains, window specifications, SCD handling, and testing assumptions, thereby establishing a high-quality baseline suitable for controlled evaluation.

**High-level requirement formulation.** Building on this baseline, we define task statements grounded in realistic enterprise scenarios: they provide only business context, overarching objectives, and expected outputs, without detailed metric definitions, precise calculation rules, or data constraint specifications. Such descriptions emphasize openness and cross-system characteristics, reveal gaps not covered by the existing repository, and intentionally avoid prescribing implementation paths or technical details. The model is expected to autonomously plan a blueprint—identifying key entities and dependencies, delineating layers and boundaries, and completing testing and freshness strategies—ultimately producing an executable architectural blueprint that evaluates its ability to plan end-to-end SQL projects and set constraints under incomplete information.

### E.2.2 CONSTRUCTION DETAILS OF DAComp-DE-IMPLEMENTATION

**Contract formalization.** DE-Impl is constructed by deriving a rigorous requirements specification from the vetted SQL baseline in the form of a standardized `data_contract.yaml` that follows enterprise conventions. The contract formalizes model inventory and lineage, table and column

Table 23: Focused case study of a critical Interpretation Error. The table shows a stark contrast between the agent's successful execution of calculations and its failure to synthesize those results into a meaningful, context-aware conclusion.

| Analytical Stage (from Rubric) | Agent's Performance & Justification | Outcome |
|---|---|---|
| **Case Study: Analysis of Students with Suicidal Ideation** | | |
| **Stage 1: Execution & Calculation** (Standards 1.1 – 1.4) | The agent's plan was sound and its execution was flawless. It successfully filtered the correct data population and accurately calculated all required statistical metrics (e.g., average economic/academic pressure, lifestyle habit percentages). | ✓ **PASS** |
| **Stage 2: Interpretation & Synthesis** (Standard 1.5: Create a "high-risk profile") | **CRITICAL INTERPRETATION FAILURE:** The agent failed to synthesize the previously calculated statistics into a coherent, higher-level insight. Instead of creating a "profile," the agent merely listed the numbers again. The judge noted the summary was **"not deep enough"** and **"merely restated the table's content."** | ✗ **FAIL** |
| **Stage 3: Contextual Understanding** (Standard 2.2: Provide safety disclaimer) | **CRITICAL INTERPRETATION FAILURE:** The agent's final output completely omitted the mandatory "Limitations and Safety Disclaimer." This demonstrates a failure to understand the serious and sensitive context of the topic, which is a key part of providing a responsible and complete analytical deliverable. | ✗ **FAIL** |

**Consequence of Flawed Interpretation:** This case exemplifies a pure Interpretation Error. The agent acted as a perfect calculator, producing correct data (Stage 1). However, it failed at the final and most critical stage: transforming that data into a meaningful, insightful, and contextually appropriate conclusion (Stages 2 & 3).

schemas with constraints, declared grains and time windows, metric definitions with coherent units and currency normalization, as well as data quality, freshness, and performance policies.

### E.2.3 CONSTRUCTION DETAILS OF DAComp-DE-Evolution

**Change specification.** For DE-Evol, we start from a high-quality, production-style SQL repository and propose change requests driven by realistic enterprise pressures—such as revised metric definitions, altered analytical windows, schema drift, or governance hardening. Multiple experts specify unambiguous business semantics, distinguish breaking from non-breaking changes, and design a safe migration plan that anticipates dependency revisions and testing upgrades.

### E.3 ANNOTATION DETAILS OF DAComp-DA

In this section, we present the experience regarding the annotation of DAComp-DA data, which is summarized from our previous project discussion meetings and alignment meetings.

### E.3.1 CORE DESIGN PRINCIPLES

**Strategic diversity**   The core of the Rubric is to evaluate problem-solving *strategies*, not *steps*. Each scoring Path must represent a methodologically distinct and self-contained solution. We avoid designing *complete* versus *abridged* versions of the same path. For example, *analyzing all provinces* and *analyzing a subset of provinces* should not be two separate Paths; the latter is merely an incomplete execution of the former.

**Objective evaluation**   Scoring criteria must be quantifiable and reproducible to minimize scorer subjectivity. All items should be based on explicit evidence. Guideline: Any Accuracy item requiring numerical verification must have a pre-calculated Anchor Value. For open-ended paths without a single correct answer, a Pseudocode or a clear methodological verification process must be provided.

**Dimensional separation of abilities**   Complex analytical skills are decomposed into independent scoring dimensions for a fairer and more granular assessment of model performance. Guideline: Strictly distinguish between *procedural execution* (were the steps completed?), *computational accuracy* (were the numbers correct?), and *insightful conclusion* (was the interpretation meaningful?), designing them as separate scoring items.

### E.3.2 STRUCTURAL COMPONENTS OF THE RUBRIC

The Rubric employs a four-level hierarchical structure to deconstruct tasks, ensuring comprehensive and granular evaluation.

**Requirement.**   Definition: The highest-level objective of the task, directly corresponding to a core analytical request from the user. Example: *Analyze the differences in employee attrition rates across departments and their causes.*

**Standard.**   Definition: A key analytical step that must be completed or a core conclusion that must be reached to fulfill a Requirement. Example: *Standard 1: Calculate and verify the attrition rate differences between departments*; *Standard 2: Identify the key factors causing these differences.*

**Path.**   Definition: A methodologically distinct and valid strategy for meeting a Standard. This is the core of the Rubric's design. Example: Under the standard of *verifying differences*, Path A could be *performing a statistical significance test (e.g., Chi-squared test)*, while Path B could be *making a descriptive statistical comparison (e.g., percentage difference).*

**Sub-standard / rubric item.**   Definition: The smallest scorable unit of the Rubric, nested under a specific Path and adhering strictly to the principle of dimensional separation. It comprises three main types:

- Completeness: Assesses whether all required steps for a given Path were executed. Focuses on *what was done*.

- Accuracy: Assesses whether the computational results or execution process are correct. Focuses on *if it was done correctly*. For deterministic paths, this is verified against an Anchor Value; for open-ended paths, it is verified against a methodological process or Pseudocode.

- Insightfulness: Assesses whether a reasonable and valuable conclusion or insight was derived from the correct results. Focuses on *if the results were understood*.

### E.3.3 GOLDEN RULES FOR AUTHORS

These are the disciplinary requirements to ensure the quality and consistency of the Rubric. While these guidelines ensure consistency in creating rubrics for known strategies, the following section details our methodology for fairly evaluating novel or unanticipated solutions that may not align with pre-enumerated paths.

**Calculate first, then author.** Before finalizing the rubric, authors must personally run the complete analysis with code to calculate all Anchor Values required for the Accuracy assessment. This is the cornerstone of ensuring objective scoring.

**Be specific and unambiguous.** Every statement in the Rubric must be directive and unambiguous. Avoid subjective terms like *approximately*, *good*, or *relatively comprehensive* to minimize scorer discretion.

**Avoid zero-point paths.** If a method is not worthy of credit, it should not be designed as a distinct Path. A model's output that does not match any valid path will naturally receive no score for that standard.

## F    LLM USAGE DETAILS

In compliance with ICLR 2026 policies on large language model usage, we disclose that LLMs are mainly used for three purposes in this work:

- LLM Evaluation: The core of this work is the systematic benchmarking of various large language models to assess their performance and capabilities as data agents.

- LLM-based Judging: For the open-ended tasks in our benchmark, we employed an LLM as an automated judge to score agent responses based on a detailed, expert-designed rubric.

- Writing Assistance: We utilized an LLM to assist in polishing the manuscript by refining grammar, improving phrasing, and enhancing overall clarity.

All LLM outputs were subject to careful human oversight and validation. We take full responsibility for the accuracy and integrity of all content in this paper, including any sections enhanced with LLM assistance.

## G    DISCUSS

### G.1    DISCUSSION OF HANDLING UNENUMERATED SOLUTION PATHS

Accuracy is the most critical dimension in our rubric. Since fully listing all valid analytical paths is often infeasible, we adopt a *three-tier, progressively relaxed* design for Accuracy: (i) *direct enumeration* with numeric anchors when the correct outcome can be exhaustively determined; (ii) *constrained computation* with pseudo-code anchors when procedures are well-defined but paths are not exhaustively enumerable; and (iii) *principle-based* assessment for highly open-ended cases.

**Standardized assessment for common paths.** We standardize scoring whenever we can verify correctness deterministically. *Tier-1 (numeric anchors):* for tasks whose outcomes can be *exhaustively enumerated*, we embed the reference value directly into the rubric (e.g., "How many users satisfy condition $X$?"), yielding absolute, reproducible checks. *Tier-2 (pseudo-code anchors):* for

tasks with well-specified computation but multiple equivalent derivations (e.g., a conversion rate with alternative weighting schemes), we prescribe canonical steps in *pseudo-code* to constrain the procedure. This enables process-level verification (inputs, ordering, aggregation, null/edge handling) without enumerating every path, preserving both precision and reproducibility.

**Principle-based assessment for novel paths.** A minority of tasks are intrinsically open-ended, where enumeration or pseudo-code templating is impractical. Here we evaluate Accuracy via *methodological principles* rather than a single anchor value. For example, a "key-driver identification" task may be solved by regression with coefficient interpretation (a pre-defined path), or by gradient boosting with SHAP attributions (an unenumerated path). We score such solutions on: *(1) Methodological appropriateness* (the method is suitable for the stated objective and data regime); *(2) Correctness of execution* (the pipeline is implemented soundly, with valid preprocessing, estimation, and validation); and *(3) Soundness of interpretation* (claims follow from the produced evidence, with clear caveats). This soft layer ensures valid but unconventional approaches are not penalized.

By construction, most DAComp items fall into Tiers 1–2, where numeric or pseudo-code anchors provide deterministic checks; Tier 3 is reserved for genuinely open-ended cases to maintain fairness without sacrificing rigor.

## G.2 Discussion of Ambiguous of Requirements

Implementation and Evolution tasks in DAComp-DE are designed as deterministic evaluations. To balance *realism* with *unambiguous executability*, we adopt three principles:

**1) Professionalism.** Requirements are sourced from enterprise-style projects and vetted by senior data engineers for cross-layer impact, metric definitions, SCD handling, and temporal semantics. *Implementation* tasks emphasize canonical modeling pipelines from scratch; *Evolution* tasks mirror real "change requests" (e.g., metric revision, source replacement).

**2) Unambiguity.** *Implementation (node-first):* each SQL node has atomic contracts (schema, PK/grain, time, nulls, joins, aggregation, SCD, idempotency). Multiple agents must converge under frozen contracts; discrepancies trigger tighter specifications. *Evolution (delta-first):* natural-language changes are mapped into minimal verifiable deltas (schema/logic/lineage), with explicit impact scope and before–after anchors; agent disagreement leads to refined deltas or explicit assumptions.

**3) Realism.** *Implementation:* converged nodes are composed into multi-node tasks, with contracts and assumptions documented (e.g., `data_contract.yaml`). *Evolution:* favors backward-compatible evolution (added columns/views, metric versioning); destructive changes require migration notes. All assumptions are logged for reproducibility.

## G.3 Discussion on the Selection of Judger LLM

As shown in Tab. 7, both O4-Mini and gemini-2.5-flash achieve human-level agreement, while stronger proprietary models (e.g., gemini-2.5-pro, GPT-5) yield even higher consistency. For DAComp,, we standardize on gemini-2.5-flash, as it balances (1) cost efficiency for large-scale benchmarking, (2) stable and low-latency inference, (3) reproducibility across runsand (4) community accessibility. Choosing a widely available model ensures that our evaluation pipeline can be easily adopted, verified, and extended by others.

## G.4 Discussion on End-to-End Evaluation

Current DAComp tasks span complementary stages of the data intelligence lifecycle: DE-Architecture (high-level specification and planning), DE-Implementation (multi-layer pipeline construction), DE-Evolution (safe modification under requirement changes), and DA (open-ended analysis over downstream data). Taken *together*, these stages delineate a strictly end-to-end process—from requirement articulation, through system realization and iterative evolution, to analytical insight and decision support—covering a full loop from planning and implementation to evolution and interpretation.

At present, we evaluate these stages modularly and in a decoupled fashion to enable controlled measurement at each step. Our next key objective is to integrate them into a single, end-to-end longitudinal evaluation: a single agent carries requirements through implementation and change propagation, and ultimately completes analysis and reporting. We contend this end-to-end setup offers substantial scientific and practical value: it stress-tests the *end-to-end consistency* of planning–execution–evolution–interpretation, better reflects real engineering workflows, and advances toward a comprehensive assessment of autonomous data agents' end-to-end capabilities.

