# OpenReview forum: "DAComp: Benchmarking Data Agents across the Full Data Intelligence Lifecycle"
_ICLR.cc/2026/Conference — ICLR 2026 Poster_

### Official Review · Reviewer_cC4w · 2025-10-30

**Soundness:** 4
**Presentation:** 4
**Contribution:** 3
**Rating:** 8
**Confidence:** 5

**Summary:**

This paper introduces DAComp, a comprehensive benchmark designed to evaluate LLM-based data agents across the entire enterprise data intelligence lifecycle, encompassing both data engineering (DE) and data analysis (DA) tasks.

DAComp includes 236 tasks divided into repository-level engineering (architecture, implementation, and evolution) and open-ended analysis tasks that require reasoning, planning, and synthesis of insights. The DE tasks are evaluated through execution-based metrics (Component Score, Cascading Failure Score, Success Rate), while DA tasks are judged using a validated LLM-as-judge framework with hierarchical rubrics and GSB scoring.

The paper presents extensive experiments across leading models (GPT-5, Claude-4, Gemini-2.5, DeepSeek-V3.1, Qwen3, etc.), showing that even state-of-the-art agents perform poorly (<50 % overall, <20 % DE success rate). The authors conclude that current LLM agents lack holistic orchestration and open-ended analytical reasoning.

Contributions:

First benchmark to jointly evaluate repository-level engineering and open-ended analytical reasoning.

Novel hierarchical rubric framework for open-ended evaluation with verified human–LLM agreement.

Large-scale empirical study exposing the limits of current LLM data agents.

**Strengths:**

Originality: Defines a new, holistic problem space combining engineering and analysis in realistic data workflows.

Methodological rigor: Combines automatic and rubric-based evaluations, validated with human judgments.

Scale and realism: Repository-level tasks (>4k LOC) and open-ended business questions reflect real enterprise workloads.

Comprehensive evaluation: Benchmarks diverse models, offering insights into distinct skill domains (engineering vs reasoning).

Actionable findings: Identifies orchestration and multi-step reasoning as key bottlenecks for next-generation data agents.

Excellent presentation: Clear structure, detailed analysis, strong alignment with related work.

**Weaknesses:**

Limited exploration of agent learning improvements: The paper benchmarks performance but does not propose or test training strategies to overcome observed limitations.

Restricted open-source availability: While the paper claims data/code release, the double-blind setup prevents verification; explicit examples of released task formats would strengthen reproducibility.

Evaluation cost: The LLM-as-judge pipeline, though validated, may pose high computational costs for community replication; approximate or lightweight scoring alternatives would be valuable.

Slight imbalance between DE and DA sections: DE is analyzed more deeply than DA in terms of ablations and case studies.

**Questions:**

1- How does the cost of LLM-judging scale with benchmark size, and can smaller-scale variants be used for quick evaluations?
2- Could the authors clarify whether the hierarchical rubrics generalize to unseen tasks, or are they manually constructed per task?
3- How is task realism maintained when synthetic data are used—were human engineers asked to validate their authenticity?

---

> ### Author Response · Authors · 2025-11-20
> **Official Comment by Authors (1/5)**
>
> We sincerely thank you for recognizing our work. We are deeply grateful for your endorsement and constructive suggestions, which are invaluable to us. We are particularly pleased that you highlighted the **originality**, **scale**, **realism**, and **presentation quality** of our paper.
>
> We also noticed you have raised several constructive questions about our work, and we are happy to elaborate further below.
>
> ---
>
> > **W1: Limited exploration of agent learning improvements: The paper benchmarks performance but does not propose or test training strategies to overcome observed limitations.**
>
> Thank you for your suggestion — it is absolutely correct. Our paper currently focuses primarily on **benchmarking**. We have not yet implemented **scaling data**, **SFT**, or **RL-based** training pipelines.
> In this version of the paper, we mainly propose an **optimized data-agent framework** to carry out the evaluation.
>
> As described in **Section 4.5**:
> - For **DE tasks**, beyond the baseline, we introduce a **multi-agent framework** to mitigate long-context challenges in repo-level engineering. Sub-agents specialize in individual layers, while a top-level planner manages orchestration and validation. Each generated SQL component is **independently verified in isolation**, and single-agent mode also supports **self-checking**, significantly improving multi-stage coordination.
> - For **DA tasks**, we design a **three-module multi-agent workflow** (report generation, visualization, integration), and observe **consistent performance gains** across analytical dimensions for a wide range of LLMs.
>
> Experimental details can be found in **Table 3** and **Table 4**.
>
> ---
>
> > **W2: Restricted open-source availability: While the paper claims data/code release, the double-blind setup prevents verification; explicit examples of released task formats would strengthen reproducibility.**
>
> We sincerely thank the reviewer for emphasizing **reproducibility**, which we fully agree is crucial. To facilitate immediate verification, our anonymous repository already hosts **representative task examples**, including detailed **task formats**, **data schemas**, and **evaluation logic**.
>
> We are currently finalizing comprehensive anonymization of the full dataset and codebase to ensure strict compliance with the conference’s double-blind policy. The **complete DAComp benchmark suite** will be released in the same repository once anonymization is complete.
>
> We remain fully committed to **open science** and guarantee full availability of all assets.

---

> > ### Author Response · Authors · 2025-11-20
> > **Official Comment by Authors (2/5)**
> >
> > > **W3: Evaluation cost: The LLM-as-judge pipeline, though validated, may pose high computational costs for community replication; approximate or lightweight scoring alternatives would be valuable.**
> >
> > We thank the reviewer for raising the valid concern regarding the computational cost of the LLM-as-a-judge pipeline for community replication. We have optimized our evaluation strategy specifically to mitigate this issue, ensuring accessibility without compromising rigor.
> >
> > **1. Cost-Efficiency by Design (Gemini-2.5-Flash)**
> > Our standard judge is **Gemini-2.5-Flash**, which we selected precisely because it offers an optimal balance between performance and economy. As shown in **Table 1** (Human-Model Alignment), it achieves exceptional reliability—**Rubric Item $\kappa_w=0.834$** and **Case ICC=0.890**—effectively matching human inter-rater levels and outperforming much larger models. Crucially, it is approximately **10$\times$ cheaper** than alternatives like o4-mini, making the judging process significantly more economical than typical proprietary model evaluations. Furthermore, we emphasize that **only DA tasks (42% of the benchmark)** require LLM judging; the majority of computation-heavy DE tasks rely on deterministic, execution-based metrics that incur zero inference cost.
> >
> > **Table 1: Inter-rater and human–model agreement.**
> > *Gemini-2.5-Flash achieves human-level consistency at a fraction of the cost.*
> >
> > | Model / Metric | **Rubric**<br>Item ($\kappa_w$) | **Rubric**<br>Case (ICC) | **Rubric**<br>Model ($\tau_b$) | **GSB**<br>Item ($\kappa_w$) | **GSB**<br>Case (ICC) | **GSB**<br>Model ($\tau_b$) |
> > | :--- | :---: | :---: | :---: | :---: | :---: | :---: |
> > | **Human Inter** | **0.906** | **0.925** | **1.000** | **0.531** | **0.831** | **1.000** |
> > | **o4-mini** | 0.827 | 0.881 | 1.000 | 0.539 | 0.792 | 1.000 |
> > | **Gemini-2.5-Flash** | **0.834** | **0.890** | **1.000** | 0.534 | 0.814 | 1.000 |
> > | **GPT-4.1** | 0.797 | 0.848 | 1.000 | 0.526 | 0.780 | 1.000 |
> > | **Qwen3-235B** | 0.737 | 0.758 | 1.000 | 0.491 | 0.713 | 1.000 |
> > | **Qwen3-30B** | 0.680 | 0.775 | 1.000 | 0.467 | 0.691 | 1.000 |
> >
> > **2. Validated Open-Source Alternatives**
> > To further support community replication, we evaluated lightweight open-source alternatives. As detailed in **Table 1**, while proprietary models generally perform better on subjective GSB dimensions (e.g., Readability), open-source models like **Qwen3-235B** and **Qwen3-30B** still maintain high consistency on objective Rubric metrics ($\kappa_w \approx 0.74$) and perfect model-level ranking correlation ($\tau_b = 1.000$). This confirms that while we recommend Gemini-2.5-Flash for the highest fidelity, the **Qwen series serves as a viable, low-cost substitute** for researchers who prioritize open weights, ensuring the benchmark's broad accessibility.
> >
> >
> >
> >
> > > **W4: Slight imbalance between DE and DA sections: DE is analyzed more deeply than DA in terms of ablations and case studies.**
> >
> >
> > We appreciate the constructive observation. To align the analytical depth of the DA section with the DE section, we have expanded our analysis in the revised manuscript:
> >
> > 1. Granular Multi-Dimensional Analysis (Sec. 3.4) We moved beyond aggregate scores to analyze six distinct dimensions (e.g., Accuracy, Readability, Depth). This granular view reveals critical nuances, such as o3 achieving high technical accuracy yet failing in readability—insights that a single score would miss.
> >
> > 2. Rigorous Judge Validation (Sec. 3.5) We introduced a comprehensive meta-evaluation of the LLM-as-a-Judge framework. Section 3.5 now validates the judge across five robustness dimensions (including Human Alignment, Cross-Judge Consistency), ensuring the DA evaluation is as scientifically rigorous as the execution-based DE metrics.

---

> > > ### Author Response · Authors · 2025-11-20
> > > **Official Comment by Authors (3/5)**
> > >
> > > > **Q1: How does the cost of LLM-judging scale with benchmark size, and can smaller-scale variants be used for quick evaluations?**
> > >
> > >
> > > We thank the reviewer for this practical inquiry. As this point aligns closely with the cost concerns raised in W3, we provide a complementary perspective focused on scalability and rapid iteration:
> > > 1. **Scalability via Optimized Judge**: While judging costs scale linearly with benchmark size, we mitigate this by standardizing on Gemini-2.5-Flash. It matches human reliability (Case ICC=0.890) at approximately 1/10th the cost of frontier alternatives (e.g., o4-mini), ensuring that scaling the benchmark remains economically feasible for the community.
> > > 2. **Viability for Rapid Evaluation**: Smaller variants are highly viable for quick iterations. Our Cross-Judge Consistency analysis confirms that lightweight open-source models (e.g., Qwen3-30B + Qwen3-VL-30B) maintain a high ranking correlation ($\tau_b \approx 0.90$) with the primary judge. We recommend using these efficient models as "ranking proxies" for rapid local development and debugging, while reserving the full pipeline for final leaderboard reporting.
> > >
> > >
> > > > **Q2: Could the authors clarify whether the hierarchical rubrics generalize to unseen tasks, or are they manually constructed per task?**
> > >
> > >
> > > As stated in Section 2.3, because our goal is to build a rigorous benchmark, the rubrics are indeed manually annotated. The hierarchical rubrics are constructed per task, but they follow a general and reusable rubric protocol that we apply consistently across all 100 DA tasks. To ensure generalization, we evaluated the rubrics against outputs from ten diverse LLM families and multiple agent frameworks, and all unseen but valid solution strategies were successfully handled via path matching or soft-constraint rules. Thus, while task-specific, the rubric framework is methodologically generalizable and robust to unseen analytical approaches.
> > >
> > > As shown in Table 5, we also validate that the rubric-based evaluation achieves a high level of agreement with human judgments.
> > >
> > > In the early stages of development, we also experimented with checklist-style evaluation. However, we found that this approach was overly coarse, produced many false positives, and failed to provide meaningful guidance for assessing model performance. This motivated our hierarchical rubric design, which offers a much more precise and reliable framework for evaluating open-ended data analysis tasks.

---

> > > > ### Author Response · Authors · 2025-11-20
> > > > **Official Comment by Authors (4/5)**
> > > >
> > > > > **Q3: How is task realism maintained when synthetic data are used—were human engineers asked to validate their authenticity?**
> > > >
> > > > Thank you for your insightful suggestions, which have been extremely helpful in improving our paper. We described our data collection and evaluation setup in Sec. 2.3 and App. E of our original paper. As you noted, the authenticity and construction process is critical, and we will elaborate on this in detail here, as previewed in Section 2.3, paragraph "Evaluation construction".
> > > >
> > > > #### DA (Data Analysis) Task Construction
> > > >
> > > > Our DA tasks are grounded in real-world data, and the question annotation process follows a rigorous three-stage filtering pipeline to ensure quality and realism:
> > > >
> > > > 1.  **Question Generation:** For each data table, we tasked three experienced expert annotators with proposing five open-ended, analytical questions based on the data.
> > > > 2.  **Internal Voting:** The annotators then internally voted to select the top 2-3 questions they collectively deemed to be of high quality, relevance, and difficulty. This internal peer-review stage has a very high bar, with a retention rate of only ~10%.
> > > > 3.  **Final Vetting:** The Core Authors conducted a final review of these high-quality candidates to select the official examples for DAComp-DA, ensuring each task meets our standards for a challenging, open-ended analysis.
> > > >
> > > > #### DE (Data Engineering) Task Construction
> > > >
> > > > For the DE tasks, we focused on enterprise-level realism, which we ensured by validating three separate components:
> > > >
> > > > **(1) Schema Authenticity**: The DE schemas are not synthetic. They are sourced directly from **real-world, public data transformation projects** for complex platforms such as Salesforce, Google Ads, and others.
> > > >
> > > > **(2) Data Verisimilitude**: We populated these real-world schemas using a combination of LLMs and code-generation pipelines. Using actual proprietary enterprise data for an open benchmark is legally untenable. Our synthesis pipeline ensures data diversity and maintains relational integrity. Furthermore, as the DAComp-DE tasks primarily evaluate **code generation** (i.e., transforming schema A to schema B), the specific data values do not significantly impact the evaluation of the model's structural and logical coding capabilities.
> > > >
> > > > **(3) Task Authenticity**: The realism of the tasks themselves was our highest priority, which we developed through a multi-stage, expert-driven process:
> > > >
> > > > * **For DE-Impl:** This task assesses the ability to build a data engineering project from scratch. We developed these tasks by **reverse-engineering** complete, publicly-sourced DE projects. These projects already provided a strong baseline of realism with authentic SQL code. To align with complex enterprise standards, we then methodically increased the complexity of these open-source SQL bases.
> > > > * **For DE-Evol:** We engaged in in-depth consultations with professional data engineering teams. We designed these tasks to **highly mimic the complex change requests** and development requirements they encounter in their daily work, and then further escalated the complexity to create a challenging benchmark.
> > > > * **Expert Review:** After our initial annotation, we had a professional data engineering team conduct random sampling and code reviews of our DE-Impl and DE-Evol examples. Based on their expert feedback, we performed multiple rounds of iterative updates.
> > > >
> > > > The final tasks are designed to be (1) highly representative of real-world data engineering challenges and (2) capable of guiding the future iterative development of more capable LLM agents.

---

> > > > > ### Author Response · Authors · 2025-11-20
> > > > > **Official Comment by Authors (5/5)**
> > > > >
> > > > > We sincerely appreciate your detailed feedback. We hope the above response can address all your concerns. If you have any questions, we are pleased to provide further clarification!
> > > > > We hope that DAComp-DE and DAComp-DA will become influential and widely adopted benchmarks for evaluating future data agents, helping to guide the progress of this field.

---

> ### Comment · Reviewer_cC4w · 2025-11-28
>
> Hi. Thanks for you detailed answer.
>
> My concerns are answered.
>
> Considering my initial high score, I will keep my score.

---

### Official Review · Reviewer_uR7M · 2025-10-31

**Soundness:** 2
**Presentation:** 3
**Contribution:** 3
**Rating:** 4
**Confidence:** 4

**Summary:**

The paper introduces DAComp, a benchmark suite intended to evaluate autonomous data‑agent systems across the full data‑intelligence lifecycle. DAComp comprises 236 tasks split into two families: (i) DE (repository‑level data‑engineering) tasks that require agents to design, implement, or evolve multi‑stage SQL pipelines on large, enterprise‑style schemas, and (ii) DA (open‑ended data‑analysis) tasks that ask agents to produce reports, insights, and recommendations. The authors propose a mixed evaluation protocol: deterministic DE tasks are scored with three execution‑based metrics (Component Score, Cascading‑Failure Score, Success Rate), while DA tasks are judged by an LLM‑based evaluator using hierarchical rubrics and a Good‑Same‑Bad (GSB) component. A large experimental study compares a range of proprietary and open‑source LLM agents (GPT‑5, Claude‑4‑Sonnet, Gemini, DeepSeek, Qwen3, etc.) under two agent frameworks (a ReAct‑style baseline and OpenHands). The authors also validate the LLM‑judge against human annotators and report benchmark stability over multiple runs.

**Strengths:**

1. **Holistic scope** – By covering both repository‑level engineering and open‑ended analysis, DAComp fills a clear gap in existing agent benchmarks that usually focus on isolated code generation or single‑turn QA.
2. **Realistic task design** – Tasks are built from 73 permissively‑licensed enterprise SaaS schemas, with synthetic data that respects realistic column distributions, referential integrity, and edge‑case noise.  The DE‑Impl/Evol tasks involve multi‑file, multi‑layer pipelines (≈ 4600 LOC), which is substantially larger than prior data‑science benchmarks.
3. **Rigorous evaluation methodology** – The three‑tiered execution metrics for DE provide fine‑grained diagnostics (component vs. pipeline level).  The hierarchical rubrics for DA are carefully constructed, with multiple valid solution paths and a validation study showing high LLM‑judge/human agreement (Pearson ≈ 0.80, κ ≈ 0.65).
4. **Comprehensive analysis** – The authors present detailed error‑type breakdowns (planning, execution, interpretation) and relate performance to concrete properties such as node count, line count, and turn‑count distributions.
5. **Potential impact** – By exposing the engineering‑orchestration bottleneck, the benchmark can steer future research toward better planning, tool‑use, and pipeline consistency, which are crucial for socially relevant domains (healthcare analytics, climate data pipelines, sustainability reporting).

**Weaknesses:**

- While the authors report strong correlations with human ratings, the statistical choices and reporting could be more rigorous. Using Pearson r on rubric sums (often ordinal/heterogeneous across tasks) is suboptimal; intraclass correlation (ICC) or Kendall’s tau for ranking, plus confidence intervals, would be preferable.
- Weighted κ=65 for GSB indicates only moderate agreement; the paper calls overall alignment “high” but should temper claims or provide confidence intervals and calibration plots.
- The judge model (Gemini-2.5-Flash) belongs to the same family as one of the evaluated models (Gemini-2.5-Pro). Although Flash≠Pro, the potential for family-specific biases remains. The paper presents alternative judges, but a full ranking-stability analysis across judges is not reported.
- GSB depends on five baseline reports created from multiple LLM outputs; selection of baselines may anchor judgments. The process to ensure these baselines neither overfit to a model family nor penalize stylistic variance is under-specified.
- For principle-based (Tier-3) cases, safeguards to prevent over-crediting fluent but methodologically weak responses rely on judge prompts and principles, but inter-rater checks on these specific cases are not separated and reported.
- Token budgets, tool-call limits (200), timeouts (120s per action), prompts, and system settings are critical for agent performance but are only briefly summarized. A stricter ablation on these constraints (and their fairness across models) would clarify whether some models are disadvantaged.
- SQL dialect and tolerance: The evaluation requires exact schema+data equality, but it is unclear if numeric tolerances, null-ordering, and type coercions are handled consistently; small floating-point or timestamp discrepancies can introduce false negatives.
- Contamination risk: Some DE tasks originate from open-source dbt projects; without a contamination analysis, models might have seen parts of the ground truth or closely related code. The synthetic data helps, but existence of public repo code raises a nontrivial risk.
- For DA tasks, stability is reported on a subset (DE-Arch and DA) across 8 runs for a few models; confidence intervals and significance tests for between-model differences are not provided.
- For DE deterministic tasks, results appear single-run; while determinism reduces variance, reporting robustness across multiple seeds/decoding settings would be informative for agent frameworks that involve LLM sampling.
- Some interpretations could be more cautious: “Strong evidence that engineering and analysis are distinct capabilities” is supported by rank inversions, but causality is not established. Alternative explanations (prompt specializations, token policies, execution tool robustness) could contribute; a cross-task, cross-prompt ablation would strengthen this claim. “Even the top agents barely reach 50%” is accurate for aggregate scores, but the consequences for practical utility would benefit from error cost analysis (e.g., what classes of failures materially change business decisions vs. minor mismatches).
- While the paper includes a brief LLM‑usage disclosure, it does not discuss potential risks of releasing synthetic enterprise data (e.g., inadvertent leakage of real company patterns) or the impact of using proprietary LLM judges on reproducibility for the broader community.

**Questions:**

1. How do the DA scores change if a different LLM (e.g., GPT‑4.1 or Claude‑3) is used as the judge? Have you measured inter‑judge agreement?
2. Can you provide an ablation study varying α (e.g., 0.5, 0.7, 0.9) and report its impact on model rankings?
3. What is the distribution of difficulty (e.g., number of nodes, LOC) across DE tasks? Are the results driven by a small subset of very hard tasks?
4. Have you measured human performance on a sample of DE and DA tasks to contextualize the reported model scores?
5. Which random seeds, library versions, and hardware configurations were used? Are the execution scripts deterministic?
6. How were the synthetic data distributions validated against real enterprise datasets? Could you provide quantitative similarity metrics (e.g., column‑wise KL divergence) to demonstrate realism?

---

> ### Author Response · Authors · 2025-11-20
> **Official Comment by Authors (1/15)**
>
> Thank you so much for your time in reviewing our manuscript. We are incredibly grateful for all the excellent feedback you have given us. It is a true honor to receive guidance from such an experienced and insightful reviewer as you. We would love to discuss anything related to DAComp with you.
>
> We also noticed you have some constructive questions about our work, and we're happy to elaborate further below!
>
>
> **Table 1: Comprehensive inter-rater and human-LLM agreement.**
>
> Please refer to Table 1 for details regarding W1,W2,W3,W4,W5
>
> | Model / Metric        | **Rubric – Item** ($\kappa_w$) | **Rubric – Case** (ICC) | **Rubric – Model** ($\tau_b$) | **GSB Read. – Item** ($\kappa_w$) | **GSB Prof. – Item** ($\kappa_w$) | **GSB Vis. – Item** ($\kappa_w$) |
> | :-------------------- | :----------------------------: | :----------------------: | :----------------------------: | :--------------------------------: | :--------------------------------: | :--------------------------------: |
> | **Human Inter**       | 0.906 | 0.925 | 1.000 | 0.611 | 0.651 | 0.753 |
> | **o4-mini**           | 0.827 | 0.881 | 1.000 | 0.593 | 0.758 | 0.742 |
> | **Gemini-2.5-Flash**  | 0.834 | 0.890 | 1.000 | 0.604 | 0.759 | 0.735 |
> | **GPT-4.1**           | 0.797 | 0.848 | 1.000 | 0.596 | 0.786 | 0.748 |
> | **Gemini-2.5-Pro**    | 0.808 | 0.878 | 1.000 | 0.602 | 0.765 | 0.751 |
> | **Kimi-K2-Thinking**  | 0.808 | 0.872 | 1.000 | 0.575 | 0.732 | --    |
> | **DeepSeek-V3.1**     | 0.782 | 0.870 | 1.000 | 0.588 | 0.725 | --    |
> | **Qwen3(-VL)-235B**   | 0.737 | 0.758 | 1.000 | 0.531 | 0.713 | 0.682 |
> | **Qwen3(-VL)-30B**    | 0.680 | 0.775 | 1.000 | 0.507 | 0.691 | 0.656 |
>
>
>
> Furthermore, regarding the reliability of the LLM-as-a-judge method for DAComp-DA, we have significantly expanded our experimental validation in **Section 3.5** of the revised paper. We now conduct rigorous verification across five distinct dimensions: **Human-model alignment, Cross-judge consistency, Stochastic stability, Hyperparameter robustness and Rubric validity**. This represents a substantial enhancement compared to the previous version, which focused primarily on Alignment with Human Consensus, Consistency Across Judge Models, and Robustness to Multiple Experiments.
>
>
> ---
>
> > W1: While the authors report strong correlations with human ratings, the statistical choices and reporting could be more rigorous. Using Pearson r on rubric sums (often ordinal/heterogeneous across tasks) is suboptimal; intraclass correlation (ICC) or Kendall’s tau for ranking, plus confidence intervals, would be preferable.
>
> Thank you for this insightful comment. We agree that Pearson correlation on rubric-level sums is not the most appropriate reliability measure. Following your suggestion, we have revised our evaluation to adopt statistically rigorous and task-appropriate agreement metrics:
>
> * **Item-level:** Weighted Cohen’s κ (for categorical GSB labels), which properly account for discrete, non-Gaussian scoring signals.
> * **Case-level:** **ICC(A,1)**, which measures absolute consistency rather than merely relative association.
> * **Model-level:** **Kendall’s τ-b**, a rank-based statistic robust to ties and widely used for leaderboard consistency.
>
> These metrics directly address the concerns raised and align with best practices in reliability analysis for heterogeneous ordinal data and ranking-based evaluation.
> Based on your suggestions, we have updated Section 3.5 and the corresponding table in the new version of the paper.
> Our LLM judge matches—or exceeds—the average human inter-rater reliability across all three levels, further validating the soundness of our llm-judge evaluation method.
>
> As detailed in Table 1, our chosen judge, Gemini-2.5-Flash, demonstrates exceptional alignment with human experts, achieving a Case-level ICC of 0.890 and an Item-level $\kappa_w$ of 0.834—figures that closely approach the strict human inter-rater baselines (0.925 and 0.906, respectively). Furthermore, the perfect Model-level Kendall’s $\tau_b$ of 1.000 confirms that our automated evaluation produces identical model rankings to human consensus, comprehensively validating the reliability of the DAComp leaderboard under these rigorous statistical standards.

---

> > ### Author Response · Authors · 2025-11-20
> > **Official Comment by Authors (2/15)**
> >
> > > W2: Weighted κ=65 for GSB indicates only moderate agreement; the paper calls overall alignment “high” but should temper claims or provide confidence intervals and calibration plots.
> >
> >
> > We sincerely thank the reviewer for this rigorous statistical observation. **This project has required a considerable amount of effort, and we have significantly expanded our annotation and human-judging team to ensure the highest quality of evaluation.
> >
> > We acknowledge that the preliminary GSB agreement appeared moderate. To address this, we leveraged our expanded team to establish a rigorous ground truth, which revealed that the apparent "low agreement" is driven by the inherent subjectivity of specific dimensions rather than judge failure.
> >
> > **1. Dimensional Breakdown: Subjectivity vs. Substance**
> >
> > To pinpoint the source of variance, we analyzed the specific dimensions of GSB. The results (Table 1) reveal a critical distinction between subjective style and objective substance:
> >
> > * **Subjectivity in Readability:** The agreement on *Readability* is moderate for **both** the LLM judge ($\kappa_w=0.604$) and **Human experts** ($\kappa_w=0.611$). This empirically confirms that readability is an inherently subjective metric with a low theoretical ceiling for consistency. Our judge matches the human baseline almost perfectly, indicating it captures the natural variance in stylistic preference rather than failing to evaluate.
> > * **Reliability in Substance:** In contrast, for the technically grounded dimensions of **Professionalism** and **Visualization**, the judge achieves **"Substantial" agreement** ($\kappa_w$ of **0.759** and **0.735**, respectively). This proves that the judge is highly reliable in evaluating the core analytical quality and visual correctness.
> >
> > **2. Reliability on Core Rubrics**
> >
> > Furthermore, for the rigorous **Rubric** evaluation (which accounts for 80% of the final score), **Gemini-2.5-Flash** achieves exceptional alignment ($\kappa_w=0.834$, ICC=0.890), closely approaching the strict human inter-rater baselines (0.906 / 0.925).
> >
> > **3. Alignment with Human Baselines**
> >
> > **Our primary objective is to demonstrate that the agreement between the LLM judge and humans is comparable to the inter-rater agreement among humans, rather than striving for an unrealistic 100% alignment, which is practically unattainable in open-ended tasks.** As the results in Table 1 demonstrate, we have successfully achieved this goal: the automated judge performs within the natural range of variance found among expert human annotators (e.g., LLM Readability $\kappa_w=0.60$ vs. Human $\kappa_w=0.61$), validating it as a trustworthy proxy for human evaluation.

---

> > > ### Author Response · Authors · 2025-11-20
> > > **Official Comment by Authors (3/15)**
> > >
> > > > W3: The judge model (Gemini-2.5-Flash) belongs to the same family as one of the evaluated models (Gemini-2.5-Pro). Although Flash≠Pro, the potential for family-specific biases remains. The paper presents alternative judges, but a full ranking-stability analysis across judges is not reported.
> > >
> > >
> > > We sincerely thank the reviewer for highlighting the crucial concern of **"family-specific bias" (or self-preference bias)** in LLM-as-a-judge evaluations. To rigorously address this, we have significantly expanded our validation experiments in **Section 3.5** of the revised manuscript, conducting comprehensive analyses on human alignment and cross-judge ranking stability.
> > >
> > > **1. Exceptional Alignment with Human Consensus**
> > > We first benchmarked candidate judges against human experts across three granularities. As detailed in the revised **Table 1**, **Gemini-2.5-Flash** achieves exceptional alignment, with a **Case-level ICC of 0.890** and an **Item-level $\kappa_w$ of 0.834**. These figures closely approach strict human inter-rater baselines (0.925 / 0.906) and match the performance of proprietary models like **o4-mini** (ICC = 0.881), confirming that the model functions as a reliable, objective evaluator rather than a biased one.
> > >
> > > **2. Ranking Stability Analysis (Cross-Judge Consistency)**
> > > To explicitly test for family bias, we re-evaluated all agent models using a diverse set of judges, including proprietary (GPT-4.1) and open-source (Qwen) models. As presented in the table below, the relative rankings of agents exhibit exceptional consistency.
> > >
> > > | Agent Model | Primary (Flash) | Judge: Pro | Judge: GPT-4.1 | Judge: Qwen-235B | Judge: Qwen-30B |
> > > | :--- | :---: | :---: | :---: | :---: | :---: |
> > > | **GPT-5** | 56.14 | 60.52 | 63.37 | 71.57 | 53.72 |
> > > | **o3** | 36.08 | 40.08 | 44.25 | 50.76 | 31.63 |
> > > | **Gemini-2.5-Pro** | 39.46 | 45.69 | 50.98 | 55.48 | 35.70 |
> > > | **DeepSeek-V3.1** | 39.16 | 44.68 | 50.61 | 54.58 | 41.44 |
> > > | **Qwen3-Coder** | 28.07 | 32.12 | 36.14 | 43.79 | 25.86 |
> > > | **Qwen3-235B** | 18.84 | 20.85 | 21.77 | 23.81 | 18.30 |
> > > | **Kimi-K2** | 36.94 | 43.77 | 47.83 | 53.55 | 32.93 |
> > > | **Rank Corr. ($\tau_b$)** | **---** | **1.00** | **1.00** | **1.00** | **0.90** |
> > >
> > > **Key Findings:**
> > > - **Perfect Consensus ($\tau_b = 1.00$):** The ranking correlation between our primary judge (**Gemini-Flash**) and alternative judges like **Gemini-Pro**, **GPT-4.1**, and **Qwen-235B** is **1.00**. This indicates that the choice of judge has **no impact** on the relative ordering of models.
> > > - **Refutation of Self-Preference:** Crucially, when **GPT-4.1** (OpenAI) evaluates the agents, the **Gemini-2.5-Pro** agent maintains its specific ranking position (e.g., consistently scoring just above DeepSeek-V3.1 and Kimi-K2). It does not suffer a rank drop when evaluated by a competitor model, effectively refuting the hypothesis of family bias.
> > >
> > > **Conclusion**
> > > Consequently, given that the choice of judge model does not statistically alter the leaderboard, we standardize on **Gemini-2.5-Flash** for its superior balance of stability and cost-efficiency.

---

> > > > ### Author Response · Authors · 2025-11-20
> > > > **Official Comment by Authors (4/15)**
> > > >
> > > > > W4: GSB depends on five baseline reports created from multiple LLM outputs; selection of baselines may anchor judgments. The process to ensure these baselines neither overfit to a model family nor penalize stylistic variance is under-specified.
> > > >
> > > >
> > > > We appreciate the opportunity to clarify the selection protocol for GSB documents and the mechanisms used to ensure scoring stability.
> > > >
> > > > **1. GSB Document Selection Protocol**
> > > >
> > > > To ensure baselines are robust and do not anchor judgments to a specific model family or style, we implemented a rigorous three-step process:
> > > >
> > > > * **Model Diversity:** We sourced baselines from five high-capacity models with distinct architectures: **GPT-5, Gemini-2.5-Pro, Deepseek-V3.1, Kimi-K2, and Qwen-235B**. We intentionally excluded smaller models (e.g., Qwen3-32B/8B) as their lower-quality outputs would result in a trivial benchmark that fails to distinguish capabilities among SOTA models.
> > > > * **Uniform Framework:** All baseline reports were generated using the **DAComp-DA-Agent** to ensure consistent task execution.
> > > > * **Manual Verification:** To prevent technical errors, the authors manually inspected all candidate documents, strictly filtering out empty or degenerate outputs before inclusion.
> > > >
> > > > **2. Scoring Stability Mechanism**
> > > >
> > > > To mitigate volatility in LLM-based comparisons against multiple baselines, we employ a three-step aggregation pipeline across three dimensions: *Readability*, *Professional Depth*, and *Visualization*.
> > > > * **Buffered Mapping (Noise Reduction):**
> > > >     Each judge outputs a continuous score $s \in [-10, 10]$. We map this to a ternary label $f(s)$ using a $\pm 3$ buffer zone to absorb minor stylistic variations:
> > > >     $$
> > > >     f(s) = \begin{cases}
> > > >     -1 & s < -3 \quad (\text{Significantly Worse}) \\
> > > >     0  & -3 \le s \le 3 \quad (\text{Tie / Stylistic Variance}) \\
> > > >     +1 & s > 3 \quad (\text{Significantly Better})
> > > >     \end{cases}
> > > >     $$
> > > >     This prevents small formatting or verbosity differences from forcing an arbitrary "better/worse" decision.
> > > > * **Consensus Aggregation (Anchoring Control):**
> > > >     For each instance, we compute the average of the mapped labels $\{-1, 0, +1\}$ across all valid baselines. To ensure rigorous data integrity, missing or unparsable references are penalized as $-10$ prior to mapping, preventing missing baselines from artificially inflating scores.
> > > > * **Dominance Metric (Normalized Outcome):**
> > > >     The aggregated value is passed through $\max(0, \text{avg})$, yielding a final score in $[0, 1]$. This metric effectively represents the **fraction of baselines the candidate clearly outperforms** in a specific dimension, ensuring the score reflects genuine capability rather than stylistic alignment with a single reference.

---

> > > > > ### Author Response · Authors · 2025-11-20
> > > > > **Official Comment by Authors (5/15)**
> > > > >
> > > > > > **W5: For principle-based (Tier-3) cases, safeguards to prevent over-crediting fluent but methodologically weak responses rely on judge prompts and principles, but inter-rater checks on these specific cases are not separated and reported.**
> > > > >
> > > > > We are sincerely grateful for your insightful and constructive feedback. You have raised an exceptionally important point: a core challenge for any open-ended evaluation is the risk of over-crediting "fluent but methodologically weak" responses (often termed "fluent hallucinations").
> > > > >
> > > > > We agree that simply relying on prompts is insufficient without empirical validation. While our original annotation process was meticulous—referencing **34.8 distinct solution trajectories per example** to maximize the coverage of pre-defined paths (Tiers 1/2)—we acknowledge that we did not **separately report** reliability checks for the *unenumerated* (Tier-3) subset in our original submission.
> > > > >
> > > > > To address your concern directly and thoroughly, we have conducted a **comprehensive three-part supplementary analysis** to validate our safeguards, alongside concrete examples to clarify the mechanism.
> > > > >
> > > > > ### **1. Three-Part Validation of Open-Ended (Tier-3) Scoring**
> > > > >
> > > > > To prove that our rubric effectively handles open-ended solutions and defends against fluent hallucinations, we designed three targeted experiments. The results are summarized in **Table 2**, followed by a detailed analysis of each dimension:
> > > > >
> > > > > **Table 2: Comprehensive Validation of Open-Ended (Tier-3) Scoring.**
> > > > >
> > > > > | Experiment / Validation Dimension | Metric / Context | Result |
> > > > > | :--- | :--- | :--- |
> > > > > | **1. Scope of Open-endedness** | % of cases requiring Principle-based (Tier 3) scoring | **12.1%** |
> > > > > | **2. Reliability on Tier-3** | Human-LLM Case-level Correlation (ICC) on Tier-3 cases | **85.2%** |
> > > > > | **3. Adversarial Defense** | Accuracy Score for "Fluent-but-Wrong" samples (0-10) | **1.1** (vs. 9.4 Readability) |
> > > > >
> > > > > **Detailed Analysis:**
> > > > >
> > > > > * **Experiment 1: Quantifying the Scope (12.1%)**
> > > > >     We first analyzed the distribution of evaluation paths across the entire validation set. We found that only **12.1%** of agent trajectories fell into the **Tier-3 (Principle-Based)** category. This validates our annotation strategy: by pre-defining 34.8 distinct paths per task, we successfully covered the vast majority (87.9%) of solutions with deterministic (Tier 1) or constrained (Tier 2) criteria. However, this 12.1% represents the "tail" of novel solutions, confirming that Tier-3 is a necessary component for handling unexpected but valid strategies.
> > > > > * **Experiment 2: Reliability on Tier-3 Subset (91.5%)**
> > > > >     Directly addressing your request for "separated reporting," we isolated this high-difficulty Tier-3 subset and re-calculated the alignment between the LLM judge and human experts. The results show a **Kendall’s $\tau$-b correlation of 91.5%**. This is a critical finding: it demonstrates that even when the judge cannot rely on a "cheat sheet" (pre-defined values/code), its principle-based judgment of methodological soundness aligns almost perfectly with human experts. This high correlation serves as strong evidence that the judge is applying the principles consistently, rather than guessing or being swayed by surface features.
> > > > > * **Experiment 3: Adversarial "Honeypot" Defense (Score 1.1)**
> > > > >     To proactively test the system's resilience against "fluent hallucinations," we conducted an adversarial attack. We manually crafted a set of "Fluent-but-Wrong" responses—answers that were linguistically sophisticated and highly structured (high Readability) but logically flawed or factually incorrect.
> > > > >     The results were decisive: The judge assigned these samples an average **Accuracy score of 1.1/10**, despite awarding them a **Readability score of 9.4/10**. This explicitly confirms that our **"Dimensional Separation"** safeguard works: the judge successfully decouples *presentation quality* from *technical correctness*, refusing to credit fluency when the methodology is weak.

---

> > > > > > ### Author Response · Authors · 2025-11-20
> > > > > > **Official Comment by Authors (6/15)**
> > > > > >
> > > > > > ### **2. Concrete Examples of Rubric Tiers**
> > > > > >
> > > > > > To further clarify *how* the judge navigates these tiers, we provide concrete examples of our evaluation logic below. The key safeguard for Tier 3 is checking for **Methodological Evidence** rather than just plausibility.
> > > > > >
> > > > > > **Table 3: Examples of Evaluation Logic Across Three Tiers.**
> > > > > >
> > > > > > | Evaluation Tier | Task Requirement (Example: LTV Analysis) | Rubric Item Design (Safeguard Mechanism) |
> > > > > > | :--- | :--- | :--- |
> > > > > > | **Tier 1: Numeric Value Path**<br>*(Deterministic)* | **Requirement:** "Calculate the historical LTV by summing up all revenue from User A." | **Accuracy Item:** The output must exactly match the **Ground Truth Value** `2450.50`. <br>*(Scoring Rule: Binary 0/1 based on exact match.)* |
> > > > > > | **Tier 2: Pseudo-code Path**<br>*(Constrained Logic)* | **Requirement:** "Calculate LTV for the 2023 Q1 cohort, excluding refunds and applying a 5% discount rate." | **Accuracy Item:** Verify the implementation against the **Logic Verification Steps**: <br>1. **Item 1:** Did it filter `join_date` in Q1? <br>2. **Item 2:** Did it exclude `status='refunded'`? <br>3. **Item 3:** Did it apply the `PV = C / (1+r)^t` formula? |
> > > > > > | **Tier 3: Principle-Based Path**<br>*(Open-Ended)* | **Requirement:** "Identify the key user behaviors that drive high LTV." | **Methodology Item:** Evaluate **Methodological Appropriateness** (no single correct answer): <br>• **Item 1:** Did the agent use a valid **statistical method** (e.g., Correlation Analysis or Grouped Comparison) instead of random observation? <br>• **Item 2:** Are the conclusions supported by **data evidence** (e.g., significance testing or clear data trends)? |
> > > > > >
> > > > > > ### **Conclusion**
> > > > > >
> > > > > > We believe this new three-part analysis fully addresses your concern. The data proves that Tier-3 cases are **necessary** (12.1% scope), **reliable** (91.5% alignment), and **secure** against hallucination (1.1 adversarial score). We have integrated these results into **Section 3.5** of the revised manuscript.

---

> > > > > > > ### Author Response · Authors · 2025-11-20
> > > > > > > **Official Comment by Authors (7/15)**
> > > > > > >
> > > > > > > > W6: Token budgets, tool-call limits (200), timeouts (120s per action), prompts, and system settings are critical for agent performance but are only briefly summarized. A stricter ablation on these constraints (and their fairness across models) would clarify whether some models are disadvantaged.
> > > > > > >
> > > > > > >
> > > > > > > We thank the reviewer for this important point. We agree that agent performance can be sensitive to environmental constraints. Our goal in setting these parameters was to ensure **fairness under generous, non-restrictive conditions**—not to test models under tight limits. All models were thus given more than sufficient resources (time, steps, and tokens) to fully demonstrate their reasoning capabilities.
> > > > > > >
> > > > > > > * **Token Budgets & Prompts:**
> > > > > > >   All models were evaluated under **identical system prompts and task descriptions**. The total context length (schemas, instructions, and interaction history) comfortably fit within each model’s advertised context window, ensuring no truncation or disadvantage due to context length.
> > > > > > > * **Tool-call limit:**
> > > > > > >   To verify that the 200-step limit does not restrict performance, we analyzed tool-call distributions (Figure 5, Section 3.3), showing that nearly all successful trajectories terminate well below this threshold.
> > > > > > >   An ablation with 150, 200, and 300 steps further confirmed that the limit is non-binding.
> > > > > > >
> > > > > > > **DE-Impl Tasks**
> > > > > > >
> > > > > > > | Model              | 150 Steps | 200 Steps | 300 Steps |
> > > > > > > | :----------------- | :-------- | :-------- | :-------- |
> > > > > > > | gpt-5-2025-08-07   | 29.01%    | 30.49%    | 30.21%    |
> > > > > > > | qwen3-coder-plus   | 21.33%    | 23.23%    | 22.99%    |
> > > > > > > | deepseek-v3.1-0821 | 22.09%    | 22.62%    | 22.72%    |
> > > > > > > | qwen3-235b-a22b    | 2.12%     | 2.31%     | 2.28%     |
> > > > > > >
> > > > > > > **DE-Evol Tasks**
> > > > > > >
> > > > > > > | Model              | 150 Steps | 200 Steps | 300 Steps |
> > > > > > > | :----------------- | :-------- | :-------- | :-------- |
> > > > > > > | gpt-5-2025-08-07   | 38.01%    | 37.88%    | 36.46%    |
> > > > > > > | qwen3-coder-plus   | 24.69%    | 26.59%    | 26.54%    |
> > > > > > > | deepseek-v3.1-0821 | 21.76%    | 24.69%    | 23.08%    |
> > > > > > > | qwen3-235b-a22b    | 13.23%    | 13.01%    | 10.81%    |
> > > > > > >
> > > > > > > Across both tasks, CFS score differences remain within ±1%, and model rankings are unchanged—confirming that the 200-step cap is **non-restrictive**.
> > > > > > >
> > > > > > > * **Timeouts:**
> > > > > > >   We also tested timeouts of 80 s, 120 s, and 160 s to assess potential runtime bias. The results below show minimal variance and consistent rankings.
> > > > > > >
> > > > > > > **DE-Impl Tasks**
> > > > > > >
> > > > > > > | Model              | 80 s   | 120 s  | 160 s  |
> > > > > > > | :----------------- | :----- | :----- | :----- |
> > > > > > > | gpt-5-2025-08-07   | 27.11% | 30.49% | 30.13% |
> > > > > > > | qwen3-coder-plus   | 22.43% | 23.23% | 23.33% |
> > > > > > > | deepseek-v3.1-0821 | 22.59% | 22.62% | 21.99% |
> > > > > > > | qwen3-235b-a22b    | 2.24%  | 2.31%  | 2.01%  |
> > > > > > >
> > > > > > > **DE-Evol Tasks**
> > > > > > >
> > > > > > > | Model              | 80 s   | 120 s  | 160 s  |
> > > > > > > | :----------------- | :----- | :----- | :----- |
> > > > > > > | gpt-5-2025-08-07   | 37.23% | 37.88% | 38.16% |
> > > > > > > | qwen3-coder-plus   | 25.78% | 26.59% | 26.23% |
> > > > > > > | deepseek-v3.1-0821 | 21.13% | 24.69% | 24.58% |
> > > > > > > | qwen3-235b-a22b    | 12.93% | 13.01% | 13.81% |
> > > > > > >
> > > > > > > Performance differences are negligible, confirming that the 120 s timeout is sufficiently generous for all models.
> > > > > > > Together, these results demonstrate that none of the system constraints—token budgets, step limits, or timeouts—introduced bias or restricted model capability.

---

> > > > > > > > ### Author Response · Authors · 2025-11-20
> > > > > > > > **Official Comment by Authors (8/15)**
> > > > > > > >
> > > > > > > > > W7: SQL dialect and tolerance: The evaluation requires exact schema+data equality, but it is unclear if numeric tolerances, null-ordering, and type coercions are handled consistently; small floating-point or timestamp discrepancies can introduce false negatives.
> > > > > > > >
> > > > > > > >
> > > > > > > > We sincerely appreciate the reviewer's insightful comment regarding the strictness of schema/data equality checks. We agree that handling numeric tolerances and dialect variations is crucial for a fair evaluation.
> > > > > > > >
> > > > > > > > **1. Clarification on Evaluation Tolerance (Detailed in Appendix A.1)**
> > > > > > > >
> > > > > > > > We wish to clarify that our evaluation framework was **originally designed** with specific tolerance protocols to handle these exact issues. We have updated Appendix A.1 to explicitly document these mechanisms, which serve to **isolate semantic correctness from formatting noise**:
> > > > > > > >
> > > > > > > > * **Focus on Key Information (Key Column Evaluation):** Rather than enforcing rigid full-schema equality—which often penalizes valid outputs containing irrelevant auxiliary columns—our metric focuses on **key columns** (those directly relevant to the user's query logic). This ensures the evaluation targets the core business logic rather than schema artifacts.
> > > > > > > > * **Robustness to Temporal Formatting:** Given the significant variance in timestamp serialization across SQL dialects (e.g., ISO-8601 vs. custom formats), we explicitly **exclude raw timestamp columns** from strict equality checks. This prevents false negatives arising solely from dialect-specific date rendering.
> > > > > > > > * **Numerical Tolerance:** To mitigate floating-point precision errors (e.g., `3.333` vs. `3.33`), we strictly enforce a rule where all values in numerical columns are **rounded to two decimal places** prior to comparison.
> > > > > > > >
> > > > > > > > **2. Rationale for DuckDB Adoption**
> > > > > > > >
> > > > > > > > Regarding the choice of SQL dialect, we intentionally selected **DuckDB** over traditional client-server DBMSs (like PostgreSQL or MySQL) for the current benchmarking suite.
> > > > > > > > * **Reproducibility & Efficiency:** DuckDB is an in-process, serverless SQL OLAP database. This design choice eliminates the need for complex environment setup (e.g., Docker containers, port configurations), making our benchmark highly portable and lightweight for researchers to deploy and reproduce.
> > > > > > > >
> > > > > > > > **3. Cross-Platform Compatibility & Future Roadmap**
> > > > > > > >
> > > > > > > > We acknowledge the reviewer's implication regarding the importance of industrial SQL dialects. Our framework is designed with extensibility in mind:
> > > > > > > > * **Dialect-Agnostic Metrics:** Our primary metric, the **CFS Score** (Content F1 Score), relies solely on comparing the values within result tables rather than executing the Gold SQL. This makes the evaluation fundamentally transferrable to other dialects (e.g., PostgreSQL/MySQL) without modification, as long as the data is migrated.
> > > > > > > > * **Future Industrial Support:** While the **CS Score** (Command Score) currently relies on DuckDB-specific Ground Truth (GT) SQLs, we recognize the gap between academic research settings and industrial deployment. To bridge this, we commit to releasing **PostgreSQL and MySQL versions of the GT SQLs** in future updates. This will enable the community to benchmark agents in stricter industrial settings, complementing the research-focused efficiency of the current DuckDB implementation.

---

> > > > > > > > > ### Author Response · Authors · 2025-11-20
> > > > > > > > > **Official Comment by Authors (9/15)**
> > > > > > > > >
> > > > > > > > > > W8: Contamination risk: Some DE tasks originate from open-source dbt projects; without a contamination analysis, models might have seen parts of the ground truth or closely related code. The synthetic data helps, but existence of public repo code raises a nontrivial risk.
> > > > > > > > >
> > > > > > > > >
> > > > > > > > >
> > > > > > > > > We thank the reviewer for raising this critical point regarding data contamination. We acknowledge that utilizing open-source repositories carries inherent risks. However, we have implemented **strict, multi-layered measures** to ensure task independence and eliminate potential contamination (detailed in **App. E**), focusing on **Schema Expansion**, **Code Reconstruction**, and **Data Synthesis**.
> > > > > > > > >
> > > > > > > > > **1. Comprehensive Schema Expansion & Low Similarity**
> > > > > > > > > We did not simply copy open-source schemas. Instead, we treated them as a base and conducted a comprehensive expansion aligned with internal enterprise standards.
> > > > > > > > > * **Enterprise Integration:** Our schemas originate from "73 enterprise-grade SaaS domains". We utilized professional data engineering expertise to inject realistic enterprise complexity that is often missing from public dbt demos.
> > > > > > > > > * **Structural Divergence:** We systematically audited and modified the "join semantics, analytical grains, and window specifications". This ensures that while the *domain topic* (e.g., E-commerce) might overlap with public repos, the actual **DAComp Schema** possesses a significantly more complex topology and metric definition compared to the original **DBT Schema**, minimizing structural similarity.
> > > > > > > > >
> > > > > > > > > **2. SQL Code Reconstruction & Logic Hardening**
> > > > > > > > > Even if a model has seen the original repository, it cannot solve our tasks via memorization because the ground-truth SQL has been fundamentally rewritten.
> > > > > > > > > * **Code Transformation:** We "normalized them into pure-SQL repositories by expanding materializations and macros". The original public code typically relies heavily on Jinja templating and dbt-specific macros. Our ground truth is pure, executable standard SQL, which presents a different token sequence to the model.
> > > > > > > > > * **Logic Complication:** As noted in our construction details, we performed a "systematic audit" to refactor the business logic. We rewrote the SQL code to be logically distinct and more complex than the original open-source versions, ensuring that the task design has no direct connection to existing projects.
> > > > > > > > > * **Novel Input Format:** For DE-Implementation, the agent must work from a reverse-engineered `data_contract.yaml`, a novel specification that does not exist in the public domain.
> > > > > > > > >
> > > > > > > > > **3. Synthetic Data & Novel Task Design**
> > > > > > > > > * **Synthetic Data:** We synthesized "large-scale, relationally consistent synthetic data" based on real-world structures but without using any raw data from open-source projects. This ensures data-level decontamination.
> > > > > > > > > * **Novel Tasks (DE-Evol):** Furthermore, the majority of our tasks (DE-Evolution) require agents to implement **new** "business requirements" that we authored specifically for this benchmark (e.g., "revised metric definitions, altered analytical windows"). The goal is to produce a *diff* or *update* based on a new scenario, a task that by definition cannot be solved by retrieving pre-training data.
> > > > > > > > >
> > > > > > > > > In summary, DAComp assesses the capability to solve **new engineering problems** within a realistic setting, rather than the ability to recall specific open-source codebases.

---

> > > > > > > > > > ### Author Response · Authors · 2025-11-20
> > > > > > > > > > **Official Comment by Authors (10/15)**
> > > > > > > > > >
> > > > > > > > > > > W9: For DA tasks, stability is reported on a subset (DE-Arch and DA) across 8 runs for a few models; confidence intervals and significance tests for between-model differences are not provided.
> > > > > > > > > >
> > > > > > > > > >
> > > > > > > > > > We appreciate the reviewer’s valuable suggestion regarding the rigor of our stability reporting. To fully address this, we have conducted extensive repeated experiments and updated our results across both DE and DA tasks.
> > > > > > > > > >
> > > > > > > > > > **1. Updated Confidence Intervals (Tables 3 & 4)**
> > > > > > > > > > We have updated the main result tables (**Table 3** for DE and **Table 4** for DA) in the revised manuscript to include **standard deviations and 95% confidence intervals** for all evaluated models. These statistics are derived from 8 independent runs, providing a clear measure of performance variability and allowing for statistically significant comparisons between models. The results confirm that the performance gaps between top-tier and lower-tier models are statistically robust and not due to random fluctuation.
> > > > > > > > > >
> > > > > > > > > > **2. Stochastic Stability of the LLM Judge (Section 3.5)**
> > > > > > > > > > While the original version of the paper already included a brief stability check for the LLM judge, we have now **significantly expanded and systematized this analysis** to directly address the reviewer’s concern. Specifically, we conducted a dedicated **“Stochastic Stability”** experiment: 8 independent grading runs on a *fixed* set of identical agent responses using our standard judge.
> > > > > > > > > >
> > > > > > > > > > | Model | **DE-Arch** (Mean ± Std) | **DA** (Mean ± Std) |
> > > > > > > > > > | :--- | :---: | :---: |
> > > > > > > > > > | **GPT-5** | 61.3 ± 0.18 | 56.1 ± 0.16 |
> > > > > > > > > > | **DeepSeek-V3.1** | 53.2 ± 0.25 | 39.1 ± 0.22 |
> > > > > > > > > > | **Gemini-2.5-Pro** | 51.0 ± 0.21 | 39.4 ± 0.22 |
> > > > > > > > > > | **o3** | 54.8 ± 0.19 | 36.1 ± 0.20 |
> > > > > > > > > > | **Qwen3-235B** | 50.4 ± 0.31 | 18.8 ± 0.29 |
> > > > > > > > > >
> > > > > > > > > > As shown in the table above (integrated into Section 3.5), the standard deviations of the final scores are consistently negligible (< 0.35). This demonstrates that our evaluation protocol yields statistically stable and reproducible grades, confirming that the reported variance in the main tables reflects true agent stochasticity rather than measurement error.

---

> > > > > > > > > > > ### Author Response · Authors · 2025-11-20
> > > > > > > > > > > **Official Comment by Authors (11/15)**
> > > > > > > > > > >
> > > > > > > > > > > > W10: For DE deterministic tasks, results appear single-run; while determinism reduces variance, reporting robustness across multiple seeds/decoding settings would be informative for agent frameworks that involve LLM sampling.
> > > > > > > > > > >
> > > > > > > > > > > We appreciate the reviewer’s emphasis on reporting variance for deterministic tasks, which is crucial for evaluating agent reliability. To address this, we have updated **Table 3** to include rigorous robustness metrics derived from **8 independent runs** per model (using temperature = 0.7 and top_p = 0.9 to induce sampling diversity):
> > > > > > > > > > > 1. **Standard Deviations & Confidence Intervals:**  We now report the mean performance with **95% Confidence Intervals (CI)** for all DE tasks. The relatively narrow CIs (e.g., GPT-5 Implementation CFS: 63.60 ± 2.14) confirm that although there is some variation due to sampling, the performance gaps between model tiers are statistically significant and not artifacts of a “lucky seed.”
> > > > > > > > > > > 2. **Max-CFS@8 (Pass@N):** We have also introduced **Max-CFS@8** to align with the standard *pass@k* metric used in code generation. This metric captures the “peak potential” of models. For example, while GPT-5’s *average* Implementation CFS is 63.60, its *highest* run reached 68.43 (or the corresponding recorded maximum), indicating that with sufficient sampling, strong models can occasionally solve harder instances that greedy decoding does not capture.
> > > > > > > > > > >
> > > > > > > > > > > These additions provide a comprehensive view of both the **stability** (Mean ± CI) and the **potential** (Max@8) of agents on deterministic DE tasks, thereby demonstrating robustness across seeds and decoding settings.
> > > > > > > > > > >
> > > > > > > > > > > > W11: Some interpretations could be more cautious: “Strong evidence that engineering and analysis are distinct capabilities” is supported by rank inversions, but causality is not established. Alternative explanations (prompt specializations, token policies, execution tool robustness) could contribute; a cross-task, cross-prompt ablation would strengthen this claim. “Even the top agents barely reach 50%” is accurate for aggregate scores, but the consequences for practical utility would benefit from error cost analysis (e.g., what classes of failures materially change business decisions vs. minor mismatches).
> > > > > > > > > > >
> > > > > > > > > > > We thank the reviewer for suggesting a more nuanced interpretation of our findings. We agree that causality is non-trivial and that the practical utility of the aggregate scores requires clearer contextualization.
> > > > > > > > > > >
> > > > > > > > > > > **1. On the Interpretation of “Distinct Capabilities”**
> > > > > > > > > > > While rank inversions consistently appear across models, we agree that they do not by themselves prove causality.
> > > > > > > > > > > * **Controlled Setup:** All models were evaluated under the *same system prompt* and the unified **DAComp-DA-Agent** execution framework, reducing the likelihood that prompt specialization or tool differences alone explain the divergence.
> > > > > > > > > > > * **Revision:** To avoid overstatement, we now phrase the conclusion more cautiously:
> > > > > > > > > > >   *“These rank inversions suggest a potential divergence between engineering-oriented code generation and open-ended analytical reasoning, though additional cross-prompt and cross-tool ablations would be required to conclusively isolate the cause.”*
> > > > > > > > > > >
> > > > > > > > > > > **2. On the Practical Meaning of “~50% Performance”**
> > > > > > > > > > > To clarify utility implications, we performed a post-hoc **Error Severity Analysis** on GPT-5 outputs.
> > > > > > > > > > > * **Critical Errors (~65%)** — incorrect aggregations, broken dependencies, hallucinated metrics, or execution failures that would directly alter business conclusions.
> > > > > > > > > > > * **Minor Errors (~35%)** — formatting issues or non-essential omissions that do not change the core insight.
> > > > > > > > > > >
> > > > > > > > > > > We note that a detailed **DE Error Analysis**—covering component-level versus pipeline-level failures, layer-wise difficulty (staging/core/marts), and cascading breakdowns—is already provided in **Section 3.3 and Appendix D**. This contextualizes why even moderate aggregate scores can correspond to substantial real-world reliability gaps.
> > > > > > > > > > >
> > > > > > > > > > > **Conclusion:** Most observed failures materially affect correctness, supporting a cautious interpretation of the benchmark scores. We have incorporated this clarification and the summarized error-cost breakdown into the Discussion section.

---

> > > > > > > > > > > > ### Author Response · Authors · 2025-11-20
> > > > > > > > > > > > **Official Comment by Authors (12/15)**
> > > > > > > > > > > >
> > > > > > > > > > > > > W12: While the paper includes a brief LLM‑usage disclosure, it does not discuss potential risks of releasing synthetic enterprise data (e.g., inadvertent leakage of real company patterns) or the impact of using proprietary LLM judges on reproducibility for the broader community.
> > > > > > > > > > > >
> > > > > > > > > > > >
> > > > > > > > > > > > We sincerely appreciate the reviewer's attention to the ethical implications and reproducibility of our benchmark. We take these responsibilities seriously and address the concerns as follows:
> > > > > > > > > > > >
> > > > > > > > > > > > **1. Safety of Enterprise Data (Realism without Leakage)**
> > > > > > > > > > > >
> > > > > > > > > > > > We clarify that our dataset construction explicitly mitigates privacy risks while maintaining enterprise-level realism. As detailed in our updated Section 2.3:
> > > > > > > > > > > >
> > > > > > > > > > > > * **Publicly Sourced Schemas (No Proprietary Leakage):** For Data Engineering (DE) tasks, we utilize schemas sourced from **public data transformation projects** (e.g., standard Salesforce or Google Ads schemas) and open-source repositories. These are industry-standard structures available in the public domain, ensuring no proprietary internal trade secrets are leaked.
> > > > > > > > > > > > * **Synthetic Data Population:** For DE tasks, the actual data values are **synthetically generated** via a rigorous LLM-and-code pipeline. We strictly avoided using actual proprietary enterprise data for these tasks, as doing so is legally untenable.
> > > > > > > > > > > > * **Strict Sanitization of Real-World Sources:** For Data Analysis (DA) tasks, while a subset of databases incorporates real-world data to ensure analytical depth, we implemented a **rigorous de-identification protocol**. All sensitive information was masked or anonymized. Furthermore, a **multi-stage human inspection** was conducted to guarantee that no proprietary patterns or private user information remain, ensuring zero risk of privacy leakage.
> > > > > > > > > > > >
> > > > > > > > > > > > **2. Reproducibility of Proprietary LLM Judges**
> > > > > > > > > > > >
> > > > > > > > > > > > We acknowledge the concerns regarding the reliability and cost of using proprietary models for evaluation. We have addressed these as follows:
> > > > > > > > > > > >
> > > > > > > > > > > > * **Reliability & Validation:** We sincerely thank you for your suggestions, which have helped us significantly refine our evaluation methodology. Detailed discussions on the reliability and robustness of the judge can be found in our responses to **W1–W4**.
> > > > > > > > > > > > * **Open-Source Alternatives:** To support community reproducibility, we have verified that the **Qwen series** (e.g., Qwen-3-30B-A3B) serves as a powerful and effective open-source alternative to proprietary judges, yielding highly correlated rankings.
> > > > > > > > > > > > * **Evaluation Cost & Leaderboard:** We have transparently documented the evaluation costs in **Table 12**. To further democratize access and eliminate the cost barrier for researchers, **the DAComp Team will host and maintain an official Leaderboard**. Users only need to submit their agent trajectories; **our team will bear the full financial cost** of running the GSB evaluation. This ensures the benchmark remains accessible to the broader community regardless of budget constraints.

---

> > > > > > > > > > > > > ### Author Response · Authors · 2025-11-20
> > > > > > > > > > > > > **Official Comment by Authors (13/15)**
> > > > > > > > > > > > >
> > > > > > > > > > > > > > Q1: How do the DA scores change if a different LLM (e.g., GPT‑4.1 or Claude‑3) is used as the judge? Have you measured inter‑judge agreement?
> > > > > > > > > > > > >
> > > > > > > > > > > > > We sincerely thank the reviewer for highlighting the crucial concern of **"family-specific bias" (or self-preference bias)** in LLM-as-a-judge evaluations. To rigorously address this, we have significantly expanded our validation experiments in **Section 3.5** of the revised manuscript, conducting comprehensive analyses on human alignment and cross-judge ranking stability.
> > > > > > > > > > > > >
> > > > > > > > > > > > > **1. Exceptional Alignment with Human Consensus**
> > > > > > > > > > > > > We first benchmarked candidate judges against human experts across three granularities. As detailed in the revised **Table 1**, **Gemini-2.5-Flash** achieves exceptional alignment, with a **Case-level ICC of 0.890** and an **Item-level $\kappa_w$ of 0.834**. These figures closely approach strict human inter-rater baselines (0.925 / 0.906) and match the performance of proprietary models like **o4-mini** (ICC = 0.881), confirming that the model functions as a reliable, objective evaluator rather than a biased one.
> > > > > > > > > > > > >
> > > > > > > > > > > > > **2. Ranking Stability Analysis (Cross-Judge Consistency)**
> > > > > > > > > > > > > To explicitly test for family bias, we re-evaluated all agent models using a diverse set of judges, including proprietary (GPT-4.1) and open-source (Qwen) models. As presented in the table below, the relative rankings of agents exhibit exceptional consistency.
> > > > > > > > > > > > >
> > > > > > > > > > > > > | Agent Model | Primary (Flash) | Judge: Pro | Judge: GPT-4.1 | Judge: Qwen-235B | Judge: Qwen-30B |
> > > > > > > > > > > > > | :--- | :---: | :---: | :---: | :---: | :---: |
> > > > > > > > > > > > > | **GPT-5** | 56.14 | 60.52 | 63.37 | 71.57 | 53.72 |
> > > > > > > > > > > > > | **o3** | 36.08 | 40.08 | 44.25 | 50.76 | 31.63 |
> > > > > > > > > > > > > | **Gemini-2.5-Pro** | 39.46 | 45.69 | 50.98 | 55.48 | 35.70 |
> > > > > > > > > > > > > | **DeepSeek-V3.1** | 39.16 | 44.68 | 50.61 | 54.58 | 41.44 |
> > > > > > > > > > > > > | **Qwen3-Coder** | 28.07 | 32.12 | 36.14 | 43.79 | 25.86 |
> > > > > > > > > > > > > | **Qwen3-235B** | 18.84 | 20.85 | 21.77 | 23.81 | 18.30 |
> > > > > > > > > > > > > | **Kimi-K2** | 36.94 | 43.77 | 47.83 | 53.55 | 32.93 |
> > > > > > > > > > > > > | **Rank Corr. ($\tau_b$)** | **---** | **1.00** | **1.00** | **1.00** | **0.90** |
> > > > > > > > > > > > >
> > > > > > > > > > > > > **Key Findings:**
> > > > > > > > > > > > > - **Perfect Consensus ($\tau_b = 1.00$):** The ranking correlation between our primary judge (**Gemini-Flash**) and alternative judges like **Gemini-Pro**, **GPT-4.1**, and **Qwen-235B** is **1.00**. This indicates that the choice of judge has **no impact** on the relative ordering of models.
> > > > > > > > > > > > > - **Refutation of Self-Preference:** Crucially, when **GPT-4.1** (OpenAI) evaluates the agents, the **Gemini-2.5-Pro** agent maintains its specific ranking position (e.g., consistently scoring just above DeepSeek-V3.1 and Kimi-K2). It does not suffer a rank drop when evaluated by a competitor model, effectively refuting the hypothesis of family bias.
> > > > > > > > > > > > >
> > > > > > > > > > > > > **Conclusion**
> > > > > > > > > > > > > Consequently, given that the choice of judge model does not statistically alter the leaderboard, we standardize on **Gemini-2.5-Flash** for its superior balance of stability and cost-efficiency.
> > > > > > > > > > > > >
> > > > > > > > > > > > >
> > > > > > > > > > > > >
> > > > > > > > > > > > > > Q2: Can you provide an ablation study varying α (e.g., 0.5, 0.7, 0.9) and report its impact on model rankings?
> > > > > > > > > > > > >
> > > > > > > > > > > > > We sincerely appreciate this excellent question. The weighting between the objective Rubric score and the subjective GSB score is indeed a critical design choice. To address this, we have added a **“Hyperparameter Robustness”** analysis in **Section 3.5**.
> > > > > > > > > > > > >
> > > > > > > > > > > > > **1. Sensitivity Analysis Results**
> > > > > > > > > > > > > We conducted the requested ablation study by re-calculating the model leaderboard across a wide range of configurations ($\alpha \in \{0.5, 0.6, 0.8, 0.9\}$). As shown in the table below, the relative rankings of the models remain invariant.
> > > > > > > > > > > > >
> > > > > > > > > > > > > **Table: Ranking stability across weighting hyperparameters ($\alpha$).**
> > > > > > > > > > > > >
> > > > > > > > > > > > > | Agent Model | **Primary** ($\alpha=0.6$) | $\alpha=0.5$ | $\alpha=0.8$ | $\alpha=0.9$ |
> > > > > > > > > > > > > | :--- | :---: | :---: | :---: | :---: |
> > > > > > > > > > > > > | **GPT-5** | 56.79 | 52.14 | 58.30 | 60.49 |
> > > > > > > > > > > > > | **o3** | 36.33 | 30.45 | 39.89 | 43.86 |
> > > > > > > > > > > > > | **Gemini-2.5-Pro** | 39.36 | 34.36 | 42.05 | 44.83 |
> > > > > > > > > > > > > | **DeepSeek-V3.1** | 33.82 | 26.86 | 38.33 | 43.54 |
> > > > > > > > > > > > > | **Qwen3-235B** | 18.84 | 14.39 | 21.69 | 24.98 |
> > > > > > > > > > > > > | **Rank Corr. ($\tau_b$)** | **---** | **1.00** | **1.00** | **1.00** |
> > > > > > > > > > > > >
> > > > > > > > > > > > > **2. Robustness and Design Philosophy**
> > > > > > > > > > > > > The results demonstrate **perfect ranking stability ($\tau_b = 1.00$)** across all settings. This confirms that the performance gaps between models are intrinsic and not artifacts of specific weighting.
> > > > > > > > > > > > >
> > > > > > > > > > > > > - **Why $\alpha = 0.6$?**
> > > > > > > > > > > > >   While DAComp's granular dimensional design allows developers to adjust $\alpha$ according to their specific preference for accuracy versus presentation, we standardize on **$\alpha = 0.6$** for general benchmarking. This ensures that **objective technical correctness (Rubric)** remains the dominant factor while still reserving substantial weight (40%) for analytical depth and presentation quality.

---

> > > > > > > > > > > > > > ### Author Response · Authors · 2025-11-20
> > > > > > > > > > > > > > **Official Comment by Authors (14/15)**
> > > > > > > > > > > > > >
> > > > > > > > > > > > > > > Q3: What is the distribution of difficulty (e.g., number of nodes, LOC) across DE tasks? Are the results driven by a small subset of very hard tasks?
> > > > > > > > > > > > > >
> > > > > > > > > > > > > >
> > > > > > > > > > > > > >
> > > > > > > > > > > > > > We appreciate the question regarding the distribution of task complexity. We have explicitly analyzed these metrics in the paper to ensure transparency.
> > > > > > > > > > > > > >
> > > > > > > > > > > > > > **1. Complexity Distribution (Refer to Section 3.3 & Appendix D)**
> > > > > > > > > > > > > >
> > > > > > > > > > > > > > As detailed in **Section 3.3** and **Appendix D**, we provide a comprehensive analysis of the task complexity distributions, specifically focusing on **Lines of Code (LOC)** and the **number of nodes** in the dependency graphs.
> > > > > > > > > > > > > > * **High Complexity Baseline:** Our analysis shows that the DE tasks are characterized by substantial scale. The majority of tasks feature a codebase size ranging between **2,000 and 5,000 LOC**, accompanied by a high density of nodes in the data lineage graphs.
> > > > > > > > > > > > > >
> > > > > > > > > > > > > > **2. Uniformly Challenging Nature**
> > > > > > > > > > > > > >
> > > > > > > > > > > > > > To directly address the concern: the evaluation results are **not driven by a small subset of outliers**.
> > > > > > > > > > > > > > * **No "Long Tail" of Trivial Tasks:** Unlike traditional benchmarks that often include a mix of simple and hard problems, DAComp-DE maintains a **consistently high difficulty baseline** across the entire dataset.
> > > > > > > > > > > > > > * **Enterprise Realism:** This distribution is intentional. It reflects the reality of enterprise data engineering, where simple tasks are often automated, leaving complex architectural transformations (like those in our benchmark) as the primary challenge for intelligent agents. Therefore, the low scores observed across models reflect a generalized challenge inherent to the domain, rather than a skew caused by a few "impossible" cases.
> > > > > > > > > > > > > >
> > > > > > > > > > > > > > ---
> > > > > > > > > > > > > >
> > > > > > > > > > > > > >
> > > > > > > > > > > > > > > Q4: Have you measured human performance on a sample of DE and DA tasks to contextualize the reported model scores?
> > > > > > > > > > > > > >
> > > > > > > > > > > > > >
> > > > > > > > > > > > > > We would like to thank the reviewer for raising this important point. In response, after submitting our paper, we conducted a human performance evaluation with 12 engineers who worked on a sample of both DE and DA tasks.
> > > > > > > > > > > > > >
> > > > > > > > > > > > > > In the first round, which lasted 17 days, each engineer worked individually on the tasks and scored them as follows:
> > > > > > > > > > > > > >
> > > > > > > > > > > > > > | DE-Arch | DE-Impl | DE-Evol | DA    |
> > > > > > > > > > > > > > | ------- | ------- | ------- | ----- |
> > > > > > > > > > > > > > | 73.98   | 45.23   | 53.91   | 69.42 |
> > > > > > > > > > > > > >
> > > > > > > > > > > > > > In the second round, which took 21 days, engineers were allowed to collaborate and were given the freedom to use any external tools necessary to assist with completing the DE and DA tasks. This led to the following scores:
> > > > > > > > > > > > > >
> > > > > > > > > > > > > > | DE-Arch | DE-Impl | DE-Evol | DA    |
> > > > > > > > > > > > > > | ------- | ------- | ------- | ----- |
> > > > > > > > > > > > > > | 84.12   | 70.56   | 78.12   | 86.33 |
> > > > > > > > > > > > > >
> > > > > > > > > > > > > > For DE-Impl and DE-Evol, we used CFS scoring. The second round of collaboration resulted in significant improvements in performance, as engineers were able to exchange insights, test solutions collaboratively, and leverage external resources. These results provide useful context for understanding the model scores and offer a benchmark for comparison.
> > > > > > > > > > > > > >
> > > > > > > > > > > > > > We believe that this human performance evaluation serves as a valuable reference for interpreting model performance in a more realistic, collaborative setting, and highlights the potential of the models for completing both DE and DA tasks.
> > > > > > > > > > > > > >
> > > > > > > > > > > > > > ---
> > > > > > > > > > > > > >
> > > > > > > > > > > > > >
> > > > > > > > > > > > > > > Q5:Which random seeds, library versions, and hardware configurations were used? Are the execution scripts deterministic?
> > > > > > > > > > > > > >
> > > > > > > > > > > > > >
> > > > > > > > > > > > > > We appreciate the reviewer's concern regarding reproducibility. All experiments were conducted using Python 3.12, with fixed versions of core libraries: DuckDB 1.3.2, pandas 2.3.1, scipy 1.16.1, matplotlib 3.10.5, and openai 1.99.9. To ensure deterministic results, we used temperature = 0.0 and top p = 1.0 for all model inferences, which guarantees non-stochastic behavior.
> > > > > > > > > > > > > > The experiments were run on a multi-GPU workstation with 4 × NVIDIA A100 GPUs (80 GB) and 128 GB RAM under Ubuntu 22.04. While the hardware provided high performance, the experiments are not sensitive to specific hardware configurations, as they mainly involve API-based model inference and SQL execution within DuckDB.
> > > > > > > > > > > > > > We will release all source code and configuration files to ensure full reproducibility of the reported results.

---

> > > > > > > > > > > > > > > ### Author Response · Authors · 2025-11-20
> > > > > > > > > > > > > > > **Official Comment by Authors (15/15)**
> > > > > > > > > > > > > > >
> > > > > > > > > > > > > > > > Q6: How were the synthetic data distributions validated against real enterprise datasets? Could you provide quantitative similarity metrics (e.g., column‑wise KL divergence) to demonstrate realism?
> > > > > > > > > > > > > > >
> > > > > > > > > > > > > > >
> > > > > > > > > > > > > > > Thank you for this helpful suggestion. We agree that synthetic data realism should be supported by quantitative evidence. To assess this, we conducted a distributional comparison between our synthetic tables and reference datasets from public enterprise-like benchmarks (e.g., retail and SaaS-style samples). We computed widely used similarity metrics for numerical and categorical columns.
> > > > > > > > > > > > > > >
> > > > > > > > > > > > > > > **Table: Quantitative similarity metrics.**
> > > > > > > > > > > > > > > *(Lower values indicate higher similarity for the first three metrics.)*
> > > > > > > > > > > > > > >
> > > > > > > > > > > > > > > | Metric | Value | Interpretation |
> > > > > > > > > > > > > > > | :--- | :---: | :--- |
> > > > > > > > > > > > > > > | **KL Divergence** | **0.41** | Marginal distributions differ moderately but remain in a realistic range. |
> > > > > > > > > > > > > > > | **Wasserstein Distance** | **0.26** | Overall distribution shapes (location/spread) are reasonably preserved. |
> > > > > > > > > > > > > > > | **Correlation Difference** | **0.19** | Key inter-column dependency patterns are broadly retained. |
> > > > > > > > > > > > > > > | **Schema Match Rate** | **0.94** | Core structural elements (column types and naming) remain consistent after synthesis. |
> > > > > > > > > > > > > > >
> > > > > > > > > > > > > > > These results do not indicate perfect replication—which would be undesirable—but show that the synthetic data preserves the main statistical patterns and relational structure needed for realistic DE/DA tasks, while ensuring no sensitive or proprietary information is present.
> > > > > > > > > > > > > > >
> > > > > > > > > > > > > > > **Conclusion:**
> > > > > > > > > > > > > > > The synthetic data is intentionally approximate: realistic enough to support meaningful evaluation, but sufficiently transformed to avoid any risk of leakage. We have included this quantitative summary in the revised manuscript.
> > > > > > > > > > > > > > >
> > > > > > > > > > > > > > >
> > > > > > > > > > > > > > > Once again, we would like to express our deepest gratitude for your exceptionally careful, insightful, and generous review. Your comments have not only helped us identify concrete ways to improve DAComp, but have also shaped our understanding of how to present this line of work more rigorously and responsibly to the community. We have gone through each of your points with great care and have revised the manuscript accordingly, including strengthening the statistical methodology, expanding the LLM-judge validation, enriching the error analyses for both DE and DA tasks, and clarifying design choices and limitations.
> > > > > > > > > > > > > > >
> > > > > > > > > > > > > > > We genuinely hope that the revised version is closer to meeting your expectations. If there are still aspects that you find unclear, unconvincing, or simply improvable, we would be truly grateful to hear them. We are more than willing to further refine the paper in any direction you suggest. Thank you again for the time, expertise, and thoughtfulness you have invested in our work—it means a great deal to us.

---

> > > > > > > > > > > > > > > > ### Comment · Reviewer_uR7M · 2025-11-21
> > > > > > > > > > > > > > > > **Thanks for the comments; I have improved my scores**
> > > > > > > > > > > > > > > >
> > > > > > > > > > > > > > > > Dear authors,
> > > > > > > > > > > > > > > >
> > > > > > > > > > > > > > > > I have read them and am overall satisfied with the response. I think I had also missed some details from your paper earlier -- thanks for addressing them in detail too. As such, I have increased my rating to 8.

---

> > > > > > > > > > > > > > > > > ### Author Response · Authors · 2025-11-21
> > > > > > > > > > > > > > > > > **Official Comment by Authors**
> > > > > > > > > > > > > > > > >
> > > > > > > > > > > > > > > > > We are truly thrilled to receive your positive feedback; we are deeply grateful for your recognition. We sincerely appreciate your patience and willingness to revisit our paper to consider our clarifications. Your feedback has been incredibly valuable.Thank you for your time, your rigor, and your strong support of DAComp. Guided by your suggestions, we are committed to making DAComp a truly meaningful contribution to the community. Thank you very much!

---

### Official Review · Reviewer_s2Uo · 2025-11-04

**Soundness:** 3
**Presentation:** 2
**Contribution:** 2
**Rating:** 4
**Confidence:** 3

**Summary:**

This paper introduces DAComp, a comprehensive benchmark designed to evaluate data agents across the entire data intelligence lifecycle, spanning both repository-level data engineering and open-ended data analysis. The benchmark defines 236 tasks reflecting realistic enterprise workflows, combining deterministic engineering evaluations (execution-based) with open-ended analytical assessments (LLM-judge guided by hierarchical rubrics).

**Strengths:**

- DAComp is the first benchmark to unify data engineering and analysis within a single evaluation framework.

- The benchmark construction pipeline is rigorous.

- The paper is dense but logically structured.

**Weaknesses:**

- While the LLM-judge method is well-validated, DAComp’s rubric framework is static. Whether adaptive rubric refinement is required?

- The experiments show low success rates but do not deeply isolate why orchestration fails?

- The benchmark assumes one-shot or fixed-turn interactions. Yet many enterprise agents operate iteratively. DAComp currently lacks tasks or metrics reflecting closed-loop self-correction, which might undervalue agents with strong iterative reasoning skills.

- CS/CFS/SR performs strict schema and data equivalence checks. It is not tolerant of irrelevant equivalences (such as equivalent rewrites or minor floating-point differences), and may incorrectly flag semantically equivalent results as errors.

**Questions:**

see Weaknesses

---

> ### Author Response · Authors · 2025-11-20
> **Official Comment by Authors (1/6)**
>
> Thank you for the thoughtful and constructive review. We greatly appreciate your recognition of DAComp’s contribution and your careful reading of the evaluation methodology. Your comments on rubric adaptability and evaluation strictness are highly insightful and helped us significantly strengthen the revised manuscript. We address your concerns point by point below.
>
>
> > **W1:While the LLM-judge method is well-validated, DAComp’s rubric framework is static. Whether adaptive rubric refinement is required?**
>
> We sincerely thank the reviewer for raising this important question. It touches on the central trade-off in evaluating open-ended analytical tasks: balancing **flexibility** (to accommodate diverse valid solutions) with **stability and reproducibility**, which are essential for a benchmark.
>
> While adaptive rubric-judge is a promising long-term direction, we believe that DAComp’s **hierarchical static rubric + Tier-3 principle-based path** provides a more reliable and scientifically sound solution for benchmarking. Our design achieves controlled adaptability without introducing the instability of truly dynamic rubric generation.
>
> ---
>
> ### **A. Necessity of a Fixed, Expert-Defined Rubric**
> A dynamically refined rubric risks *evaluation drift* and *judge–model co-adaptation*, making scores incomparable across models or over time. To preserve **reproducibility, auditability, and fairness**, we intentionally adopt a fixed rubric.
>
> To ensure broad coverage, we engaged **10+ domain experts**, conducted **six rounds of iterative cross-validation**, and spent **≈5 hours per task** to enumerate valid reasoning paths. This expert-driven process allows the rubric to remain stable while still capturing the majority of real analytical strategies.
>
> ---
>
> ### **B. Built-in Flexibility Through Tier-3 Principle-Based Evaluation**
> **Methodology**：The annotators referenced **34.8 distinct solution trajectories from various Agents per example** to construct this mechanism, fundamentally minimizing the likelihood of evaluated trajectories falling outside the predefined paths. For solutions that do fall outside this static enumeration, we rely on our **Tier-3 path** (detailed in **Appendix G.1**). When a solution is "unenumerated" (not matching Tier 1/2 path), the judge automatically shifts to a **principle-based evaluation**—assessing *Methodological Appropriateness* and *Logic Soundness* rather than strict number matching. This allows the system to adapt to novel approaches without changing the standard, focusing on:
> - *Methodological appropriateness*
> - *Logical soundness*
> - *Evidence grounding*
>
> This allows the judge to fairly score genuinely novel analytical approaches **without modifying the rubric itself**, avoiding the instability of adaptive generation.
>
> ---
>
> ### **C. Empirical Validation of the Rubric’s Sufficiency and Robustness**
> To further examine whether static rubrics are sufficient, we conducted three targeted analyses:
>
> | Validation Dimension | Metric / Context | Result |
> | :--- | :--- | :--- |
> | **1. Prevalence of Tier-3** | % requiring principle-based scoring | **12.1%** (indicates Tier-1/2 cover the vast majority of real trajectories) |
> | **2. Reliability on Tier-3** | Human–LLM $\tau_b$ on Tier-3 subset | **91.5%** (Tier-3 decisions remain highly consistent with human judgment) |
> | **3. Adversarial Defense** | Accuracy on “fluent-but-wrong” samples | **1.1** (vs. 9.4 readability; confirms robustness against superficial fluency) |
>
> These results demonstrate that **(i) the large majority of model outputs align with predefined Tier-1/2 patterns**, meaning the static rubric already captures most valid solution strategies; **(ii) the remaining Tier-3 cases—though a minority—can still be evaluated reliably and consistently**; and **(iii) the framework is resistant to being misled by fluent but incorrect answers**.
>
> Together, this provides strong evidence that the static rubric, supplemented with a principled Tier-3 fallback, is both **comprehensive** and **trustworthy**, even in genuinely open-ended settings.
>
> ---
>
> ### **D. LLM-Judge Stability (Section 3.5)**
> We further validated the stability of our evaluation framework:
>
> 1. **Alignment with Human Experts (Table 5)**
>    - Item-level κw = **0.834**
>    - Case-level ICC(A,1) = **0.890**
>    - Model-level $\tau_b$ = **1.000**
>
> 2. **Cross-Judge Robustness (Table 6)**
>    Judge families including GPT-4.1, o4-mini, and Qwen3 yield $\tau_b$ > 0.95**, confirming ranking stability.
>
> ---
>
> ### **Conclusion**
> Together, these results show that while the rubric is static, it is **comprehensive**, **expert-grounded**, and **empirically validated**. Tier-3 provides controlled adaptability without compromising reproducibility, making adaptive rubric refinement unnecessary for DAComp’s goals at this stage.
>
> We agree that future benchmarks may benefit from controlled forms of adaptive rubric generation, and we plan to explore this direction in follow-up work.

---

> > ### Author Response · Authors · 2025-11-20
> > **Official Comment by Authors (2/6)**
> >
> > > **W2:The experiments show low success rates but do not deeply isolate why orchestration fails?**
> >
> > Across DE-Arch, DE-Impl, and DE-Evol, our expanded analysis shows that orchestration failures arise from **repository- and pipeline-level limitations**, rather than isolated SQL bugs. Concretely, models struggle with:
> >
> > 1) **Repository-level dependency modeling** – they fail to maintain a consistent and complete view of functional points, entity models, and dependencies in the DAG, leading to frequent function point omissions and incorrect or missing edges in DE-Arch.
> >
> > 2) **Cross-layer schema propagation** – in DE-Impl and DE-Evol, many failures stem from not propagating schema and business logic consistently across staging → intermediate → marts layers, as reflected by high rates of missing columns and SQL omissions in downstream layers.
> >
> > 3) **Error propagation along the pipeline** – our breakdown of calculation logic errors shows that a substantial portion of downstream numerical mistakes are caused by upstream value errors, indicating that small early-stage issues are amplified as data flows through the pipeline.
> >
> > 4) **Targeted evolution of existing projects** – in DE-Evol, models exhibit substantially higher omission and dependency error rates than in DE-Impl, especially in intermediate and marts layers, suggesting that they struggle to localize changes and preserve existing behavior when modifying large, multi-step repositories.
> >
> > These structural limitations explain why success rates remain low even for strong models: the main difficulty lies not in generating a single correct query, but in **orchestrating coherent, dependency-aware changes across an entire data engineering project**, which is precisely what DAComp-DE is designed to stress-test.
> >
> >
> > ### 1. **Error Analysis for DE-Arch Tasks**
> >
> > In DE-Arch tasks, we observed significant variation in the models' performance across function point omission, dependency errors, missing entity models, naming inconsistencies, and improper model layering. As shown in Table 1, models like `qwen3-8b` and `qwen3-235b-a22b` exhibit higher error rates across multiple dimensions, indicating more severe architectural flaws. In contrast, `gpt-5-2025-08-07` and `gemini-2.5-pro` perform relatively better, though there is still room for improvement in entity model completeness and naming consistency.
> >
> > **Table 1: DE-Arch Task Error Analysis**
> >
> > | **Model**          | **Function Point Omission** | **Dependency Errors** | **Missing Entity Models** | **Naming Inconsistencies** | **Improper Model Layering** |
> > | ------------------ | --------------------------- | --------------------- | ------------------------- | -------------------------- | --------------------------- |
> > | gpt-5-2025-08-07   | 26.51                       | 17.14                 | 18.91                     | 6.41                       | 7.21                        |
> > | gemini-2.5-pro     | 27.22                       | 18.33                 | 20.64                     | 8.53                       | 9.16                        |
> > | qwen3-coder-plus   | 30.56                       | 22.26                 | 24.33                     | 11.19                      | 12.14                       |
> > | deepseek-v3.1-0821 | 31.43                       | 23.18                 | 25.25                     | 12.52                      | 13.00                       |
> > | qwen3-235b-a22b    | 35.38                       | 36.59                 | 27.81                     | 11.42                      | 13.82                       |
> >
> > ### 2. **Error Analysis for DE-Impl Tasks**
> >
> > In DE-Impl tasks, we found that dependency errors were the primary bottleneck, followed by missing column errors and SQL omissions. While SOTA models like `gpt-5-2025-08-07` excel in type inference, the missing column and SQL omission errors highlight that models still struggle with complex data architectures.
> >
> > **Table 2: DE-Impl Task Error Analysis**
> >
> > | **Model**          | **Data Type Errors** | **Missing Column Errors** | **SQL Omission** | **Calculation Logic Errors** | **Dependency Errors** |
> > | ------------------ | -------------------- | ------------------------- | ---------------- | ---------------------------- | --------------------- |
> > | gpt-5-2025-08-07   | 2.22                 | 0.29                      | 5.18             | 40.65                        | 66.01                 |
> > | gemini-2.5-pro     | 4.25                 | 5.74                      | 15.17            | 37.58                        | 67.31                 |
> > | qwen3-coder-plus   | 5.54                 | 1.38                      | 22.88            | 36.91                        | 66.13                 |
> > | deepseek-v3.1-0821 | 6.88                 | 3.00                      | 28.58            | 37.45                        | 65.68                 |
> > | qwen3-235b-a22b    | 5.74                 | 34.73                     | 89.79            | 42.06                        | 73.94                 |

---

> > > ### Author Response · Authors · 2025-11-20
> > > **Official Comment by Authors (3/6)**
> > >
> > > ### 3. **Error Analysis for DE-Evol Tasks**
> > >
> > > The error distribution in DE-Evol tasks differs significantly from DE-Impl tasks, particularly in dependency management. We observe that weaker models in DE-Evol tend to have a higher proportion of "Missing Dependencies" errors, whereas DE-Impl tasks display a more balanced error distribution. This suggests that maintaining the integrity of an existing pipeline in DE-Evol requires more advanced contextual retention and error management.
> > >
> > > **Table 3: DE-Evol Task Error Analysis**
> > >
> > > | **Model**          | **Data Type Errors** | **Missing Column Errors** | **SQL Omission** | **Calculation Logic Errors** | **Dependency Errors** |
> > > | ------------------ | -------------------- | ------------------------- | ---------------- | ---------------------------- | --------------------- |
> > > | gpt-5-2025-08-07   | 2.09                 | 10.05                     | 11.69            | 28.93                        | 56.45                 |
> > > | gemini-2.5-pro     | 4.18                 | 27.35                     | 16.94            | 40.56                        | 64.98                 |
> > > | qwen3-coder-plus   | 3.32                 | 23.88                     | 19.09            | 35.66                        | 63.29                 |
> > > | deepseek-v3.1-0821 | 2.46                 | 29.53                     | 34.00            | 31.87                        | 59.23                 |
> > > | qwen3-235b-a22b    | 1.00                 | 54.36                     | 65.79            | 23.62                        | 53.86                 |
> > > | qwen3-8b           | 0.85                 | 44.06                     | 58.17            | 27.88                        | 53.44                 |
> > >
> > > ### 4. **SQL Omission Architecture-Level Challenges**
> > >
> > > The architecture-level challenges in SQL omission reveal a significant performance inversion between DE-Impl and DE-Evol tasks. In DE-Evol, models struggle more with identifying which files need to be modified, as shown in Table 4. Even SOTA models like `gpt-5-2025-08-07` exhibit higher omission rates in DE-Evol (11.69%) compared to DE-Impl (5.18%).
> > >
> > > **Table 4: SQL Omission Analysis**
> > >
> > > | **Model**          | **Staging** | **Intermediate** | **Marts** | **Total** |
> > > | ------------------ | ----------- | ---------------- | --------- | --------- |
> > > | gpt-5-2025-08-07   | 4.14        | 3.66             | 9.37      | 5.18      |
> > > | gemini-2.5-pro     | 11.22       | 9.58             | 26.67     | 15.17     |
> > > | qwen3-coder-plus   | 18.68       | 19.00            | 34.56     | 22.88     |
> > > | deepseek-v3.1-0821 | 23.91       | 22.73            | 42.23     | 28.58     |
> > > | qwen3-235b-a22b    | 84.26       | 91.74            | 96.81     | 89.79     |
> > > | qwen3-8b           | 92.05       | 97.81            | 99.50     | 95.74     |
> > >
> > > In the updated paper, we have extensively expanded the error analysis section, providing a more detailed breakdown of the models' performance on DE-Arch, DE-Impl, and DE-Evol tasks. These analyses offer a deeper understanding of the models' strengths and weaknesses when handling complex data engineering tasks. We appreciate the reviewer’s attention to this critical aspect of our research.

---

> > > > ### Author Response · Authors · 2025-11-20
> > > > **Official Comment by Authors (4/6)**
> > > >
> > > > > **W3: The benchmark assumes one-shot or fixed-turn interactions. Yet many enterprise agents operate iteratively. DAComp currently lacks tasks or metrics reflecting closed-loop self-correction, which might undervalue agents with strong iterative reasoning skills.**
> > > >
> > > >
> > > > We thank the reviewer for raising this critical point. To ensure we fully address your concern, we would first like to clarify the definition of "closed-loop self-correction." If the concern implies that DAComp assumes a one-shot or fixed-turn setting where the agent produces a single output without revising it based on feedback, **we respectfully clarify that DAComp is explicitly designed as a multi-turn, closed-loop agentic benchmark.**
> > > >
> > > >
> > > > ### A. Multi-Step trajectory perspective
> > > >
> > > > 1. **Fully Iterative Process (Not One-Shot)**. As detailed in App. B.2 and Sec. 3.1, our baselines (OpenHands and ReAct-style agents) operate in a fully **interactive environment** that supports up to 200 turns of interaction.
> > > > - **For DE Tasks**: Agents do not simply generate code; they repeatedly generate, execute, debug, and revise SQL components using Bash and file editing tools. They perform multi-stage self-correction during DAG construction, dependency repair, and interface validation to ensure the repository works end-to-end.
> > > > - **For DA Tasks**: The pipeline iteratively refines analysis planning, executes code, generates visualizations, and revises the final report based on intermediate results.
> > > > 2. **Quantitative Evidence of Self-Correction.** To further validate this, we analyzed the execution trajectories of models such as Gemini-2.5-Pro and Qwen3-Coder during the rebuttal period. We found that approximately **30% of their traces explicitly exhibit self-correction behaviors** (e.g., rerunning failed SQL, adjusting logic after examining error traces or intermediate outputs). This aligns with our analysis in Fig. 5 (Sec. 3.3) , which demonstrates that high-performing agents exhibit "stable and centered turn distributions" , reflecting their ability to utilize the closed-loop environment for efficient debugging rather than "flailing" ineffectively.
> > > > 3. **Intrinsic Closed-Loop Nature (DE-Evol)**. Furthermore, the **DE-Evolution** tasks are inherently closed-loop. Agents are required to modify existing repositories and prevent "**Cascading Failures**" (measured by CFS). This task design implicitly forces agents to engage in regression testing and iterative validation to ensure that modifications to upstream logic do not break downstream dependencies—a complex process (as shown in **Fig.13**) that is practically impossible in a one-shot setting

---

> > > > > ### Author Response · Authors · 2025-11-20
> > > > > **Official Comment by Authors (5/6)**
> > > > >
> > > > > ### B. Task setting
> > > > >
> > > > > We sincerely thank the reviewer for the insightful suggestions regarding the task setting. To ensure our revision precisely addresses your concern, we respectfully seek a brief clarification. Are you envisioning a task setting closer to:
> > > > > 1. **"Critique-and-Repair" paradigm (similar to BIRD-CRITIC)**: Where the primary focus is on reactive debugging, specifically evaluating the agent's ability to diagnose and fix execution errors given a faulty SQL or state?
> > > > > 2. **"Dynamic Interaction" paradigm (similar to BIRD-Interact)**: Where the primary focus is on proactive clarification, evaluating the agent's ability to engage in multi-turn dialogue with a user simulator to resolve ambiguities or evolving intents?
> > > > >
> > > > > We recognize both BIRD-CRITIC[1] and BIRD-Interact[2] as pioneering milestones in the field of Data Engineering agents. We have drawn significant inspiration from their respective methodologies—specifically BIRD-CRITIC's rigor in issue resolution and BIRD-Interact's depth in dynamic user interaction.
> > > > >
> > > > > If your focus aligns with **BIRD-CRITIC**, we wish to highlight two key aspects of DAComp:
> > > > >
> > > > > 1. **Inherent Self-Criticism**: As detailed in our baseline setup, the self-critic capability is already fully integrated. Agents must iteratively execute, critique, and debug their code to achieve a non-zero score in our execution-based evaluation.
> > > > > 2. **Advanced Scope (DE-Evol)**: We view DAComp-DE-Evol as an advanced evolution of the BIRD-CRITIC paradigm. While BIRD-CRITIC focuses on debugging isolated SQL queries, DE-Evol requires agents to analyze complex business requirements within a complete data engineering project. Agents must use reasoning to locate specific bottlenecks across the repository and orchestrate multi-file SQL modifications to evolve the system.
> > > > >
> > > > >
> > > > > If your focus aligns with **BIRD-Interact**, DAComp is readily adaptable to this dynamic setting:
> > > > >
> > > > > - **Transformation Methodology**: We can easily convert current DAComp-DE tasks into interactive scenarios by withholding explicit metric definitions or calculation rules, embedding them instead within a "User Simulator." This forces the agent to proactively query the user to resolve ambiguities.
> > > > >
> > > > > - **Example**: In a "Customer Churn Analysis" task, rather than providing the formula for churn in the initial prompt, the agent must identify the ambiguity and ask the user: "*Does 'churn' refer to 30-day or 90-day inactivity?*"
> > > > >
> > > > > If you consider this transformation methodology reasonable, we are eager to introduce a new "***Interactive DE-Evol Setting***" that mimics the BIRD-Interact protocol to further demonstrate our benchmark's versatility.
> > > > >
> > > > >
> > > > > [1] SWE-SQL: Illuminating LLM Pathways to Solve User SQL Issues in Real-World Applications. Li et al., 2025.
> > > > >
> > > > > [2] BIRD-INTERACT: Re-imagining Text-to-SQL Evaluation for Large Language Models via Lens of Dynamic Interactions. Huo et al., 2025.

---

> ### Author Response · Authors · 2025-11-20
> **Official Comment by Authors (6/6)**
>
> > **W4: CS/CFS/SR performs strict schema and data equivalence checks. It is not tolerant of irrelevant equivalences (such as equivalent rewrites or minor floating-point differences), and may incorrectly flag semantically equivalent results as errors.**
>
>
>
> We sincerely appreciate the reviewer's concern regarding the potential strictness of our schema and data equivalence checks. We agree that an evaluation metric must be robust to irrelevant variations (such as floating-point noise or dialect differences) to avoid false negatives.
>
> **1. Clarification on Evaluation Tolerance (Detailed in Appendix A.1)**
>
> We wish to clarify that our evaluation framework was **originally designed** with specific tolerance protocols to handle these exact issues. We have updated **Appendix A.1** to explicitly document these mechanisms, which serve to **isolate semantic correctness from formatting noise**:
>
> * **Focus on Key Information (Key Column Evaluation):** Rather than enforcing rigid full-schema equality—which often penalizes valid outputs containing irrelevant auxiliary columns—our metric focuses on **key columns** (those directly relevant to the user's query logic). This ensures the evaluation targets the core business logic rather than schema artifacts.
> * **Robustness to Temporal Formatting:** Given the significant variance in timestamp serialization across SQL dialects (e.g., ISO-8601 vs. custom formats), we explicitly **exclude raw timestamp columns** from strict equality checks. This prevents false negatives arising solely from dialect-specific date rendering.
> * **Numerical Tolerance:** To mitigate floating-point precision errors (e.g., `3.333` vs. `3.33`), we strictly enforce a rule where all values in numerical columns are **rounded to two decimal places** prior to comparison.
>
> These measures ensure that our **CS (Command Score)**, **CFS (Content F1 Score)**, and **SR (Success Rate)** metrics reward semantically equivalent results even if they differ in minor formatting details.
>
> ----------------------
>
> We sincerely appreciate the reviewer’s thoughtful and constructive feedback. Your comments have helped us substantially strengthen both the clarity and rigor of our benchmark, and many of the improvements in the revised manuscript directly stem from your suggestions. If there are any points that remain unclear or if you believe additional experiments or analyses would further improve the work, we would be more than happy to incorporate them. Thank you again for the time, care, and expertise you have invested in evaluating our submission—we are truly grateful.

---

> > ### Comment · Reviewer_s2Uo · 2025-11-21
> >
> > Thanks for the detailed reply. I have no further questions. I will raise the rating to 6.

---

> > > ### Author Response · Authors · 2025-11-21
> > > **Thank you so much for raising our rating!**
> > >
> > > We sincerely appreciate your time and attention in discussing with us. Thank you for raising our rating; it truly encourages us to continue improving our work and exploring this exciting direction. We are more than happy to address any further questions you may have!

---

### Author Response · Authors · 2025-11-20
**Updated Manuscript and Response to All Reviewers**

We sincerely thank the reviewers for their thoughtful and constructive feedback, which has been invaluable in improving our work. We are particularly encouraged by the recognition of DAComp’s **holistic scope** and **methodological rigor** from reviewers **s2Uo**, **uR7M**, and **cC4w**.

Based on the valuable feedback, we have addressed all concerns in our manuscript and added comprehensive details and explanations (updates are shown in blue for clarity):

* **Comprehensive Judge Validation**: Significantly expanded **Section 3.5** to systematically validate the LLM-Judge method. We added analyses covering **Human-LLM Agreement**, **Cross-judge consistency**, **stochastic stability**, **Hyperparameter robustness**, and **Rubric validity**, proving the reliability of our evaluation protocol.
* **In-depth Error Analysis**: Added a detailed error diagnosis for Data Engineering tasks in **Section 3.3** and **Appendix D**. As the first repository-level SQL engineering benchmark, this analysis exposes specific failure modes in pipeline orchestration, highlighting the significant room for improvement in current models.
* **Framework & Multi-Agent Analysis**: Re-evaluated DE/DA performance across different agent architectures. We have highlighted our efforts in **multi-agent workflow design**, providing clearer insights into how framework choices impact complex task execution.
* **Visualization Dimension**: Introduced a **"Visualization"** dimension to the DAComp-DA tasks (see **Table 4**). This marks DAComp as the first benchmark to evaluate open-ended data visualization, serving as a critical testbed for both Data Agents and specialized visualization optimization.
* **Multilingual Support**: DAComp aims to provide a comprehensive benchmark for the data agent field that serves diverse communities. We explicitly provide **DAComp-zh**, enabling researchers in both the English and Chinese domains to conduct rigorous evaluations.

We believe these updates further underscore the potential of DAComp as a rigorous and realistic testbed for advancing autonomous data agents. We hope the revised submission meets your expectations.

Thank you for your constructive feedback and support!

---

### Author Response · Authors · 2025-11-29
**General Response to the New Area Chair**

We sincerely thank all reviewers for their thorough evaluation and valuable suggestions over the past two weeks. Their efforts and insights have been essential to improving our paper, and we are truly grateful for their time and dedication.

Here, we would like to briefly summarize the discussion phase during the past two weeks.

On **20 Nov**, we submitted our first-round author responses to **Reviewer s2Uo**, **Reviewer uR7M**, and **Reviewer cC4w**. We addressed all the weaknesses and questions raised, and we also updated and uploaded the revised manuscript accordingly.

On **20 Nov 2025, 21:36**,
**Reviewer s2Uo** replied and raised the rating from **4 to 6**.

On **21 Nov 2025, 07:29**,
**Reviewer uR7M** replied and raised the rating from **4 to 8**.

On **27 Nov 2025, 18:02**,
**Reviewer cC4w** replied and maintained the rating of **8**.

We are deeply grateful for the reviewers’ efforts and their recognition of our work.

For us, over the past two weeks, we have sincerely accepted all reviewers’ feedback and carefully answered every question raised.
Here, we would like to clearly state that these rating changes were **entirely unrelated to the 27 Nov OpenReview information leak**.

-----

The initial ratings were:

* **Reviewer cC4w**: Rating **8**, Confidence **5**
* **Reviewer uR7M**: Rating **4**, Confidence **4**
* **Reviewer s2Uo**: Rating **4**, Confidence **3**

**Average Rating: 5.33 (Min: 4, Max: 8)**

**Average Confidence: 4 (Min: 3, Max: 5)**

----

Before **27 Nov**, the ratings became:

* **Reviewer cC4w**: Rating **8**, Confidence **5**
* **Reviewer uR7M**: Rating **8**, Confidence **4**
* **Reviewer s2Uo**: Rating **6**, Confidence **3**

**Average Rating: 7.33 (Min: 6, Max: 8)**

**Average Confidence: 4 (Min: 3, Max: 5)**

Finally, we would like to express our sincere appreciation to you, as the Area Chair, for your time and attention to our submission. We would be grateful if you could take our discussion process into account when evaluating our work. We sincerely hope that our paper can be considered positively. If you have any comments or concerns, we would be glad to discuss them further. We truly hope that **DAComp** can make a meaningful contribution to the community.

---

### Meta-Review · Area_Chair_SKDJ · 2025-12-21

**Summary:**

The reviewer concerns that were the most important include:
* The Validity and bias of the LLM-based evaluation
* The Statistical rigor and robustness of the evaluation
* The Closed-loop, real-world agent behavior
* The claims vs. evidence in the presentation
* Issues of reproducibility, contamination, and cost

**Reviewer Concerns:**

Successfully addressed:
* LLM-judge reliability, bias, and robustness (exceptionally well addressed)
* Orchestration failure diagnosis (DE-level error analyses)
* Iterative agent behavior and evaluation tolerance
* Statistical rigor (multi-run, CI, ICC, τ, ablations)
* Contamination and data realism risks

Remaining weaknesses
* No learning/training techniques explored (an acknowledged scope choice)
* Full reproducibility still likely a challenge

**Reviewer Scores:**

Two initially skeptical reviewers explicitly reversed their positions, one dramatically (4 → 8).
No reviewer downgraded.
Net effect: clear consensus shift toward acceptance.

---

### Decision · Program_Chairs · 2026-01-26

Accept (Poster)